# The functional landscape of alternative splicing in hematopoietic lineage commitment

Xiao Hu[1,8], Jinrui Wang[1,8], Li Chen[2,8], Qin Yang[1,8], Manuel Tardaguila[3,8], Bin Mao[1], Shenghui Niu[1], Zijie Xu[1], GuiHua Wang[1], Dan Zhang[1], Yating Zhang[1], Zhen Zhou[1], Jing Luo[1], Zhifeng He[1], Defu Liu[1], Chao Tang[2], Nicole Soranzo[3,4,5,6,7,9]✉, Jing-wen Lin[2,9]✉, Da Jia[1,9]✉ & Lu Chen[1,9]✉

Alternative splicing (AS) is a ubiquitous post-transcriptional regulatory mechanism, that has greatly expanded the transcriptomic and proteomic diversity in vertebrates. While gene regulation of hematopoiesis has been extensively researched in vertebrates, the functions of species- and cell lineage-specific splice variants in vertebrates are largely unknown. Here, we curate transcriptomic data on fetal hematopoietic organ development in six vertebrates and hematopoietic cell differentiation in humans and mice. To identify functional exon-skipping events among thousands of cassette exons in protein-coding genes for a specific differentiation lineage and species, we develop a machine-learning model interrogating 19 features including dynamic expression, protein structure, and evolutionary conservation, and integrate them into a single prediction score, named Functional AS Score (FAScore). Using FAScore, we identify four previously-uncharacterized functional AS events in which deletion of the AS exon leads to defects in erythropoiesis and myelopoiesis. Furthermore, we demonstrate that deletion of exon 15 of *TBC1D23* reduces erythropoiesis in mice and zebrafish through elevated binding capacity to RANBP2/RANGAP1 leading to increased SUMOylation level of *HDAC1*. Collectively, our study presents a valuable tool to identify functional exon skipping (ES) events during hematopoietic lineage commitment, and establishes a research paradigm that can be broadly applied to other biological processes.

Alternative splicing (AS), a key post-transcriptional regulatory mechanism, increases the complexity of eukaryotic transcriptomes and proteomes. High-throughput transcriptomic analyses unveiled that 95% of multi-exonic genes undergo AS in human[1]. AS not only contributes to transcript diversity but also plays a critical role in regulating essential physiological processes, including organ development including brain[2–4] and sexual organs[5,6], as well as cell differentiation processes[2–9] such as hematopoiesis[10–13]. In line with this,

abnormal AS can influence disease states[14–16] and consequently understanding the role of AS is essential for developing gene repair-based therapeutic strategies.

Hematopoiesis has been extensively studied as a model for understanding stem cell biology and developmental processes. In mammals, hematopoiesis begins during the embryonic stage, with hematopoietic stem cells (HSCs) first emerging in the aorta-gonad-mesonephros (AGM) region. As development proceeds, these cells

migrate to the fetal liver (FL), and then to bone marrow (BM). Hematopoiesis is the process by which various blood cell lineages (collectively referred to as 'lineage' thereafter) are produced. We categorize the mature blood cells derived from the same progenitor cells into three lineages, including erythroid, myeloid and lymphoid lineage. The erythroid lineage, originating from megakaryocyte-erythrocyte progenitor (MEP), encompasses erythrocytes and megakaryocytes. The myeloid lineage, arising from granulocyte-monocyte progenitor (GMP), comprises morphologically phenotypically, and functionally distinct cell types, such as granulocytes, macrophages, and monocytes. The lymphoid lineage, derived from common lymphoid progenitor (CLP), includes T cells, B cells, and natural killer (NK) cells. AS plays significant roles in regulating the functional diversity of hematopoietic cells in human and mouse, including HSC formation[7,9,17], erythropoiesis[18,19], megakaryopoiesis[19,20] and granulopoiesis[21]. Despite extensive efforts, only a small fraction of the AS introduced transcript diversity has been functionally validated as it remains experimentally challenging.

The advancement of machine learning (ML) algorithms provided new solutions to predict the functions of isoforms[22–29]. However, previous ML approaches were mainly based on static genomic information, such as conservation, sequence motif, protein domain, structure or AS level in certain cell types, tissues, organs or diseases[17,30,31], and lacked dynamic expression information of the isoforms. Cross-species dynamics for genes[32], lncRNAs[33], and AS[34] has proven to be a powerful framework to assess their evolutionary conservation in mammalian organ development. Thus, incorporating data on both dynamic and static features may offer a better strategy to screen for functional AS. Here, we developed a machine-learning model called functional AS score (FAScore), interrogating 19 features, including dynamic expression pattern, evolutionary conservation, and protein structural information. Since ES events represents the most prevalent type of AS and has been reported to contribute more functional events compared to other types, we focused our model on this AS type. FAScore method used (a) PU-learning coupled with rebalancing methods to address the challenges of limited positive sets, absence of negative sets, and data imbalance, and (b) Random Forest (RF) to screen for functional AS involved in organ development and hematopoietic differentiation. Each AS event has a single FAScore of functional relevance in a given differentiation lineage and species. As an example of the application of FAScore, we identified four AS events that were important for specific hematopoietic lineage commitment.

## Results

### Dynamically regulated alternative splicing atlases of vertebrate hematopoiesis

To study the evolution of alternative splicing of hematopoietic lineage commitment, we integrated 271 bulk RNA-seq samples spanning hematopoietic stem and progenitor cells (HSPCs) and fetal hematopoietic organ (FHO) development. The HSPC dataset includes 17 cell types belonging to three lineages in mouse bone marrow (BM)[35] and human umbilical cord blood[17] (Supplementary Data 1). We also analyzed datasets of the fetal liver, the major fetal hematopoietic organ[36] across five mammals (human, rhesus, rabbit, rat and mouse)[32]. In addition, we included the caudal hematopoietic tissue (CHT) which is the embryonic hematopoietic organ in zebrafish from 36 h post fertilization (hpf) to 4 days post fertilization (dpf)[37] (Supplementary Data 1, and Fig. 1a).

We identified and quantified alternatively spliced exons and AS events as described in MeDAS (Metazoan Developmental Alternative Splicing database)[38]. The percent spliced-in index (PSI) for each exon or 'exon part' (exon-centric PSI, ePSI) was calculated. AS types were subsequently inferred using SUPPA2[39] by comparing exon regions with reference annotations (see "Methods"). Additionally, we

identified 1,640,242 splicing junctions (SJs), of which ~11.9% were unannotated canonical SJs (Supplementary Fig. 1a). Exon skipping (35.7%) was the most frequent AS types (Supplementary Fig. 1b). On average, 17,110 cassette exons (from 2953 to 32,400), originating from 6390 protein-coding genes (from 1824 to 9015), were contained across species and lineages (Supplementary Data 2). Given the large numbers of cassette exons and more functional ES events were documented than other AS types, we focused on ES in the subsequent analyses.

Genes or AS events that are dynamically regulated during developmental or cell differentiation trajectories are more likely to play critical roles in these biological processes, as their temporal regulation often reflects direct functional involvement in key cellular transitions or fate decisions[34]. To quantify the dynamic changes in gene expression and alternative splicing, we computed a series of feature scores based on the expression levels or PSI values across developmental time points or differentiated cell types (Fig. 1b). Specifically, we calculated the expression breath (Gene.Range and AS.Range) to assess the magnitude of change, the correlation coefficient (Gene.cor and AS.cor) and its statistical significance (Gene.cor.p and AS.cor.p) to evaluate the association with temporal or differentiation trajectories, the linear regression slope (Gene.slope and AS.slope) and its statistical significance (Gene.slope.p and AS.slope.p) to determine the rate and statistical significance of the progressive changes, as well as tau (Gene.tau and AS.tau) to measure expression specificity (Fig. 1b, see "Methods"). As a representative example, the MEIS1, a transcription factor whose inactivated impaired hematopoietic stem cell niche development, megakaryocyte and platelet deficiencies in mice[40–42]. We quantified six features of the dynamic gene expression of MEIS1 in human erythroid lineage. It showed a gradually decreased expression in hematopoiesis with an upregulation in megakaryocyte (Fig. 1c), consistent with a previous report[43]. Meanwhile, MEIS1 contains two ES events, the exon (E) 1 and 12. The MEIS1-E12 exhibits high level of changes (Fig. 1d), consistent with its function in inducing the fate decision of human hematopoietic progenitors toward megakaryocyte-erythroid progenitors (MEPs)[44]. In contrast, the MEIS1-E1 showed a stable expression across cell types with low values of dynamic features (Fig. 1e).

To facilitate intuitive assessment of the dynamics of genes and AS events, we integrated the dynamic features described above into a unified dynamism score (DyScore). Based on its bimodal distribution, we applied Gaussian Mixture Modeling (GMM) to classified the genes or AS events into "dynamic" and "non-dynamic" (Supplementary Fig. 1c, e, see "Methods"). Overall, 27% of AS events and 56% of genes were classified as dynamic (Supplementary Fig. 1d, f). Notably, in both human and mouse, dynamically regulated AS events were more likely to be cell lineage-specific, with 63% dynamic in only one hematopoietic lineage compared to 29% of dynamically expressed genes (Supplementary Fig. 1g). Moreover, dynamically regulated AS exhibited significantly higher evolutionary conservation scores in the regions flanking the splice sites across all lineages and fetal hematopoietic organ (paired t-test, $p < 1.5 \times 10^{-4}$, Supplementary Fig. 1h). These results suggest that dynamically regulated AS events may play specialized and conserved regulatory roles in hematopoietic lineage, highlighting the importance of splicing dynamics in the spatio-temporal control of transcriptome complexity during hematopoiesis.

### Predictive machine learning model for functional alternative splicing

To screen for functional AS in hematopoiesis, we extracted features other than transcriptomic dynamics including sequence conservation, structural homologs and integrity, domain integrity, transmembrane helices, signal peptide and target peptide. Using these features, we developed a machine-learning model named FAScore (Functional AS

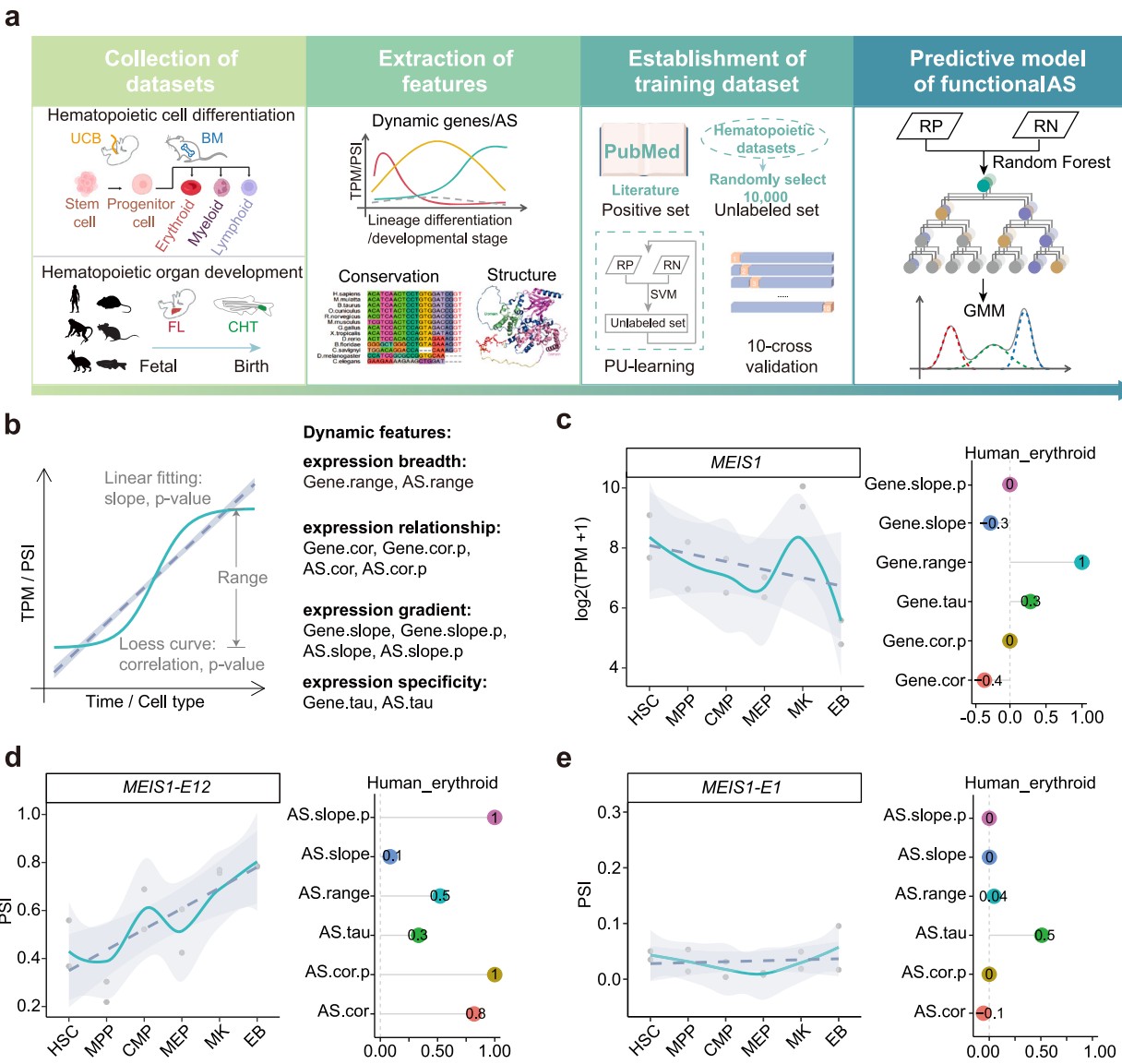

**Fig. 1 | The examples of dynamically regulated gene and AS events. a** The FAScore prediction model involves four key steps: (1) Collecting RNA-Seq datasets from hematopoietic cell differentiation and organ development across multiple species, including human umbilical cord blood (UCB), mouse bone marrow (BM), fetal liver (FL) from five species, and zebrafish caudal hematopoietic tissue (CHT). (2) Extracting AS events with 19 features grouped into three predictive categories. (3) Building a training set by curating known functional AS events from PubMed as positives and selecting 10,000 unlabeled events internally. A positive-unlabeled (PU) learning approach with 10-fold cross-validation identified reliable positive (RP) and negative (RN) sets for downstream model training. (4) Training the model with Random Forest on these sets, using a Gaussian Mixture Model (GMM) to classify AS events as functional, nonfunctional, or uncertain based on functional scores. Created in BioRender. Xiao, X. (https://BioRender.com/vy5f2hi). Created in BioRender. Xiao, X. (https://BioRender.com/9rkd6ct). Created in BioRender. Xiao, X. (https://BioRender.com/y079uo4). **b** Schematic representation of dynamic profiling for gene expression and AS. Blue dashed and green solid lines indicate the fitted mean values (linear regression and LOESS, respectively), with shaded bands showing the 95% confidence intervals. Derived features include: range of genes (Gene.range) or AS (AS.range) to assess variation breadth; correlation coefficient (Gene.cor, AS.cor) and its *p*-value (Gene.cor.p, AS.cor.p) to quantify association with temporal or differentiation trajectories; linear regression slope (Gene.slope, AS.slope) and its *p*-value (Gene.slope.p, AS.slope.p) to estimate rate and significance of change; and tau (Gene.tau, AS.tau) to evaluate specificity. **c–e** Expression and splicing patterns of *MEIS1* and its two AS events during human erythroid differentiation. **c** Expression values (transformed TPM) of *MEIS1* (Left) and the values of dynamic features (Right). **d, e** Percent spliced in (PSI) values (Left) and dynamic feature scores (Right) for the exon 12 skipping event in transcript *MEIS1-201* (d) and the exon 1 skipping event in transcript *MEIS1-202* (e). Blue dashed and green solid lines indicate the fitted mean values (linear regression and LOESS, respectively), with shaded bands showing the 95% confidence intervals.

Score) to predict functional AS events in a given differentiation lineage and species (Fig. 1a). For each AS event, the model assigns a species- and lineage-specific FAScore, designated as species_lineage_FAScore (e.g., Human_erythroid_FAScore, Mouse_erythroid_FAScore). Due to the lack of a verified negative set and the limited number of known functional AS events with experimental evidence (37 unique exons, with redundancies due to cross-species conservation and multi-lineage functionality, Supplementary Data 3, see Methods), we employed a Positive-Unlabeled (PU) Learning framework to identify reliable positive and negative sets for model training (see Methods). It defined a positive set (4803) and a negative set (5245) to form the training dataset (Supplementary Fig. 2a). A comparative analysis of feature scores among the gold-standard positives, the reliable positives, and the reliable negatives revealed minimal differences between the gold-standard and reliable positive sets, but significant differences between the reliable positives and negatives, supporting the effectiveness of

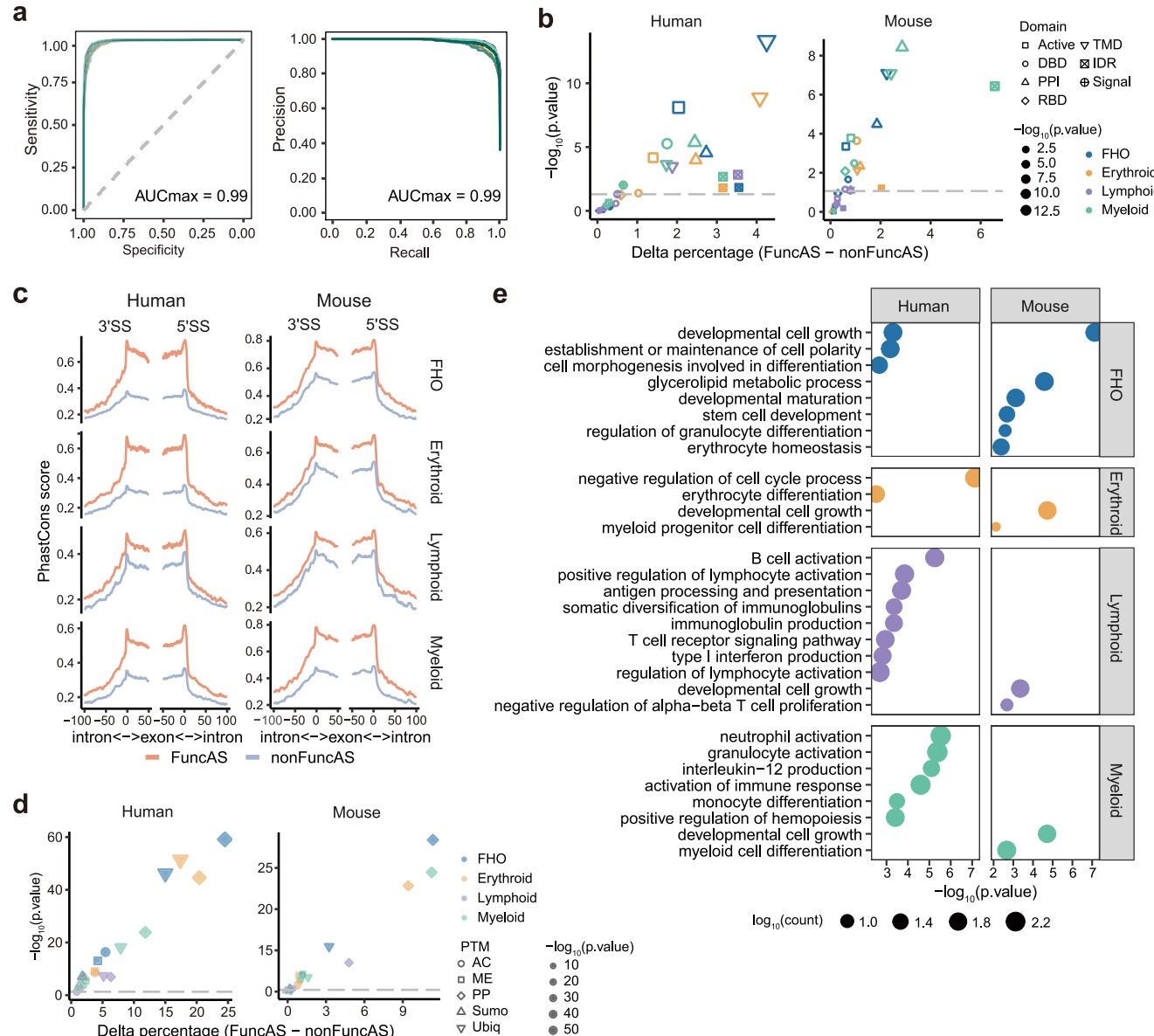

**Fig. 2 | Comparison analysis of functional and non-functional AS events.**
**a** Evaluation of FAScore prediction performance using Receiver Operating Characteristic (ROC, left) curve and precision-recall curve (PRC, right). The grey diagonal line represents random classifier performance. AUCmax: the maximum area under the curve value from 10-fold cross-validation. **b** Volcano plots illustrate significant protein domain differences between functional and non-functional annotated AS events in humans (Left) and mice (Right). The x axis shows percentage difference, while the y-axis displays -$\log_{10}$(p-value) from two-sided proportion tests. Colors denote distinct datasets, and point shapes indicate different protein domain types, including Active (Active site), DBD (DNA binding domain), PPI (Protein-protein interaction domain), RBD (RNA binding domain), TMDs (Transmembrane domains), IDR (Intrinsically disordered regions), and Signal (Signal peptide). The grey dashed line indicates statistical significance (p-value = 0.05). **c** Comparison of conservation scores (PhastCons) surrounding splice sites (3'SS and 5'SS) between functional and non-functional AS of annotated type in different lineages. The x axis shows position relative to splice sites (0), covering intronic (-100 bp) and exonic

(+50 bp) regions. The y axis shows the average PhastCons scores for each site of functional and non-functional AS. **d** Volcano plot showing important post-translational modification (PTM) sites contained in the exonic region of annotated events in humans (Left) and mice (Right). The x axis shows the percentage difference of PTM sites between functional and non-functional AS, while the y axis presents -$\log_{10}$(p-value) from the two-sided proportion test. Colors represent distinct datasets, with point shapes indicating types of PTM sites, including AC (Acetylation), ME (Methylation), PP (Phosphorylation), SUMO (SUMOylation), and Ubiq (Ubiquitination). The grey dashed line indicates statistical significance (p-value = 0.05). **e** Bubble plot shows the GO pathways analysis of functional AS-harboring genes using cell lineage-specific background sets across FHO development and lineage differentiation (including erythroid, lymphoid, and myeloid) in human (Left) and mice (Right). The dot size indicates the gene count. Statistical significance was determined by a one-sided hypergeometric test followed by Benjamini-Hochberg correction. The x-axis represents the significance of enrichment (p value < 0.05, p.adjust < 0.1).

PU-learning (Supplementary Fig. 2b). Compared to Naive Bayes (AUC: 0.94), Logistic Regression (0.96), Support Vector Machine (SVM, 0.98), AdaBoost (0.98) and CART (0.90), the Random Forest (RF) model demonstrated the best performance (0.99). Therefore, we developed a RF model on this constructed training set and selected the best-performing model by 10-fold cross-validation (AUC-ROC = 0.993,

AUC-PR = 0.992, F1 score = 0.952, Rank index = 0.243, FNR = 0.050) (Fig. 2a, and Supplementary Fig. 2c). We assessed the contribution of each feature to the prediction by calculating the relative importance score using Gini coefficient[45]. The features contributing the most to our model were dynamic splicing (AS.slope.p, 21% and AS.cor.p, 12%), sequence conservation cross-species (14%) and protein domain

(Domain.integrity, 10%) (Supplementary Fig. 2d). Our model integrates multiple features into a unified predictive score, offering a comprehensive selection criterion to screen for functional AS which may not be identified focusing on features of conservation or protein domain (Supplementary Fig. 3a). For example, using conservation as the sole criterion would exclude low-conservation yet functional AS events found in the known positive set, such as *Kdm1a-E7*[46] and *EHMT2-E10*[47] (Supplementary Fig. 3b). Moreover, the conservation and FAScore are weakly related as there are events with high conservation but having low FAScore and events with high FAScore with low conservation (Supplementary Fig. 3c, d).

Next, we measured the FAScores for annotated and unannotated AS events from fetal hematopoietic organ development and hematopoietic cell differentiation per species (Supplementary Data 2), that were not used in the model training deriving a continuous score for AS, ranging from 0 to 1, with 1 as most likely to be functional. By fitting a three-component GMM, we defined the following classes by probability density intersection method: 'functional AS' (FuncAS), 'non-functional AS' (NonfuncAS), and 'uncertain AS' (Uncertain) (Supplementary Fig. 4a). In total, we predicted 30,615 functional AS (16.3%), 53,454 non-functional (55.3%), and 104,133 uncertain (28.4%) (Supplementary Fig. 4b, and Supplementary Data 2). Furthermore, we compared the 19 features among these three groups and observed significant differences in features between the predicted functional and non-functional sets, consistent with the observation in the model training (Supplementary Fig. 4c).

We propose that AS events with highly conserved splicing sites and harboring functional domains or modification sites are more likely to play a role in complex biological processes such as hematopoiesis. To evaluate the prediction outcomes, we compared the predicted functional set (FuncAS) and non-functional set (NonfuncAS) from four distinct aspects, including protein domain, conservation, modification sites and functional pathways of harboring genes. First, we observed that compared to the non-functional, the functional AS were more likely to be annotated with functional domains in the UniProt database[48] such as transmembrane domain, active sites, protein-protein interaction, intrinsic disorder region (prop.test, $p < 0.05$) (Fig. 2b, and Supplementary Fig. 5a). Second, the sequences around the splicing junctions were more conserved in functional AS than in the non-functional AS events assessed by PhastCons score[34] (Fig. 2c, and Supplementary Fig. 5b). Third, functional AS were more likely to harbor post translational modification (PTM) sites annotated in PhosphoSitePlus[49], such as methylation, phosphorylation and ubiquitination (prop.test, $p < 0.05$) (Fig. 2d, and Supplementary Fig. 5c).

Finally, we found the genes that harbour at least one functional AS event (FuncAS-harbouring genes) tended to be enriched in splicing factors (SF), transcription factors (TF) and RNA binding proteins (RBP) (Supplementary Fig. 5d). GO analysis of these genes are associated with pathways related to the lineage, as 'erythroid differentiation' was prominently enriched in the erythroid lineage; 'regulation of lymphocyte activation' was enriched in the lymphoid lineage; and 'myeloid cell differentiation' along with 'neutrophil activation' were enriched in the myeloid lineage (Fig. 2e). In particular, for fetal hematopoietic organ development, the FuncAS-harbouring genes were enriched in pathways related to both liver development and hematopoiesis. Gene set enrichment analysis (GSEA) also showed that FuncAS-harbouring genes were enriched with transcription factors (TFs) of the myeloid, such as *KLF4*, *IRF8*, *STAT1* and *STAT3* (Supplementary Data 4, and Supplementary Fig. 5e). These results suggest the accuracy of our prediction model.

## FAScore predicts cell lineage-specific AS events

To identify conserved AS, we next mapped the genomic locations of 42,452 AS events hosted in 11,075 genes identified in the human dataset and found 16,627 AS events had at least one corresponding AS region in other vertebrate species. These AS events were then classified as primate-specific (age class 1), mammal-specific (class 4) or ancient (present in all vertebrates, class 5, dating back 429 million years ago (MYA) (Fig. 3a).

The creation of novel AS exons occurred fairly often during vertebrate evolution. For example, the 1,833 primates-specific AS events were only present in humans and macaques but not in other species (Fig. 3a). Using FAScores, we observed a marked increase in the proportion of predicted functional AS events in class 5 compared with the other classes (Fig. 3b), but not in non-functional AS exons. In total, 6,748 AS events from 3,073 genes were identified as conserved in vertebrates (class 5). It is reported that older genes have more critical functions[50], and our results suggest that ancient AS events may also play essential roles (Fig. 3b).

Hematopoiesis in the vertebrates takes place at various times and anatomical locations, including fetal liver/CHT (defined as FHO) and BM. Analyzing the spatiotemporal expression patterns of the conserved AS, we identified 24 clusters from the 6,748 vertebrate-conserved AS events (age class 5) using Mfuzz[51] (Supplementary Fig. 6a, and Supplementary Data 5). About 27.2% of AS events were functional exclusively in FHO (C2 = 1,837), while 8.8% (C6, C12 and C17, 597) were functional in all three lineages postnatally in BM (Supplementary Fig. 6a). Meanwhile, 40.1% AS events were cell lineage-specific in hematopoiesis, including 19.4% myeloid-specific (C4, C5, C15 and C22), 9.6% erythroid-specific (C8, C11 and C21) and 11.1% lymphoid-specific (C9, C10 and C14) (Fig. 3c, Supplementary Fig. 6a). These findings indicated that key AS events are likely dynamically regulated during lineage commitment and in different anatomical locations.

Next, we selected candidates with top FAScores from different cell lineages of mouse or human for experimental validation (Fig. 3d). For the erythroid lineage, *TBC1D23-E15* (Human_erythroid_FAScore: 0.998) and *EPB41L1-E14* (Mouse_erythroid_FAScore: 0.995) were chosen. For the myeloid lineage, *KLF6-E3* (Human_myeloid_FAScore: 0.997) and *SSBP3-E6* (Mouse_myeloid_FAScore: 0.915) were chosen. And for the lymphoid lineage, *FYN-E7* (Mouse_lymphoid_FAScore: 0.987) was selected (Supplementary Fig. 6b). Given that *FYN-E7* has been reported to play an important role in T cell signal transduction in mice[52], the other four events were experimentally analyzed. *SSBP3-E6* and *KLF6-E3* are more frequently included in the myeloid lineage (average PSI: 0.27 and 0.92, respectively) compared to the erythroid (average PSI: 0.08 and 0.77) and lymphoid (average PSI: 0.10 and 0.75) lineages (Fig. 3e). In contrast, *TBC1D23-E15* is more likely to be included in the erythroid lineage (average PSI: 0.78), while tends to be skipped in the lymphoid (average PSI: 0.32) and myeloid (average PSI: 0.39) lineages (Fig. 3e). We first validated that all of the candidate exons were all transcribed in the colony-forming unit-granulocyte-macrophage (GM), the colony-forming unit-granulocyte-erythrocyte-monocyte-megakaryocyte (GEMM) and the burst-forming unit-erythrocyte (BFU-E) colonies derived from colony-forming unit (CFU) assays (Supplementary Fig. 6c). To validate that they are functional in given lineages for which they have high FAScores, we performed CFU assays on the mouse HSPCs in which the candidate exons were deleted using GFP-expressing retroviral vector having two sgRNAs targeting the flanking intronic regions of the exons (Fig. 3f). The transduced cells with GFP expression were sorted and plated for CFU assays. The CFU assays revealed that the deletion of *SSBP3-E6* or *KLF6-E3* in mouse c-kit⁺ cells resulted in a reduced formation of myeloid colonies by 58% and 38% respectively, while the erythroid colonies were of wildtype level (Fig. 3g). Similarly, the CFU assays showed that the deletion of the *Tbc1d23-E15* inhibited erythroid colony formation by 35% without a significant impact on myeloid populations (Fig. 3g). *EPB41L1-E14* which exhibits a higher inclusion rate in mature myeloid cells (Fig. 3e) and the deletion of *Epb41l1-E14* markedly enhanced erythropoiesis with an increase of 38% of BFU-E colonies while no significant difference on the myeloid colonies was observed (Fig. 3g). In summary, FAScore

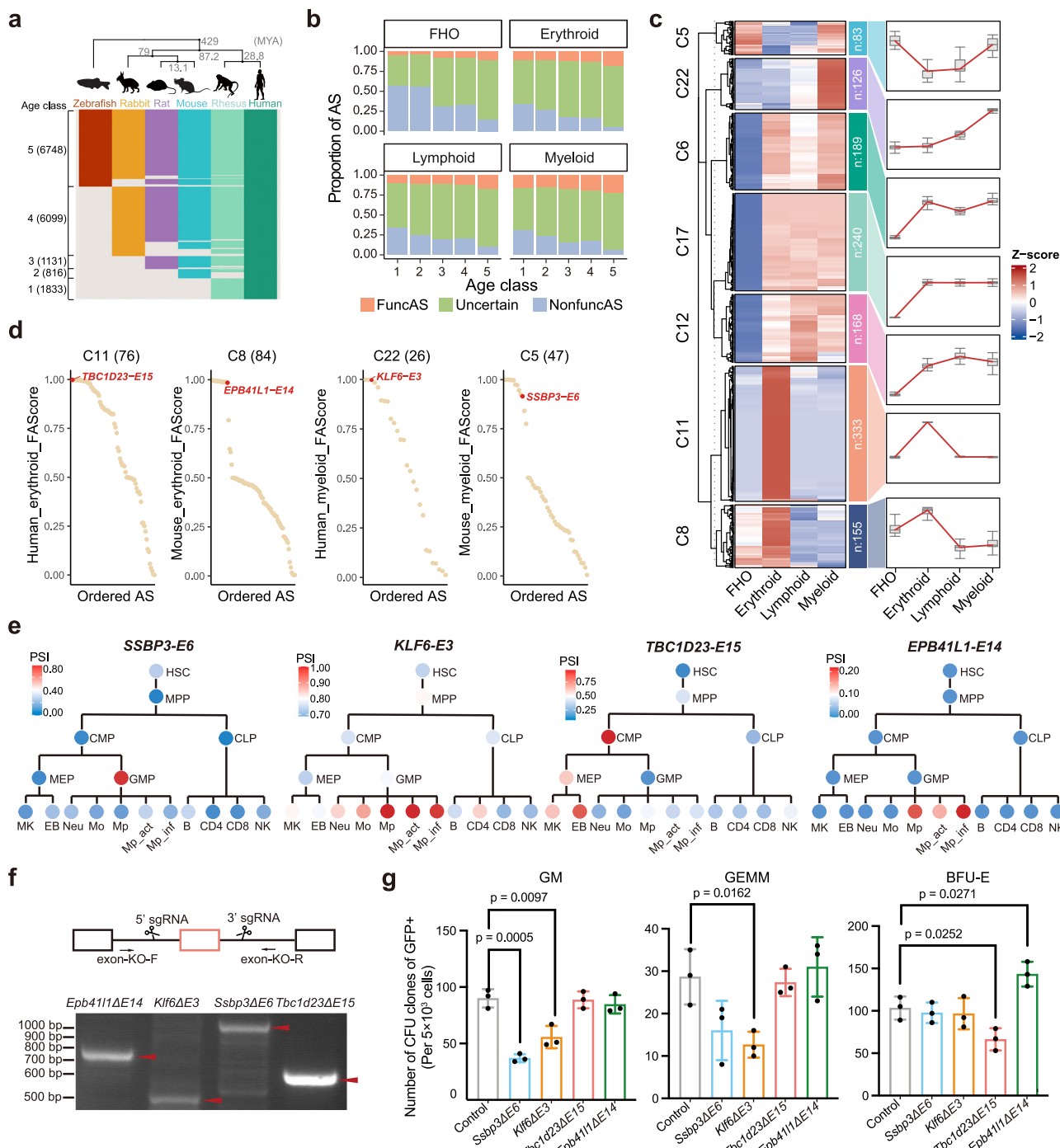

**Fig. 3 | FAScore predicts key AS events that are dynamically regulated during lineage commitment. a** Phylogenetic tree of species used for evolutionary dating (branch lengths in million years, MYA; Top). Presence/absence patterns and counts of AS exons by age class across species (Bottom). **b** Percentage of AS exons categorized as functional, non-functional, or uncertain across evolutionary age classes in human. **c** Heat map showing seven FAScore patterns of AS during FHO development and lineage differentiation (Left). Line chart illustrating corresponding trends for each pattern (Right). Box plots show median (center line) and interquartile range (IQR). Whiskers indicate ±1.5 × IQR. (n: AS number). **d** Rank-ordered functional AS scores for ES events in four clusters (11, 8, 22, 5). AS events are ranked by descending FAScore (0.00 to 1.00). Candidate AS events highlighted in red, and cluster sizes in parentheses. **e** Schematic tree view of PSI values for *EPB41L1* exon 14 (*EPB41L1-E14*), *TBC1D23* exon 15 (*TBC1D23-E15*), *KLF6* exon 3 (*KLF6-E3*), *SSBP3* exon 6 (*SSBP3-E6*) skipping events derived from RNA-seq data of human hematopoietic differentiation. *HSC* Hematopoietic stem cell, *MPP* Multipotent progenitor, *CLP*

Common lymphoid progenitor, *CMP* Common myeloid progenitor, *GMP* Granulocyte monocyte progenitor, *MEP* Megakaryocyte erythrocyte progenitor, *EB* Erythroblasts, *MK* Megakaryocytes, *Neu* Neutrophil, *Mo* Monocyte, *Mp* Macrophage, *Mp_act* Activated macrophage, *Mp_inf* Inflammatory macrophage, *B* B cell, *CD4* CD4⁺ T cell, *CD8* CD8⁺ T cell, *NK* Natural killer cell. **f** Schematic of CRISPR/Cas9-mediated exon deletion in c-kit⁺ cells from Rosa-Cas9 knock-in mice, achieved via retroviral delivery of sgRNAs targeting the flanking introns (5′ sgRNA and 3′ sgRNA, indicated by scissors) (Top). PCR was performed using exon-KO-F and exon-KO-R primers to confirm the target exon deletion (*Epb41l1ΔE14*, *Klf6ΔE3*, *Ssbp3ΔE6*, and *Tbc1d23ΔE15*). Red arrows indicate the correct bands, all confirmed by Sanger sequencing (Bottom). Representative results from three independent experiments are shown. **g** Erythroid-myeloid differentiation potential of 5000 control or mutant c-kit⁺ cells (*Epb41l1ΔE14*, *Tbc1d23ΔE15*, *Klf6ΔE3*, and *Ssbp3ΔE6*) from (**f**). Colony types quantified: BFU-E, GM, and GEMM colonies (*n* = 3 mice). (Unpaired two-tailed Student's t-test, mean ± s.d). Source data are provided as a Source Data file.

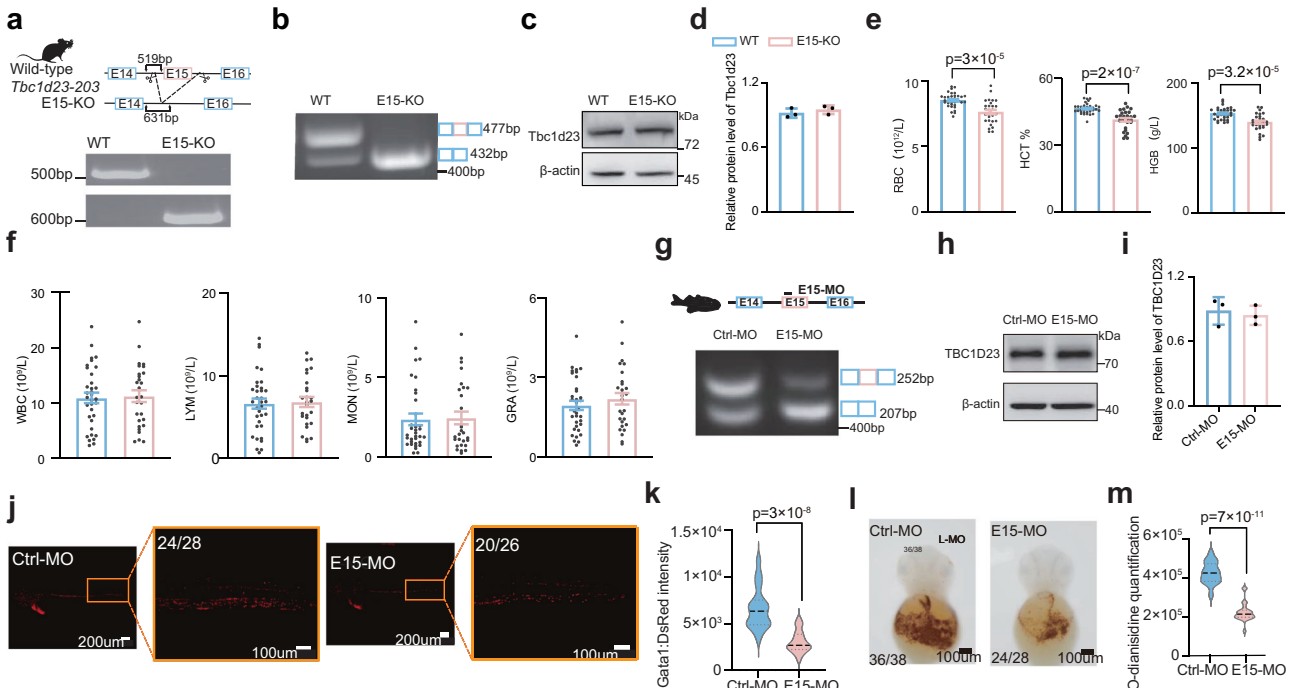

**Fig. 4 | Exon 15 of TBC1D23 regulates erythropoiesis in mice and zebrafish.**
**a** Schematic of the WT and E15-KO alleles, with arrows indicating primer sites and expected PCR product sizes (Top). Genotyping of E15-KO mice (Bottom). **b** RT-PCR validated the specific deletion of the exon 15-containing isoform (*Tbc1d23-L*) in E15-KO mice. **c** Immunoblot of Tbc1d23 total protein level in WT and E15-KO mice bone marrow. **d** Quantification of immunoblots in (**c**) (*n* = 3 independent experiments). **e** Peripheral blood RBC numbers, HCT%, and HGB in E15-KO mice (*n* = 28 mice) in comparison to WT (*n* = 34 mice). **f** Numbers of whole white cells (WBC), lymphocytes (LYM), monocytes (MON), and granulocytes (GRA) in E15-KO mice (*n* = 34 mice) in comparison to WT (*n* = 28 mice). **g** Schematic of morpholino oligonucleotide (MO) targeting Tbc1d23-E15. The horizontal line shows the position of *Tbc1d23-E15* MO (E15-MO) (Top). RT-PCR validated the specific knockdown of *Tbc1d23-L* in control-MO-injected (Ctrl-MO) and E15-MO-injected (E15-MO) zebrafish embryos at 48 hpf (Bottom). **h** Immunoblot of total Tbc1d23 protein levels in zebrafish embryos following Ctrl-MO or E15-MO treatment. **i** Quantification of immunoblots in (**h**) (*n* = 3 independent experiments). **j** Fluorescence images of transgenic zebrafish Tg (Gata1: DsRed) in Ctrl-MO, and E15-MO zebrafish embryos at 48 hpf. **k** Quantification of the Gata1: DsRed intensity at the CHT region of Ctrl-MO and E15-MO (*n* = 20 zebrafish/group). **l** Representative images of O-dianisidine staining for hemoglobin in Ctrl-MO and E15-MO zebrafish embryos at 48 hpf. **m** Quantification of O-dianisidine intensity for embryos injected with Ctrl-MO (*n* = 34 zebrafish) and E15-MO (*n* = 26 zebrafish). For Statistical analysis, the following tests were used: Two-tailed Mann-Whitney U tests (**e**, **f**, **k**, **m**), Unpaired two-tailed Student's t-test (**d**, **i**). Experiments of (**a**, **b**, **g**) were repeated three times, consistently yielding similar results in each iteration. Data in (**d**, **i**) are represented as mean ± s.d and (**e**, **f**) are represented as mean ± s.e.m, and pooled from three independent experiments. Source data are provided as a Source Data file.

accurately identified AS events play important roles in lineage fate commitment.

## Loss of *TBC1D23-E15* inhibits vertebrate erythropoiesis

As shown above, FAScore aids in identification of ES evens that may play important roles in hematopoiesis, we next experimentally analyzed the role of *TBC1D23-E15* with high Human_erythroid_FAScore (0.998) and high PSI (0.78) in erythroid lineage of human (Fig. 3d, e). *TBC1D23*, a member of the TBC family, is known to regulate participate in the regulation of the traffic between endosomes and Golgi apparatus[53,54] and is reported to influence the pathway of the innate immune response[55] (Supplementary Fig. 7a). We characterized the transcription level of Tbc1d23-E15 in the sorted HSPCs and mature cell types in mouse bone marrow using RT-PCR, the result show the PSI correlated well with the PSI derived from bulk RNA-seq analysis of mouse hematopoietic cell populations, confirming that E15 exhibits the most pronounced differential splicing in erythroid cells (Supplementary Figs. 7b, c). By comparing the adjacent regions of *TBC1D23-E15* in 13 species ranging from *C. elegans* to humans, we found that the canonical splice site dinucleotide motifs (AG at the 3' splice site and GT/GC at the 5' splice site) were present in zebrafish but not in amphioxus, suggesting that this exon arose during the evolutionary transition from invertebrates to vertebrates (Supplementary Fig. 7d). Using the human erythroleukemia K562 cell line and TBC1D23 antibody, we conducted a mass spectrometry (IP-MS) experiment and confirmed that this 45-bp exon indeed encodes a peptide of 15 amino acids (Supplementary Fig. 7e). This peptide is proximal to the binding region with FAM91A1[53,56].

To investigate the role of *TBC1D23-E15* in vertebrate hematopoiesis, we generated a transgenic mouse line in which *Tbc1d23-E15* was deleted (Fig. 4a). In contrast to wildtype (WT) mice expressing both long and short isoforms of *Tbc1d23*, the *Tbc1d23*^ΔE15/ΔE15 (E15-KO) mice express only the short isoform (Fig. 4b, and Supplementary Fig. 7f) with the overall Tbc1d23 protein level was comparable to WT mice (Fig. 4c, d). E15-KO mice exhibited significant reductions in red blood cell (RBC) numbers, hematocrit (HCT) and hemoglobin (HGB) levels compared to the WT littermate controls while white blood cell counts remained unchanged (Fig. 4e, f). In addition, to assess the conservation of this phenotype, we targeted the long isoform of *Tbc1d23* in zebrafish (*Tbc1d23-L*, including E15) by morpholino (MO) treatment. The *Tbc1d23-L*-MO (E15-MO) treatment specifically reduced the mRNA level of the long isoform while increasing the expression of the short isoform, with no significant effect on total Tbc1d23 protein level comparable to the fish treated with the control MO (Ctrl-MO) (Fig. 4g–i, Supplementary Fig. 7g). Using transgenic fish expressing DsRed under the control of the *Gata1* promoter, we found a significant (45%) reduction in RBC levels in 48 hpf fish under E15-MO treatment, compared to control (Fig. 4j, k). Consistent with this, O-dianisidine staining revealed a nearly 50% reduction of HGB levels in E15-MO-treated fish (Fig. 4l, m).

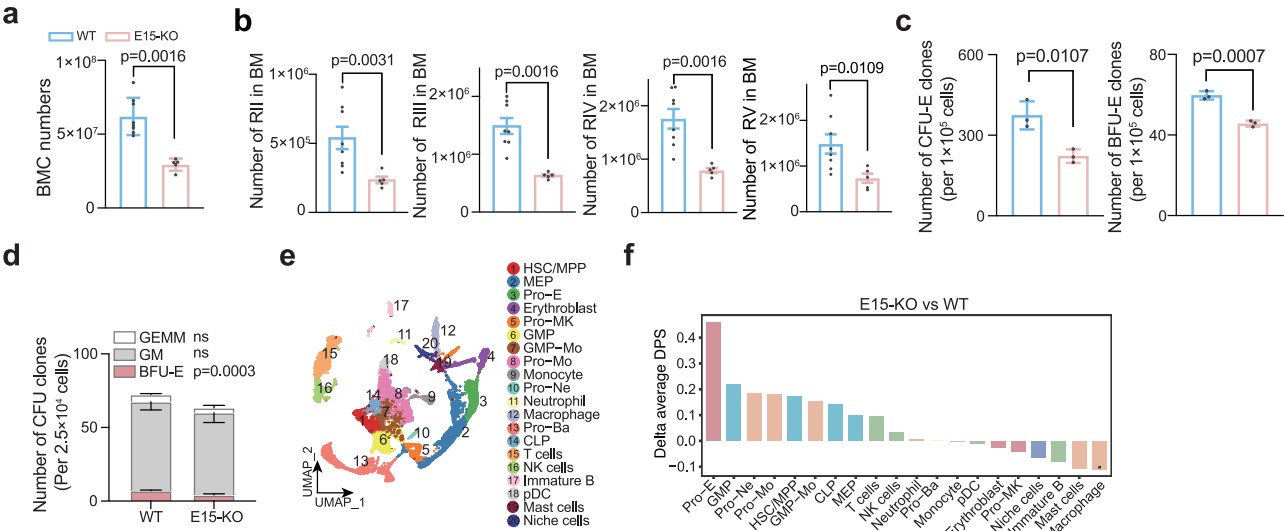

**Fig. 5 | Exon 15 of TBC1D23 regulates erythroid differentiation. a** Number of whole bone marrow cells (BMCs) in E15-KO (*n* = 5 mice) and WT controls (*n* = 8 mice). **b** Number of erythroid progenitor/precursor populations (RII, RIII, RIV, RV) among viable BM cells in E15-KO (*n* = 5 mice) and WT (*n* = 8 mice). **c** Number of colony-forming unit-erythrocyte (CFU-E) (Left) and BFU-E (Right) colonies formed by 100,000 whole BMCs from E15-KO mice and WT controls (*n* = 3 mice / group). **d** Number of GEMM, GM, and BFU-E colonies formed by 25,000 whole BMCs from E15-KO and WT (*n* = 3 mice / group). **e** The UMAP plot of single-cell subpopulation annotation. *HSC/MPP* Hematopoietic stem cell / Multipotent progenitor, *MEP* Megakaryocyte erythrocyte progenitor, *Pro-E* Pro-erythroblast, *Pro-MK* Pro-

megakaryocyte, *GMP* Granulocyte monocyte progenitor, *GMP-Mo* GMP-monocyte, *Pro-Mo* Pro-monocyte, *Pro-Ne* Pro-neutrophil, *Pro-Ba* Pro-basophil, *CLP* Common lymphoid progenitor, *pDC* plasmacytoid dendritic cell. **f** The barplot of the difference of average differentiation potential scores (DPS) between E15-KO and WT for each cell type. The color denotes the specific lineage to which the cell type is associated. For Statistical analysis, the following tests were used: Two-tailed Mann-Whitney U tests (**a**, **b**), Unpaired two-tailed Student's t-test (**c**, **d**). Data in (**a**, **b**) are represented as mean ± s.e.m and are representative of three independent experiments. Data are shown as mean ± s.d (**c**, **d**) and are representative of three independent experiments. Source data are provided as a Source Data file.

We next determined whether *Tbc1d23-E15* deletion leads to a defect in erythroid differentiation in BM which results in anemia in mice. Firstly, we found that the number of bone marrow cells (BMCs) in the E15-KO mice was 52% lower than in WT mice (Fig. 5a). Moreover, by characterizing the stages of erythroid cells (referred to here as RI, RII, RIII, RIV and RV) by flow cytometry utilizing TER119 and CD44, we observed the frequency of erythroid cells (RII, RIII, RIV and RV) in BM was also halved compared to WT mice (Fig. 5b). In contrast, the proportion of HSCs and progenitors in the BM of E15-KO mice remained unchanged (Supplementary Fig. 7h). Colony formation assay showed a 23% reduction in BFU-E colony number and a 40% reduction in colony-forming unit-erythroid (CFU-E) clones for E15-KO mice (Fig. 5c, d).

To further investigate how *Tbc1d23-E15* affects erythropoiesis, we performed a single-cell RNA sequencing (scRNA-seq) analysis on BMCs of E15-KO and WT mice (Fig. 5e, and Supplementary Fig. 8a). In total, we identified 802 differentially expressed genes (DEGs) in all cell types, with the greatest number of DEGs observed in the pro-erythroblasts (199) (Supplementary Fig. 8b, c). To identify the lineage with the most transcriptional changes between E15-KO and WT, we used Augur[57] to calculate AUC values. Our analysis revealed that erythroid cells displayed a higher AUC value than other lineages, indicating a strong response to E15-KO (Supplementary Fig. 8d). Utilizing CytoTRACE[58], we assessed the differentiation potential scores of all cell types and found that pro-erythroblasts received the highest score (0.46), compared to 0.10 of MEP (Fig. 5f), in line with our observations that deletion of *Tbc1d23-E15* affects the maturation of erythroid cells but not progenitors.

### TBC1D23-E15 modulates erythropoiesis by regulating the SUMOylation of HDAC1

To investigate how TBC1D23-E15 regulates erythropoiesis, we analyzed the interactomes of long and short isoforms of TBC1D23 by an integrated multi-omics approach. We generated three in both K562 and HEK293T, one lacking TBC1D23 expression (KO) and the other two

expressing only TBC1D23-L or TBC1D23-S (Supplementary Fig. 9a, and Fig. 6a, b), and performed IP-MS in the K562 cell lines (Fig. 6c, Supplementary Data 6). RanBP2 was identified as one of the candidate proteins showing differential binding toward TBC1D23-S or TBC1D23-L (Supplementary Fig. 9b, c). RanBP2 is a known Sumo E3 ligase that forms a complex with SUMO1-modified RanGAP1[59]. The immunofluorescence analysis showed that *TBC1D23* co-localized with *RanBP2* (Fig. 6d). Using the TBC1D23-KO, TBC1D23-L and TBC1D23-S KEK293T cell lines overexpressing RanBP2 or RanGAP1 E3 SUMO ligase domain, we confirmed TBC1D23-S consistently precipitated more RanBP2 and RanGAP1 than TBC1D23-L by 5 and 4 times, respectively (Fig. 6e, f), suggesting that E15 exclusion enhances engagement with the complex (Fig. 6e, f).

To investigate whether the SUMOylation level was differentially regulated by the long and short isoforms of TBC1D23, we performed quantitative SUMO proteomics using K562 cells expressing only TBC1D23-S or -L (Fig. 7a, and Supplementary Data 7) and TBC1D23-KO cells were also included as a negative control (Supplementary Data 7). The identified substrate proteins were enriched in various biological processes, with 'regulation of hemopoiesis' ranked as one of the top 10 pathways (p = 5.2e-04) (Fig. 7b). Eleven proteins within this pathway showed differential SUMOylation in TBC1D23-S and -L expressing cells, and HDAC1, TFE3 and HMGB1/HMGB2 had higher SUMOylation levels in cells expressing TBC1D23-S compared to those expressing TBC1D23-L (Fig. 7c). To elucidate the potential key regulators, we further performed consensus co-expression network analysis on the scRNA-seq dataset using hdWGCNA[60] (Supplementary Fig. 10a). The most significantly changed network out of the seven identified co-expression modules (M) were M1 and M5 (Supplementary Fig. 10b, c). M1 was associated with erythrocyte differentiation and protein modification, while M5 was associated with immune response (Supplementary Fig. 10d). Notably, HDAC1 (histone deacetylase 1) and PRMT1 were present both in M1 and the differential SUMOylation dataset related to the regulation of hemopoiesis (Supplementary Fig. 10e).

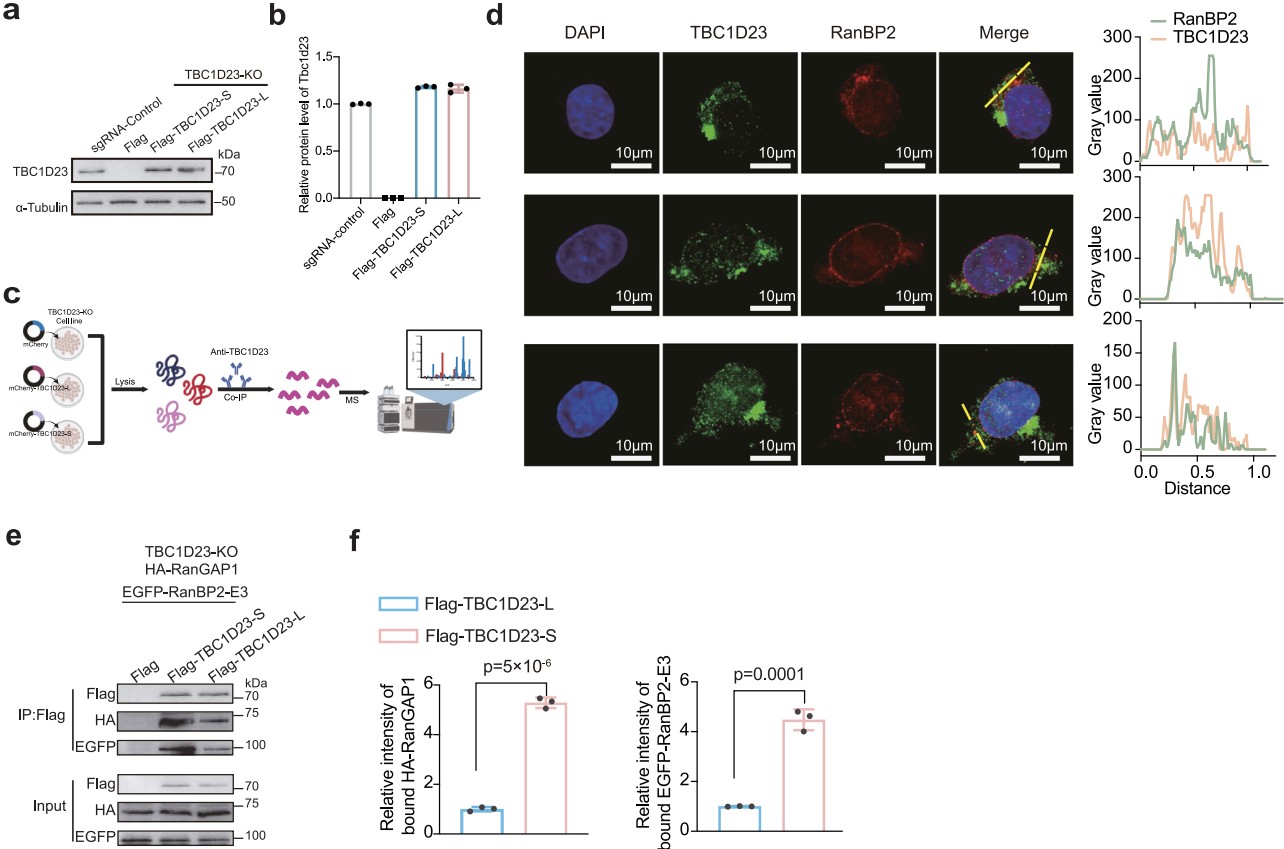

**Fig. 6 | TBC1D23-E15 leads to differential binding capacity to the RanBP2/RanGAP1 complex. a** Western blot analysis of TBC1D23 protein expression levels in HEK293T cells under four conditions: sgRNA-control cells, TBC1D23-KO cells reconstituted with either empty Flag vector (Flag), Flag-tagged short isoform of TBC1D23 (Flag-TBC1D23-S), or Flag-tagged long isoform of TBC1D23 (Flag-TBC1D23-L). **b** Quantification of TBC1D23 expression levels from (**a**) ($n = 3$ independent experiments). **c** Scheme of the IP-MS study design. TBC1D23-KO K562 cells were reconstituted with mCherry vector (mCherry), mCherry-TBC1D23-L, or mCherry-TBC1D23-S, followed by IP assays with an anti-TBC1D23 antibody. Created in BioRender. (https://BioRender.com/optv69y). **d** IF image of HEK293T cells stained with antibodies to DAPI (blue), TBC1D23 (green), and RanBP2 (red) (Left). Fluorescence intensity profiles along the dashed line are shown for RanBP2 (green) and TBC1D23 (orange) (Right). **e** Co-immunoprecipitation (co-IP) analysis of the binding capacity of TBC1D23-S and TBC1D23-L to the RanBP2 complex. HA-RanGAP1 and EGFP-RanBP2-E3 were co-transfected into TBC1D23-KO HEK293T cells reconstituted with Flag, Flag-TBC1D23-S, or Flag-TBC1D23-L. Cell lysates were subjected to immunoprecipitation (IP) with anti-Flag antibody and immunoblot with anti-Flag, anti-HA, and anti-EGFP antibodies. **f** Quantification of HA-RanGAP1 and EGFP-RanBP2-E3 band intensities in IP samples from (**e**) ($n = 3$ independent experiments). For Statistical analysis, the following tests were used: Unpaired two-tailed Student's t-test (**d**, **f**). Data in (**d**, **f**) are shown as mean ± s.d and pooled from three independent experiments. Experiments of (**c**, **e**) were repeated three times, consistently yielding similar results in each iteration. Source data are provided as a Source Data file.

Consistent with the tighter binding between TBC1D23-S and the RanBP2/RanGAP1 complex, the SUMO1-mediated SUMOylation level of HDAC1 was 5 times higher in cells expressing TBC1D23-S than those of TBC1D23-L. Importantly, TBC1D23-S and -L displayed similar binding toward *HDAC1* (Fig. 7d, e, Supplementary Fig. 10f). To confirm that HDAC1 is a substrate of RanBP2, we expressed HDAC1 with RanBP2[wt] or a catalytic-dead mutant, RanBP2[mut] (F2735A/F2736A/C2737A), and found that HDAC1 was drastically SUMOylated only in RanBP2[wt]-expressing cells (Supplementary Fig. 10g, and Fig. 7f). Lys476 of HDAC1 was identified in our assay with higher SUMOylated levels in cells expressing TBC1D23-S. Lys476 was reported as a major SUMOylation site of HDAC1, together with Lys444[61]. To determine whether the two sites were SUMOylated by the RanBP2/RanGAP1 complex, we generated cell lines expressing FLAG-tagged HDAC1[wt] (K444/K476) or HDAC1[mut] (R444/R476) and found that RanBP2-dependent SUMO1-mediated SUMOylation can only be detected in HDAC1[wt] expressing cells (Fig. 7f). Furthermore, HDAC1 SUMOylation was elevated by 1.5-fold in BMCs from E15-KO mice compared to those of WT controls (Fig. 7g, h). Importantly, the reduced CFU-E colony numbers of E15-KO BMCs could only be rescued by the expression of HDAC1[mut], but not by that of HDAC1[wt] (Fig. 7i). Thus, *TBC1D23-E15* regulates erythropoiesis likely by modulating the SUMOylation of HDAC1.

To reconstruct the gene regulatory networks (GRN) affected by *Tbc1d23-E15*, we analyzed the activity of transcription factors (TFs) in erythroid scRNA-seq data of E15-KO and WT BMCs by SCENIC[62] (Supplementary Data 8). This analysis yielded 204 regulons out of 1,721 initial co-expression modules, with significantly enriched motifs for the corresponding TFs in erythropoiesis. Twelve TFs were significantly overlapped between these TF regulons and Hdac1-related TFs of erythropoiesis[63] ($p = 0.003$, Supplementary Fig. 10h, and Supplementary Data 9). For example, the downregulation of *Myc* is essential for terminal erythroid maturation[64], whose activity increases in E15-KO and is associated with 2,774 predicted target genes. In contrast, *Klf1*, a master erythroid gene regulator[65], shows decreased activity in E15-KO, with 1,198 predicted target genes (Supplementary Fig. 10i, and Supplementary Data 9). Furthermore, we found five correlation networks amongst different TFs by hierarchical clustering, in which TFs targeted by Hdac1 mainly belonged to network1 (N1) (*Spi1, Gata2, Myc, Stat5a, Rcor1*) and network2 (N2) (*Tal1, Klf11, Klf1, Gata1, Sp1*) (Supplementary Fig. 10j). Additionally, we obtained a co-regulation network conducted by the 10 TFs and their target genes to affect erythroid differentiation (Supplementary Fig. 10k). These results indicate that skipping E15 of Tbc1d23 leads to SUMO-modification of Hdac1, which may modulate the deacetylation and activation of the target TFs, impairing erythroid cell

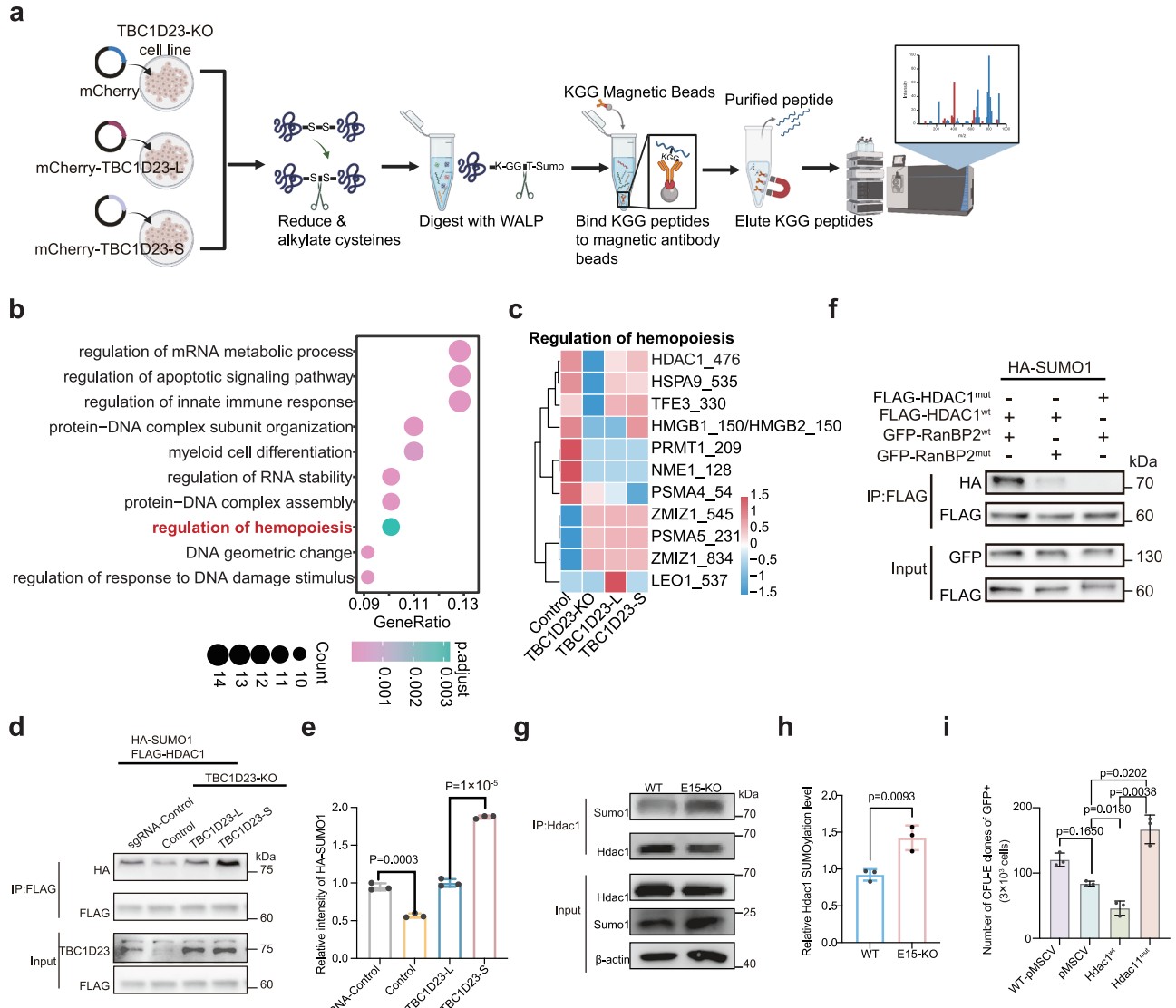

**Fig. 7 | TBC1D23-E15 modulates erythropoiesis by regulating the SUMOylation of HDAC1. a** Experimental workflow for MS analysis of SUMOylation peptides. TBC1D23-KO K562 cells were reconstituted with mCherry, mCherry-TBCD23-L and mCherry-TBC1D23-S. After lysis and digestion, peptides were purified and SUMOylated peptides were enriched with motif-specific (K-ε-GG) magnetic beads for LC-MS/MS. Created in BioRender. (https://BioRender.com/8p9xt4c). **b** Bubble plot showed the top 10 GO pathways enriched for down-regulated SUMO-modified proteins in KO samples compared to control, TBC1D23-S and TBC1D23-L (log$_2$FC ≤ -0.25) ranked by term count. Bubble size indicates gene count; color indicates enrichment significance (one-sided well-adopted hypergeometric test and corrected by Benjamini-Hochberg). **c** Heatmap showing scaled abundance of SUMO-modified proteins in the hematopoiesis regulation pathway. Row labels indicate protein SUMO-modified sites. **d** Co-IP in HEK293T cells co-transfected with Flag-HDAC1 and HA-SUMO1 under four conditions: sgRNA-control, or TBC1D23-KO cells reconstituted with empty vector (control), TBC1D23-S, or TBC1D23-L. Lysates were subjected to IP with anti-Flag and immunoblot with anti-HA, anti-Flag. Input lysates were immunoblotted with anti-TB1CD23 and anti-Flag. **e** Quantification of HA-

SUMO1 band intensity from (**d**) ($n$ = 3 independent experiments). **f** Co-IP in HEK293T cells stably expressing HA-SUMO1 and co-transfected with Flag-HDAC1$^{WT}$ / Flag-HDAC1$^{mut}$ (K444R/K476R) and GFP-RanBP2$^{WT}$/GFP-RanBP2$^{mut}$ (F2735A/F2736A/C2737A). Crosslinking immunoprecipitations with anti-Flag. Immunoblotting analysis of HA-SUMO1 modified HDAC1 by the indicated antibody after washing away impurities with 500 mM KCl buffer. **g** Co-IP analysis of the HDAC1 SUMOylation level in WT and E15-KO mice. Bone marrow cells (1 × 10$^8$) were lysed and subjected to IP with anti-HDAC1 and immunoblot with anti-SUMO1. **h** Quantification of HDAC1 SUMOylation level from (**g**) ($n$ = 3 independent experiments). **i** CFU-E colony counts from sorted 3000 WT CD117$^+$ cells transduced by retroviruses with empty vector (WT-pMSCV) and E15-KO CD117$^+$ cells transduced by retroviruses with HDAC1$^{wt}$ or HDAC1$^{mut}$ or pMSCV (n = 3 mice/group). For Statistical analysis, the following test was used: Unpaired two-tailed Student's t-test (**e, h, i**). Data are shown as mean ± s.d and pooled from three independent experiments. Experiments of (**d, f, g**) were repeated three times, consistently yielding similar results in each iteration. Source data are provided as a Source Data file.

differentiation. This suggests the significant role of this AS event in regulating transcriptional networks during erythroid lineage commitment.

## Discussion

Functional alternative transcripts are typically identified by testing differential exon or isoform usage in regions that affect protein domains or structures[26,66,67]. However, thousands of differentially used exons or isoforms exist, and most AS events do not result in alterations to annotated motifs or domains. With the growing availability of cross-species RNA-seq datasets encompassing various developmental stages and cell lineages, we can now explore the dynamic patterns of genes, lincRNAs, and alternative splicing isoforms that are key for tissue development and cell differentiation. Despite this wealth of data, an

optimized computational algorithm that integrates these diverse datasets to identify functional AS events remains lacking. Here, we analyzed 271 RNA-seq datasets across multiple species, developmental stages, and lineage differentiation, incorporating both static and dynamic features to develop a predictive model for functionally significant AS events in hematopoiesis.

Conventional supervised learning methods require well-defined positive and negative sets, which is challenging for functional AS prediction: experimentally validated isoforms (positives) are limited, and instances of non-functional AS are virtually undocumented, making a reliable negative set difficult to obtain. To address this, we employed the PU-learning framework, inferring potential positive and negative AS from unlabelled AS, to improve classifier performance. We further used the classical supervised model, Random Forest, for scoring, classification, and feature importance analysis due to its high prediction accuracy and model interpretability. Our model integrates multiple features into a unified predictive score, and can automatically quantify the contribution of different features to the prediction, avoiding the subjective biases in features-based filtering approaches. Finally, compared to traditional selecting methods that yield a large set of candidate AS events without prioritization, FAScore provides a continuous prediction score. This enables ranking candidates by their relative score, thereby offering a quantitative framework for prioritizing targets and increasing the efficiency of downstream functional validation.

Rapid evolutionary change in AS patterns has been proposed to be driven by the gain and loss of functional units such as protein domains, transmembrane helices, signal peptides, or coiled-coil regions[68]. Furthermore, AS tends to insert/delete complete rather than partial functional modules[69]. However, there is a large percentage of AS events that do not have a clear predicted functional unit such as domain or protein structure. Our model complemented the known functional prediction with the dynamically regulated of AS that identifies developmental- and/or lineage-regulated AS. Employing the FAScore, we successfully identified the specific functions of *KLF6-E3* and *SSBP3-E6* during myeloid differentiation, as well as confirmed the unique roles of *EPB41l1-E14* and *TBC1D23-E15* during erythroid differentiation. Our prediction enables prioritizing strong candidate AS for functional validation. We further unveiled the molecular mechanism of *TBC1D23* isoforms in regulating erythropoiesis, determining the SUMOylation levels of multiple proteins through changes in binding capacity to RanBP2. This event may have evolved to modulate the timing of transitions in the production of erythroid progenitors and mature red blood cell to affect erythropoiesis and linage commitment. Supporting this notion, we observe that the affected target proteins are associated with the regulation of hemopoiesis functions and that their SUMOylation levels are significantly affected by exon 15 skipping levels during erythroid differentiation. We further illustrated that this exon-skipping event enhanced the SUMOylation of HDAC1 to influence its downstream transcription factors, altering the expression pattern of thousands of genes and impairing erythropoiesis. Thus, a single AS event can elicit a far-reaching cascade of biological effects, altering the lineage commitment landscape. In sum, our study provides a paradigm to characterize functional AS using a machine-learning model and investigate the downstream regulation through experiments and scRNA-seq. Our work improves understanding of the regulation mechanisms during the lineage commitment. Moreover, this paradigm can be extended to other spatially and temporally dynamic biological processes, such as organ development, cell differentiation, aging and disease progression.

## Methods

All research presented in this manuscript complies were carried out following the protocols approved by the ethics committee of West China Second University Hospital.

## FAScore algorithm overview

The framework of the FAScore consists of several steps: (1) calculation of the dynamic features of genes or alternative splicing events; (2) extraction of structural features of isoforms from the APPRIS database; (3) Training of predictive model; and (4) calculation of the probability to generate the functional AS scores (FAScore). We implemented and tested FAScore in R (v4.0) and designed it for an R package. FAScore code is open source and available on GitHub (https://github.com/LuChenLab/FAScore.git), along with detailed descriptions of functions and tutorials.

### Step1: Calculation of the dynamic features

**Data collection.** To train the functional AS predictor and study the evolution of lineage commitment and developmental AS, we leveraged some bulk RNA-seq datasets comprising 1,985 libraries spanning the hematopoietic cell differentiation (87) and organ development (1,898). The hematopoietic differentiation datasets include 17 cell types belonging to three lineages extracted from the bone marrow (mouse)[35] and umbilical cord blood (human)[17] (Supplementary Data 1). And the organ development datasets were downloaded from Henrik Kaessmann et al.[32] contain seven organs (forebrain/cerebrum, hindbrain/cerebellum, heart, kidney, liver, ovary and testis) from early organogenesis (mid-organogenesis for the heart) to adulthood across six mammals and chicken. In addition, we also exacted the dataset of the liver in the embryonic period across five mammals (human, rhesus, mouse, rat and rabbit) from the above datasets, and added the dataset of the caudal hematopoietic tissue (CHT) which was the embryonic hematopoietic organ in zebrafish from 36 hpf to 4 dpf stage[37]. Fetal liver and CHT are collectively referred to as the fetal hematopoietic organ (FHO) in the remainder of the article (Supplementary Data 1).

**RNA-seq data preprocessing.** All RNA sequencing libraries were processed using standard pipelines. Briefly, the quality of the raw sequence data was checked using FastQC software (v0.11.8). Low-quality reads were trimmed using Trimmomatic v0.38[70] with parameters of 'ILLUMINACLIP: adapters.fa:2:30:10 LEADING:3 TRAILING:3 and SLIDINGWINDOW:4:15 MINLEN:25'. Trimmed reads were aligned to the respective reference genomes (Supplementary Data 10) using STAR v2.6.1a[71] with default parameters, supplemented by '-quantMode' of TranscriptomeSAM, with the resulting BAM files, input into RSEM v1.3.1[72] to quantify the gene expression and isoform abundance as raw counts and transcripts per million (TPM) (Supplementary Fig. 11a–c). The strand specificity of each library was passed to the '-strandedness' option of RSEM. The AS events were identified and quantified based on the exonic-centric method using the reported pipeline from MeDAS[38] which identifies the exonic parts as AS events (Codes are available at https://github.com/LuChenLab/MeDAS). Constitutive exons exhibit a PSI of 1, while alternatively spliced exons range from 0 < PSI < 1. Based on their PSI values across all samples, exonic parts were categorized into: 'not expressed' (no PSI data available), 'low inclusion' (PSI < 0.05), 'alternatively spliced' (0.05 ≤ PSI ≤ 0.95), or 'constitutive' (PSI > 0.95). For downstream analysis, AS events were selected based on two criteria: (1) the exonic part must be classified as 'alternatively spliced' in at least two samples, and (2) the exonic part length must be ≥ 3 bp.

Besides, AS types were inferred using SUPPA2 with 'local AS events' mode[39], including ES, retained intron (RI), alternative 5' splice site (A5), alternative 3' splice site (A3), mutually exclusive exons (MX), alternative first exon (AF, resulting in mRNA isoforms with distinct 5' UTRs), alternative last exon (AL). NA indicated novel AS events without annotated. For downstream analysis, we selected AS events whose annotated AS type included "ES" or those that were unannotated (novel).

### Identification of dynamically expressed genes and AS events

**Feature extraction.** We extracted 6 features from the expression matrices, including the expression breadth (the range value, *R*), the

expression relationship with differentiation or development (the Spearman's rank coefficient, $\rho$; the Spearman's test p-value, $p_\rho$), the expression specificity (the specificity index, $\tau$) and the expression gradient (the linear regression coefficient, $\beta$; the p-value of the regression variable, $p_\beta$).

(1) **Expression breadth:** range value ($R$)

The value range ($R$) represents the difference between the maximum and minimum expression values, reflecting the extent of variation. For genes, expression breadth ($R_{gene}$) was calculated as follows: a pseudo-count of 1 was added to the TPM values, which were then log2-transformed; $R_{gene}$ was then computed as the difference (max − min) among the transformed values for each gene (Supplementary Fig. 11d). when the expression breath is 0, it means no difference. Meanwhile, when expression breath is 1 or greater than 1, it means a minimum 2-fold expression difference ($\log_2(FC) = 1$) within a gene. Thus, we applied a transformation step wherein any $R_{gene}$ value $\geq 1$ was set to 1 (Supplementary Fig. 11e). For AS events quantified using PSI, the range ($R_{AS}$) was directly computed as the difference in PSI values for each AS event. Since PSI is naturally bounded between 0 and 1, $R_{AS}$ values also inherently fall within the 0–1 range (Supplementary Fig. 11f).

(2) **Expression relationship** with differentiation or development: Spearman's rank coefficient ($\rho$) and Spearman's test p-value ($p_\rho$).

The correlation of expression profile of development stages or cell types was calculated as genes or AS during organ development or hematopoietic stem cell differentiation. The developmental stages of organs or cell types along the lineage commitment were defined as 'time-points' starting from 1. To measure the relationship between the development or differentiation time-points and gene expression levels or AS PSI levels, we calculated two features, Spearman's rank coefficient ($\rho$) and Spearman's test p-value ($P_\rho$, two-sided t-test)[73]. The rank coefficient ranges from −1 to 1, with the positive values indicating a direct correlation and the negative values indicating an inverse correlation. The p-values were transformed into binary form. The values below 0.05 were labeled as significant (1) and the rest as not significant (0).

(3) **Expression gradient:** regression coefficient ($\beta$) and p-value of the regression variable ($P_\beta$)

To objectively characterize the linearity of relationships, we employed generalized additive modeling (GAM) and effective degrees of freedom (edf), which quantifies the complexity or "wiggliness" of a smooth term, reflecting the extent to which the model deviates from a straight line. The observation that edf values clustered near 1[74] indicates that gene expression and splicing levels exhibit predominantly linear tendencies (51.8%) (Supplementary Fig. 12a-c). Accordingly, we primarily conducted linear regression analyses, where the regression coefficient ($\beta$) represents the magnitude of influence exerted by the independent variable on the dependent variable, and the p-value of the regression variable ($P_\beta$) reflects the statistical significance of this association.

To mitigate potential skewness in gene expression distributions while preserving biological interpretability, we applied a $\log_2(TPM + 1)$ transformation to gene expression values, which were then subjected to linear regression against normalized differentiation cell types or developmental time points. Although this transformation reduced bias from highly expressed genes, the resulting slopes initially ranged from −2 to 2, with most values concentrated near 0. These initial slopes showed only weak correlation with slopes derived from raw TPM values (Pearson $r = 0.17$, Supplementary Fig. 12d). We therefore applied capping to extreme values at the range [−1, 1], which affected fewer than 1% of genes. This adjustment substantially improved the correlation to $r = 0.84$ (Supplementary Fig. 12e),

suggesting enhanced biological relevance. The optimized slope values also support direct additive integration across different features, providing a mathematically consistent basis for downstream dynamic score calculation. For AS events, PSI values inherently yield slopes within [−1, 1] without additional processing (Supplementary Fig. 12f), ensuring analytical consistency with processed gene expression metrics. Furthermore, p-values were binarized: values below 0.05 were assigned a significance label of 1, and all others were labeled 0.

(4) **Expression specificity:** specificity index ($\tau$)

The specificity index ($\tau$) was calculated to measure the expression specificity of gene/AS in organ development or lineage commitment[75].

$$\tau = \frac{\sum_{i=1}^{N}(1 - x_i)}{N - 1},\tag{1}$$

where $N$ is the number of developmental stages or lineage commitment cell types, and $x_i$ is the gene expression or AS PSI normalized against the highest expression or PSI value in each gene or AS. $\tau$ interpolate the entire range between 0 for general and 1 for strict expression specific.

### Definition of dynamically regulated genes and AS

The dynamic expression score (DyScore) was calculated as the arithmetic average of the six features described above. The DyScore is defined as:

$$DyScore = \frac{R + |\rho| + P_\rho + |\beta| + P_\beta + \tau}{n},\tag{2}$$

where $R, P_\rho, P_\beta$ and $\tau$ represent the scaled range values, the Spearman's test p-value, the p-value of the regression variable, and the specificity index, respectively. The $|\rho|$ indicates the absolute value of the Spearman's rank coefficient, and $|\beta|$ is the absolute value of the regression coefficient. And $n$ is the number of features. The DyScore value is between −1 to 1. The symbol indicates the direction of change, consistent with correlation.

Based on bimodal distribution of absolute dynamic score, we applied Gaussian Mixture Modeling (GMM) to classify the genes or AS events into "dynamic" and "non-dynamic". A GMM with $k = 2$ components effectively captured the bimodal distribution. The optimal threshold separating the two components was determined by probability density intersection method, resulting in a cut-off of 0.467 for genes and 0.324 for AS events. We then calculated the arithmetic mean of the gene-level and AS-level cutoffs, and it was 0.40. Finally, genes or AS events were divided into two classes, including dynamically expressed genes/AS events ($\geq 0.4$ or $\leq -0.4$), and non-dynamically expressed genes/AS events (-0.4 - 0.4).

### Step2: Extraction of structural features

Structural characteristics of isoforms were derived from APPRIS database[76] (https://appris.bioinfo.cnio.es), including 6 features:

(1) the structural homologs and integrity score. The presence of structural homologs in the Protein Data Bank (PDB) was evaluated and the integrity of the 3D structure was tested using Matador3D[77].

(2) functionally important residues score. Conserved and functionally important amino acid residues were predicted by firestar[78].

(3) domain integrity score. Domain integrity score was counted using the SPADE Pfamscan program[79].

(4) trans-membrane helices score. Transmembrane helices were predicted by three separate trans-membrane predictors of THUMP, including MEMSAT3[80], Phobius[81] and PRODIV-TMHMM[82] methods.

(5)  signal peptide score. Signal peptides were predicted using SignalP.

(6)  target peptide score. The presence of N-terminal presequences, including signal peptide (SP), mitochondrial transit peptide (mTP), chloroplast transit peptide (cTP) or thylakoid luminal transit peptide (luTP), were predicted using TargetP[83].

The above features of annotations for five Ensembl species were downloaded, including human (GRCh38), rhesus (Mmul10), mouse (GRCm38), rat (Rnor6.0) and zebrafish (GRCz11).

**Step3: Training of predictive model**

**Benchmark dataset.** AS events with actual evidence of functionality and phenotype were searched in PubMed[84] using the terms of 'alternative splicing' with 'brain' OR 'heart' OR 'liver' OR 'kidney' OR 'ovary' OR 'testis' OR 'hematopoiesis' OR 'myeloid' OR 'lymphoid' OR 'erythroid'. A total of 37 uniquely exon-skipping events were experimentally verified as functional in organ development or hematopoietic differentiation, among 2 AS events observed in two species and 15 observed in multiple lineages or organs (Supplementary Data 3), adding up to 58 positive instances. These were used to train model totally. The benchmark dataset can be defined as:

$$S = S^{IP} \cup S^U \qquad (3)$$

where $S^{IP}$ represents the initial positive set containing above 58 functional AS events, and $S^U$ is the unlabeled set, containing 9,997 randomly selected ES AS events of unknown function. The $S^U$ was assembled by randomly selecting 400 AS events from each dataset, containing seven organs and three hematopoietic lineages of both human and mouse, while excluding the subset with 'NA' features.

**Prediction workflow.** The Random Forest (RF)-based algorithm was selected as the final model due to its solid performance in binary classification tasks, compared to Naive Bayes (AUC: 0.94), Logistic Regression (0.96), Support Vector Machine (SVM, 0.98), AdaBoost (0.88) and CART (0.90). Known for their robust performance across different datasets, RF algorithms can handle various feature types—categorical, Boolean, and continuous—without needing extensive feature selection. They effectively manage correlated features and require fewer hyperparameters to avoid overfitting. Moreover, RF models can reliably predict even with missing data and are adaptable to datasets lacking complete predictive feature coverage, making them applicable across different species and annotation sets. While non-linear models like RF used to be seen as less interpretable, recent advancements in understanding tree-based models have improved by explaining both the global impact of features and their influence on individual predictions.

The framework of the prediction model contained three steps: (i) the initial features were generated for the functional AS based on the dynamic, conservative and structural information; (ii) The unlabeled set was labeled by an ensemble PU learning process, including positive and negative labels. All labeled samples were divided into training and test sets; (iii) The Random Forest classifiers were trained based on training dataset. Best prediction model was selected with highest performance from 10-fold cross-validation.

**Predictive features.** The 19 features of the predictive model can be classified into three main categories:

a.  Dynamic features of genes (six features) or AS (six features). This category depicts expression characteristics in lineage differentiation or organ development, including the expression breadth (the range value, $R$), the expression relationship with differentiation or development (the Spearman's rank coefficient, $\rho$; the Spearman's test p-value, $P_\rho$), the expression specificity (the specificity index, $\tau$) and the expression gradient (the linear regression coefficient, $\beta$; the p-value of the regression variable, $P_\beta$).

b.  Evolutionary Feature. This category captures cross-species sequence conservation, assessed by mapping vertebrate orthologs to each variant and counting the number of orthologs that are correctly aligned without gaps using CORSAIR[85].

c.  Structural Feature. Structural characteristics of isoforms were derived from APPRIS database[76], including six features which are the structural homologs and integrity score, functionally important residues score, domain integrity score, trans-membrane helices score, signal peptide and subcellular location scores.

**Positive unlabeled learning based on random forests.** To alleviate the false negative problem of the unknown functional AS and to fully utilize the limited validated functional AS, positive unlabeled learning (PU learning)[86] was adopted in this study and combined with random forests (PU-RF) to determine the potential functionality of AS. Moreover, cross-validation is a key approach to evaluate the performance of predictors. We employed the 10-fold cross-validation to evaluate the performance of PU learning result. The $S^P$ were randomly divided into 10 folds, with one-fold defined as the validation set $S^{val}$ and the remaining nine folds used as the initial positive set $S^{IP}$.

In step 1, the 'initial negative' training subset was randomly selected from the unlabeled subset $S^U$ with the same number of positive subsets $S^{IP}$. Numerous learning processes were implemented based on $T$ ($T$=100) different initial training datasets. For each base learning process, the initial training dataset was composed of two subsets:

$$S_i^{train} = S^{IP} \cup S_i^{IN} \quad i = 1, 2, ..., T \qquad (4)$$

$$\text{with } |S^{IP}| = |S_i^{IN}| \quad i = 1, 2, ..., T \qquad (5)$$

where $S^{IP}$ is the initial positive training subset, and $S_i^{IN}$ is the initial negative training subset. The numbers of AS in $S^{IP}$ and $S_i^{IN}$ are the same. Because there is no validated non-functional AS, $S_i^{IN}$ is composed of the randomly selected AS from the 'unknown' subset $S^U$.

The initial training dataset was used to train the classifier based on the support vector machine (SVM). And the rest of the unlabeled subset were identified with probability scores by the classifier, the subsets with scores greater than 0.8 were considered as 'reliably positive' $S^{RP}$ and less than 0.2 were considered as 'reliably negative' $S^{RN}$.

In step 2, the 'initial and reliably positive' subset and the 'initial and reliably negative' subset above step 1 were used to train the SVM classifier again. The classifier was used for the rest of the unlabeled dataset. The reliably positive and negative samples were selected by the probability scores greater than 0.9 and less than 0.1. The step was iterated until the number of remaining unlabeled datasets ($N$) reached zero or no new reliable samples were generated.

$$S_{j+1}^P = S_j^P + S_j^{RP} \quad j = 0, 1, 2, ..., L \qquad (6)$$

$$\text{when } j = 0, \; S_1^P = S^{IP} \qquad (7)$$

$$S_{j+1}^N = S_j^N + S_j^{RN} \quad j = 0, 1, 2, ..., L \qquad (8)$$

$$\text{when } j = 0, \; S_1^N = S^{IN} \qquad (9)$$

$$S_{ij}^{train} = S_{ij}^P \cup S_{ij}^N \quad i = 1, 2, ..., T \; j = 0, 1, 2, ..., L \qquad (10)$$

where $S_j^P$ represents the total positive subset in the $j^{th}$ iteration in step 2, and $S_j^{RP}$ represents the reliably positive subset in the $j^{th}$ iteration in step 1. Similarly, $S_j^N$ represents the negative subset in the $j^{th}$ iteration in step 2, and $S_j^{RN}$ represents the negative positive subset in the $j^{th}$ iteration in step 1. The training set $S_{ij}^{train}$ in the $j^{th}$ iteration based on the $i^{th}$ initial training datasets were composed of two subsets: $S_{ij}^P$ and $S_{ij}^N$.

Finally, classifiers with a recall of less than 0.85 on the validation set $S^{val}$ were filtered out from the set of $T$ classifiers. The unlabeled AS $S^U$ were definitively classified based on the average score from the retained classifiers. The final labeled dataset was split into training and test sets in an 8:2 ratio. A RF model was trained on the training set with mtry = 7 and ntree = 1000, and its performance was evaluated on the test set.

$$S^{Label} = \sum S_{iL}^{train} / T \qquad (11)$$

$$S^{Label} = S_{train} + S_{test} \qquad (12)$$

## Performance evaluation of the model

The area under the receiving operator characteristic curve (AUROC)[87], the associated precision-recall curves (AUPRC)[88,89] and F1 score were used to evaluate the prediction quality of FAScore.

$$Sensitivity = \frac{TP}{TP + FN} \qquad (13)$$

$$Specificity = \frac{TN}{TN + FP} \qquad (14)$$

$$Precision = \frac{TP}{TP + FP} \qquad (15)$$

$$Recall = \frac{TP}{TP + FN} \qquad (16)$$

$$AUROC = \int_0^1 TPR(t)d(FPR(t)) \qquad (17)$$

$$AUPRC = \int_0^1 Precision(t)d(Recall(t)) \qquad (18)$$

$$F1 = \frac{2 \times Precision \times Recall}{Precision + Recall} \qquad (19)$$

where TP, FP, TN and FN represent true positives, false positives, true negatives and false negatives, respectively.

To evaluate the ability of FAScore to identify functional AS, all AS in the test set were ranked in descending order according to the average scores. The rank index[90–92] was used to measure the average rank of the functional AS in all the AS of test set. The rank index is defined as:

$$rank\ index = \frac{1}{|S_{test}^P|} \sum_{b \in S_{test}^P} \frac{r_b}{|S_{test}|} \qquad (20)$$

where $|S_{test}|$ is the number of all AS in the test set $S_{test}$, and $|S_{test}^P|$ is the number of AS in the positive test subset $S_{test}^P$. $r_b$ is the rank position of functional AS $b$ in all the AS of test set. The lower value of the rank index represents the better performance of FAScore.

## Features of importance

We applied the mean decrease in the Gini Coefficient[45] to evaluate the importance of the feature. This coefficient indicates how well features

split the data at each node in the trees, reflecting its substantial role in improving node purity and classification accuracy. Gini impurity reduction was calculated for each split in decision trees. The decrease in Gini of each feature was summed up across all the nodes and trees. The sum was divided by the total number of splits of features, producing the Mean Decrease Gini index for each feature. The mean decrease Gini index was calculated from the final model and transformed to log2 value.

## Step4: Calculation of the probability to generate the functional AS scores

By evaluating the performance of the model, we chose the best model with the highest AUROC value, AUPRC value, F1 score and the lowest rank index from 10-fold cross-validation. The probability calculated by the best model was the functional AS score (FAScore).

We applied a Gaussian Mixture Model (GMM) with three components using the R package mclust (v5.4.9) to model the distribution of FAScore. The parameters include G = 3 and modelNames = "V". GMM assumes that the data is generated from a mixture of Gaussian distributions, and is represented as:

$$p(x) = \sum_{k=1}^3 \pi_k N(x|\mu_k, \Sigma_k), \qquad (21)$$

where $\pi_k$ represents the mixture proportions, and $N(x, |, \mu_k, \Sigma_k)$ is the Gaussian distribution with mean $\mu_k$ and covariance $\Sigma_k$ for each component $k$.

Based on GMM, AS events were classified to three classes including functional AS, non-functional AS and uncertain AS.

**Verification of functional AS.** The predicted functional AS events were validated for their significance from four different perspectives, including conservation of splice site flanking sequence, function enrichment analysis of host genes, protein functional domain and post-translational modification analysis in regions of alternative splicing exons.

(1) Conservation of splice site flanking sequence

The UCSC phastCons files store conservation scores for the human genome (hg38.phastCons100way.bw) calculated from multiple alignments with other 99 vertebrate species and for the mouse genome (mm10.60way.phastCons.bw) calculated from multiple alignments with other 59 vertebrate species[93]. Two PhastCons files were downloaded from the UCSC database. The PhastCons conservation scores of upstream 100 bp region (intronic) and downstream 50 bp region (exonic) of splicing sites for the functional and non-functional AS were compared.

(2) Enrichment analysis

Gene Ontology (GO) enrichment analysis was performed separately for each lineage using the R package clusterProfiler[94] (v4.2.2). For each lineage, the set of genes harboring functional and uncertain ES events was tested for enrichment against a background set comprising all genes that were expressed (TPM > = 0.5) in at least one sample within one lineage to control for cell lineage-specific expression biases. And "GSEA" function was used to perform enrichment analysis of cell lineage-specific transcription factor sets in genes harboring all AS events from myeloid datasets.

(3) Protein function prediction

The protein domains and motifs of human and mouse were obtained from the UniProt database (http://www.uniprot.org/)[48]. The functional information was classified by Singh[95], containing the annotation information from the UniProt, including active site, domain, transmembrane region, repeat, zinc finger region, compositionally biased region, DNA-binding region, region of interest, lipid moiety-binding region, short sequence motif, calcium-binding region, nucleotide phosphate-binding

region, metal ion-binding site, topological domain. These regions were further classified into seven categories:

a. active site and catalytic site;
b. DNA binding domain (DBD), including C2H2-type, PHD-type, C3H1-type, KRAB, Bromo, Chromo, DNA-binding, C4-type, CHCR, A.Thook, bZIP, bHLH, CCHC-type, CHCH, Bromodomain-like, CH1, C6-type, A.Thook-like, C4H2-type and CHHC-type;
c. protein-protein interaction domain (PPI), including WD, ANK, TPR, LRR, HEAT, Sushi, EF-hand, ARM, PDZ, PH, SH3, RING-type, LIM zinc-binding, WW, SH2, BTB, FERM, CH, Rod, Coil 1 A, MH2, WD40-like repeat, t-SNARE coiled-coil homology, Coil 1B, Cb1-PTB, Coil, CARD, SH2-like, DED, IRS-type PTB, SP-RING-type, EF-hand-like, RING-CH-type, v-SNARE coiled-coil homology, Arm domain, LIM protein-binding, GYF, PDZ domain-binding and PDZD11-binding;
d. RNA binding domain (RBD), including RRM, SAM, KH, DRBM, RBD, Piwi, PAZ, S1 motif, Punmilio and THUMP;
e. transmembrane domain (TMD), including transmembrane region, ABC transmembrane type−1, ABC transporter and ABC transmembrane type-2;
f. Signal peptide;
g. Intrinsically disordered region (IDR), disordered.

(4) Post-translational modification (PTM) analysis

The PTM information of human and mouse was obtained by the PhosphoSitePlus database[49] including acetylation (AC), methylation (ME), phosphorylation (PP), sumoylation (SUMO), ubiquitination (Ubiq). The position of the above region and PTM were converted to the genome coordinate position, and intersected the position of the functional or non-functional AS.

**Analysis of homologous exonic parts.** Using liftOver[96] from the UCSC database, the exonic part regions of human were as used input to retrieve the corresponding homologous exonic part regions (EPRs) from other species (rhesus, mouse, rat, rabbit and zebrafish) in FHO datasets. EPRs with a single corresponding region mapped in other species were retained for downstream analysis. The homologous alternative exonic part (AS exon) was defined to undergo alternative splicing in the species. Based on the evolutionary relationship of the species involved in this study, the exon evolutionary ages of human were inferred with parsimony[97] For a given AS exon in human, we obtained the exon family to which this exon belongs and determined the species with the greatest evolutionary distance from human among those covered by this exon. The evolutionary distance between this species and human is used as the exon evolutionary ages.

**Sequence alignment and phylogenetic construction of TBC1D23.** The DNA and amino acid sequences of long isoform (TBC1D23-L) were downloaded for multiple species, including *Homo sapiens* (ENST00000394144), *Macaca mulatta* (ENSMMUT00000028144), *Mus musculus* (ENSMUST00000023431), *Bos taurus* (ENSBTAT00000032386), *Oryctolagus cuniculus* (ENSOCUT00000017705), *Gallus gallus* (ENSGALT00000024650), *Xenopus tropicalis* (ENSXETT00000025919), *Danio rerio* (ENSDARG00000044357), *Caenorhabditis elegans* (F20D1.2.1) and *Branchiostoma floridae* (XP_035681897.1) from ENSEMBL or NCBI databases. Multiple sequences were aligned using Jalview (2.11.2.4) based on Mafft algorithms with default parameters. The phylogenetic tree was constructed by MEGA (v11) with Neighbor-Joining algorithms with default parameters. Based on the amino acid alignment, the sequences of the 15th exon of TBC1D23 and the flanking regions were extracted from the genomes of multiple species.

**Pattern clustering of FAScores of AS.** The FAScore of the ancient AS from age class 5 were extracted, and the median of all species was used

to cluster functionality patterns by ClusterGVis package (v0.1.1, https://github.com/junjunlab/ClusterGVis) choosing cluster method Mfuzz. The matrix was filtered with a minimum standard deviation threshold of 0.1, and the number of clusters was inferred from the retained AS events by the getClusters function (clusters = 24). The cluster results were visualized using the visCluster function.

**Mice.** C57BL/6 J mice (stock no. N000013) were purchased from GemPharmatech Co. Ltd. The Tbc1d23$^{\Delta E15/\Delta E15}$ (E15-KO) mice were generated in the C57BL/6 J background by GemPharmatech Co. Ltd, whose introns flanking exon 15 of *Tbc1d23* were targeted by 4 sgRNAs using CRISPR/Cas9 technology. Rosa-Cas9 knock-in mice were originally purchased from the Jackson Laboratory (stock no. 024858) and backcrossed onto C57BL/6Jbackground. Both heterozygous (Cas9$^{+/-}$) and homozygous (Cas9$^{+/+}$) Rosa-Cas9 knock-in mice were used in this study and collectively designated as Cas9$^+$ (Supplementary Fig. 6d). All mice used were female and were between 8 and 12 weeks of age. The oligonucleotide (oligo) sequences of sgRNA and genotyping primers were listed in Supplementary Data 11. Unless otherwise stated, all oligos used in this study were synthesized by Sangon Biotech.

All mice were housed and bred under SPF (Specific Pathogen Free) conditions with a 12 hr light/dark cycle at the Laboratory Animal Center of West China Second University Hospital, with free access to food and water. All animal experiments were carried out following the protocols approved by the ethics committee of West China Second University Hospital [(2018) Animal Ethics Approval No.004].

**Genotyping.** The Mouse Tail Direct PCR kit (FORE GENE, #TP-01331) was used to extract DNA from mouse ails. Genomic DNA was extracted from sorted GFP$^+$ cells using the TIANamp Genomic DNA Kit (TIANGEN, #DP304) and cells were lysed in elution buffer containing proteinase K and incubated at 65°C for 1 hour, followed by heat inactivation at 95°C for 5 minutes. PCR was performed for genotyping with primer sequences were listed in Supplementary Data 11.

**Zebrafish.** All zebrafish (*Danio rerio*) experiments were performed according to standard procedures as previously described[56,98,99]. Briefly, both adult fish and embryos were maintained at 28.5 °C in Aquatic Ecosystems. AB (wild-type) and Tg (gata1a:DsRed) strains were used in this study. Larvae were imaged at 48 h postfertilization (hpf), a developmental stage at which sex is not yet distinguishable. All experimental protocols were approved by the Animal Ethical Committee, West China Hospital of Sichuan University.

**Morpholino injection and validation.** Antisense morpholino oligonucleotides (MO) were purchased from Gene Tools. Tbc1d23-E15-MO (E15-MO) targets the exon 15 of the zebrafish Tbc1d23 gene. Control-MO (Ctrl-MO) is a standard, mismatched control. MO were injected (3 ng per embryo) into the embryo at the one-cell stage. To determine the efficiency of MO, total RNAs of ~40 zebrafish embryos at 48 hpf were extracted and were reverse transcribed to complementary DNA (cDNA) for RT-qPCR (qPCR) and RT-PCR (see below). Oligo sequences of MO and primers were listed in Supplementary Data 11.

**Zebrafish fluorescent imaging.** Live embryos were anesthetized and photographed. The fluorescent images were captured by a ZEISS AXIO Zoom (V16) microscope and analyzed using the ZEN (3.1) software and Image J (1.52a, github.com/imagej/ImageJ).

**Whole Embryo Staining for Globin Expression.** O-dianisidine staining was performed to determine the hemoglobin level in the zebrafish embryos as described before[100,101]. The dechorionated embryos at 48 h post-fertilization (hpf) were incubated in the O-dianisidine (Sigma, #243-737-5) staining solution (100 mg O-dianisidine in 70 ml 100% ethanol) for 30 min in dark at room temperature (RT). Embryos were

then washed with PBS with 0.1% Tween 20 (Sigma, #9005-64-5) to stop the reaction and fixed in 4% paraformaldehyde (Biosharp, #BL539A) for at least 3 h before storage in PBS at 4 °C. The stained embryos were imaged on a Nikon SMZ18 stereomicroscope, and the images were used to quantify the staining signals using ImageJ.

**RNA extraction, cDNA synthesis and PCR.** Total RNAs of zebrafish embryos, mouse organs or cell lines were extracted using TRIzol (Life Technologies, #15596026) according to the manufacturer's instructions. Complementary DNA (cDNA) was synthesized using the RevertAid first-strand cDNA synthesis kit (Thermo Scientific, #K1621). RT-PCR was performed using the 2 × Phanta UniFi Master Mix (Vazyme, P516). And real-time qPCR was performed using the Blue qPCR SYBR Master Mix (YEASEN, #11184) on a Bio-Rad (CFX Connect) thermocycler. Primers were listed in Supplementary Data 11.

**Blood routine, cell isolation and flow cytometry analysis.** The blood routine was assayed using BOWLINMAN (BM660). Whole BM cells were collected from the tibia, femur and ilium, filtered through 70-μm nylon mesh, quantified by PBS and counted by Countstar Altair (Counstar). For the analysis of hematopoietic stem and progenitor cells (HSPCs), BM cells were lysed with red blood cell lysis buffer (Solarbio, #R1010), washed with PBS and filtered through 70-μm nylon mesh.

HSPCs were further enriched using c-kit/CD117 MicroBeads (Miltenyi Biotec, #130-09224) or EasySep™ Mouse Hematopoietic Cell Isolation Kit (STEMCELL Technologies, #19856) with a biotinylated anti-mouse lineage cocktail (CD11b, B220, Gr-1, TER119 and CD3e) according to the manufacturer's instructions.

Single cell suspensions were stained with antibodies at 4 °C for 30 min in dark. The antibodies used in the study were listed in Supplementary Data 12. Flow cytometric analysis was performed using an Attune Nxt (Life Technologies) flow cytometer and cell sorting were performed using a FACS Aria III (BD Biosciences). The data were analyzed using FlowJo software (v.10.8.1, BD). Gating strategies for all flow cytometry experiments are illustrated in Supplementary Fig. 13.

**Colony-forming unit assay.** Freshly isolated whole BM cells, enriched c-kit$^+$ cells or sorted GFP$^+$ cells were plated in methylcellulose culture medium (MethoCult™, STEMCELL Technologies) and incubated at 37 °C with 5% $CO_2$. For cells plated on MethoCult™ GF M3334, colonies were counted and scored as colony-forming unit erythroid progenitor cells (CFU-E) at 48 h after plating. Cells plated on MethoCult™ GF M3436 were evaluated 10–14 days after plating and scored as burst-forming unit erythroid progenitor cells (BFU-E). For cells plated on MethoCult™ GF M3434, colonies derived from BFU-E and granulocyte-macrophage progenitor cells (CFU-GM, CFU-G and CFU-M), multipotent granulocyte, erythroid, macrophage, megakaryocyte progenitor cells (CFU-GEMM) were counted 9–12 days after plating.

For cell isolation from MethoCult™ GF M3434 colonies, the methylcellulose medium containing colonies was collected. The sample was washed five times with 10 volumes of PBS. After centrifugation, the supernatant was removed to eliminate residual methylcellulose. Cells were stained with antibodies and sorted by flow cytometry. Gating strategy was illustrated in Supplementary Fig. 13a.

**Single-cell mRNA sequencing.** Female E15-KO and the WT mice aged 8–9 weeks were used for enrichment of LIN- HSPCs as mentioned above. The collected cells were resuspended in PBS containing 1% BSA at the concentration of $1 \times 10^6$ cells/mL and prepared for library preparation. Single cells were prepared in the Chromium Single Cell Gene Expression Solution using the Chromium Single Cell 3′ Gel Bead, Chip and Library Kits v2 (10× Genomics) according to the manufacturer's protocol. Briefly, the cells were loaded in each channel and then partitioned into nanoliter-scale Gel Bead-In-EMulsions (GEMs) using the Chromium instrument, where all cDNA generated from an individual cell share a common 10× Barcode. To identify the PCR duplicates, the Unique Molecular Identifier (UMI) was also added. The GEMs were incubated with enzymes to synthesize full-length cDNA, which was subsequently amplified by PCR to generate a sufficient quantity for library construction. After cDNA amplification, enzymatic fragmentation and size selection were performed using SPRI select reagent (Beckman Coulter, #B23317) to optimize the cDNA size. P5, P7, a sample index and read 2 (R2) primer sequence were added by end repair, A-tailing, adaptor ligation and sample-index PCR. Amplified cDNA and final libraries were assessed with the Agilent BioAnalyzer using a High Sensitivity DNA Kit (Agilent Technologies). Finally, the libraries were sequenced on the Illumina NovaSeq 6000 platform by Novogene, Beijing, China. On average, 101 Gb of raw data were generated for E15-KO and 88 Gb for WT HSPCs.

The sequencing data were processed using the CellRanger (v6.1.2) with default parameters and the GRCm38 mouse reference genome (downloaded from https://ftp.ensembl.org/pub/release-93/fasta/mus_musculus/dna/). The raw gene expression matrices were merged using R (v4.0.2) and converted to a Seurat object using Seurat (v4.0.2). The doublets were identified using DoubletFinder (v2.0.3). Cells having less than 200 unique molecular identifiers (UMIs) or more than 20% mitochondrial counts, or genes below 500 UMIs in all cells were filtered. The batch effect was corrected by a single integrated analysis.

Gene expression matrices of the 12,720 cells after the filter were normalized to the total cellular read count using linear regression using the ScaleData function in Seurat. To reduce the dimensionality of this dataset, the top 2000 highly variable genes were summarized using principal component analysis (PCA). Uniform manifold approximation and projection (UMAP) was generated from the first 20 dimensions using RunUMAP with default parameters. Cell types or clusters were annotated according to the canonical cell markers.

The differentially expressed genes (DEGs) between E15-KO and WT in each cell type were identified using the FindMarkers function in Seurat with default parameters (log2 fold change ≥ 1, and Bonferroni-adjusted $p$ value < 0.05, Wilcoxon Rank Sum test). Cellular differentiation states were predicted by using CytoTRACE (v0.3.3). The Augur (v1.0.3) was used to identify which lineage exhibit the most transcriptional changes between E15-KO and WT. Results were visualized using the ggplot2 package (v3.3.0).

Co-expression analysis was carried out using the hdWGCNA R package[60] (v0.2.2). Erythroid-related cells including ProE/BasoE, PolyE, Early-orthoE and Late-orthoE were selected for co-expression network analysis. We retained 19,655 genes which were expressed in at least 5% of cells from any cluster. Metacell transcriptomic profiles were constructed separately for each of the 2 conditions and each cell type using the hdWGCNA function MetacellsByGroups with parameters 'k = 10, max_shared = 10, min_cells = 0'. We selected a soft-power threshold β = 16 based on the parameter sweep performed with the TestSoftPowers function. The co-expression network was computed with the ConstructNetwork function with the default parameters. Module eigengenes were computed using the ModuleEigengenes function, and we applied Harmony to correct MEs based on the sequencing batch. Eigengene-based connectivity for each gene was computed using ModuleConnectivity. The co-expression network was embedded in two dimensions using UMAP with the RunModuleUMAP function with the top five genes (ranked by eigengene-based connectivity) per module as the input features. Differential module eigengene (DME) analysis was performed using the FindDMEs function, and the Wilcoxon test was used to compare KO and WT groups.

The co-regulatory networks of transcription factors and potential target genes were inferred using pySCENIC[62] (v0.12.1) from scRNA-seq. The erythroid-related cell types were retained. The genes with a total count of less than 27 (1 UMI x 1% of cells) or expressed in less than 1% of the cells are filtered out. 11,450 genes were retained to construct of

gene regulatory network using 'pyscenic grn' with default parameters. And we chose the motif database file with ranking in which matrix containing motifs as rows and genes as columns and ranking position for each gene and motif (based on CRM scores) as values. The search space around the TSS of the gene in which the motif is scored is defined as 500 bp upstream of the TSS and 100 bp downstream. The motif annotations file of mice based on the 2017 cisTarget motif collection was downloaded. We used the 'pyscenic ctx' function to predict the regulons and find the enriched motifs with parameters '--mode dask_multiprocessing' and '--mask_dropouts'. in the following, the activity of regulons was quantified by calculating AUC using the 'pyscenic aucell' function. The co-regulatory network of TFs and their target genes was visualized using Gephi (v0.10.1)[102].

**Cell culture and transient transfection.** HEK-293T and K562 was purchased from Type Culture Collection of the Chinese Academy of Sciences. HEK293T cells were cultured in DMEM medium (Hyclone, #SH30022.01) and K562 cells were cultured in RPMI 1640 medium (Hyclone, #SH30809.01), both supplemented with 10% fetal bovine serum (FBS, NEWZERUM, #FBS-S500) and 1% penicillin-streptomycin (Hyclone, #SV30010).

c-kit+ cells of WT or Cas9+ mice were enriched and cultured in 50% IMDM (Gibico, #12440053) and 50% DMEM medium supplemented with 20% FBS (Gibico, #10099141 C), 10 ng/mL stem cell factor (SCF, Peprotech, #250-03-10), 10 ng/mL interleukin-6 (IL-6, Peprotech, #216-16-10), 2 ng/mL interleukin-3 (IL-3, Peprotech, #213-13-10), 1% penicillin-streptomycin, and 50 μM beta-mercaptoethanol (Sigma, #SLM−063). All cells were incubated at 37 °C in 5% CO2. After 14−16 h, the cultured cells were transduced with retrovirus particles.

Transient transfections were performed with overexpression plasmids (described below) using the Liposomal Transfection Reagent (YEASEN, #40802ES03). The medium was replaced at 4−6 h after transfection, and the cells were harvested at 48 h after transfection.

For viral transduction, HEK293T cells were plated onto six-well plates in 2 ml complete DMEM medium. The next day, cells were transfected with a confluence of 80−90 %. Lentiviruses particles were generated by $CaPO_4$-mediated transfection of HEK293T cells with 2 μg of psPAX2 (Addgene, plasmid #12260), 1 μg of pMD2.G (Addgene, plasmid #12259) packaging plasmid, and 4 μg of the lentiviral vector plasmid. Alternatively, lentiviral particles were generated by Liposomal Transfection Reagent transfection of HEK293T cells with 1 μg pMD2.G, 1 μg pMDLg/RRE (Addgene, plasmid #12251), 1 μg pRsv-Rev (Addgene, plasmid #12253) packaging plasmid and 3 μg of the lentiviral vector plasmid. Retroviruses particles were generated by $CaPO_4$-mediated transfection of HEK293T cells with 1.5 μg pCL-Eco (Addgene, plasmid #12371), 0.2 μg pVSV-G (Addgene, plasmid #138479) packaging plasmids and 4 μg Retroviral vector plasmid. The medium was exchanged for fresh complete medium 6 h after the transfection. All viral supernatants were harvested at 48 h and passed through a 0.45 μm filter to remove cells.

HEK293T cells were transduced with lentivirus particles in the presence of 4 μg/mL polybrene (YEASEN, #40804ES76). K562 cells were transduced with lentivirus particles and spun at 1000 g at 37 °C for 1 h in the presence of 4 μg/mL polybrene. c-kit+ cells were transduced with retrovirus particles and spun at 1,000 g at 37 °C for 1 h in the presence of 8 μg/mL polybrene.

**Generation of cell lines expressing only TBC1D23-S or TBC1D23-L isoform.** We first generated a TBC1D23 knock-out (KO) line by CRISPR/Cas9 methodology, using lentiCRISPR v2 (Addgene, plasmid #52961). The resulting constructs were co-transfected with lentiviruses packaging plasmids (psPAX2 and pMD2.G) to generate lentiviruses particles as described above. The viral supernatants were harvested and added into K562 or HEK293T cells with 4 μg/mL polybrene. K562 or HEK293T cells were selected with 2 μg/mL or 1 μg/mL puromycin (BBI

life sciences, #A610593), respectively at 72 h after infection. Cloning was performed using limiting dilution in 96-well plates and the disruption of TBC1D23 was confirmed using Sanger sequencing of the targeted region and immunoblotting using anti-TBC1D23 antibodies.

For consecutive expression, TBC1D23-S and TBC1D23-L were amplified from K562 cDNAs and cloned into pLVX-mcherry-N1 (TaKaRa, #632562) by Easy Clone Kit I(YoungGen, #CL101A), resulting in overexpression constructs having N-terminally mCherry-tagged TBC1D23-S or TBC1D23-L under a CMV promoter. The overexpression constructs were introduced into the cloned TBC1D23 KO line by lentiviral transduction as described above and the cells were selected with 1 mg/mL G418 (BBI life sciences, #A600013). The expression TBC1D23-S and TBC1D23-L was confirmed using immunoblotting (Supplementary Fig. 9a). Oligo sequences for gRNA or primers were listed in Supplementary Data 11.

For inducible expression, TBC1D23-S and TBC1D23-L were cloned into the tetracycline-inducible lentiviral expression vector pTSB-Tight-EF1-tetR-F2A-Puro, which is a tetracycline (Tet)-inducible expression system[103]. While this system enables high-level transgene expression via Tet operator binding in the presence of tetracycline, the mini-CMV promoter lacks enhancer elements and consequently drives only low-level basal expression in the absence of tetracycline. Lentiviral particles were produced as above and used to transduce TBC1D23-KO HEK293T cells. The expression levels of TBC1D23-S and TBC1D23-L were modulated by titrating the doxycycline concentration to have the comparable TBC1D23 protein level as the control cell line. The oligo sequences for gRNA or primers were listed in Supplementary Data 11.

**Generation of overexpression cell lines.** SUMO1 was amplified from HEK293T cDNAs and cloned into pLVX. The resulting overexpression constructs contain an N-terminally HA-tagged SUMO1 expression cassette and neomycin selection marker. The overexpression constructs were introduced into HEK293T cells by lentiviral transduction with lentiviral packaging plasmids (pMD2.G, pMDLg/RRE, pRsv-Rev) as described above. The target cells were selected with 500 μg/mL G418.

The cDNA fragments of HDAC1 and RanGAP1 were amplified from the cDNAs of K562 cells. HDAC1 was cloned into pcDNA3.1(+) with an N-terminal FLAG tag. Mutant HDAC1 (HDAC1mut, K444R/K476R) was generated by PCR mutagenesis using 2 × Phanta UniFi Master Mix. RanGAP1 was cloned into pcDNA3.1(+) with an N-terminal HA tag. A DNA fragment containing the N-terminally GFP-tagged E3 ligase domain of RanBP2 was synthesized (GENEWIZ) and cloned into PcDNA3.1(+). The E3 catalytic dead mutation of RanBP2 (F2735A/F2736A/C2737A) was generated by PCR mutagenesis using the High-Fidelity PCR kit (MACLAB, #I5HM). The primers were listed in Supplementary Data 11. These overexpression constructs were transiently introduced into HEK293T cells or HA-SUMO1-expression HEK293T cells.

**Generation of exon-deleted and overexpression mouse HSPCs.** The target exon of *Tbc1d23*, *Ebp41l1*, *Ssbp3*, and *Klf6* were deleted by two sgRNAs (listed in Supplementary Data 11) were cloned into pMSCV-U6-sgRNA-IRES-GFP, which was modified from pMSCV-IRES-GFP (Addgene, plasmid #20672). The coding sequence of HDAC1 and HDAC1mut were cloned into pMSCV-IRES-GFP for overexpression.

The resulting constructs were cotransfected with retroviral packaging plasmids (pCL-Eco, pVSV-G) to generate retroviral particles as described above. The viral supernatants were harvested and added into c-kit+ cells of Cas9+ or WT mice with 8 μg/mL polybrene. At 24 h after transfection, the target cells were enriched by GFP.

**Immunoprecipitation and Immunoblotting.** Cells were harvested and lysed in RIPA (Thermo Fischer Scientific, #89900) containing Mini EDTA-free protease inhibitor cocktail (Thermo Fischer Scientific, #A32955) on ice for 30 min, and pelleted. The supernatant was

solubilized in 0.1 M Tris-HCl containing 2% SDS and 0.05 M DTT at 95% for 8 min.

For immunoprecipitation (IP), cells were harvested and lysed in Western-IP lysis buffer (Beyotime, #P0013) containing 10 mM protease inhibitors (Bimake, #B14002), 20mM N-Ethylmaleimide (Sigma, #128-53-0), 1 mM DTT similar to previous studies[104]. Lysates were centrifuged at 14,000 g at 4 °C for 20 min. Rabbit anti-TBC1D23 (Proteintech, #17002-1-AP), mouse anti-FLAG (Proteintech, #20543-1-AP), Rabbit anti-HDCA1 (CST, #34589) antibodies were equilibrated with Tris-buffered saline (TBS) (Sangon Biotech, #C520002) before use. The supernatant from cell lysates was mixed with the protein A/G magnetic beads (Bimake, #B23202) and incubated with antibodies at 4 °C overnight. The protein-antibody-magnetic beads complexes were then washed using TBS three times and eluted with sodium dodecyl sulfate (SDS) loading buffer (YEASEN, #20315ES) at 95 °C for 10 min. The solutes were loaded and separated by 10% SDS–polyacrylamide gel electrophoresis (SDS-PAGE) gel (Epizyme, #PG112) for mass spectrometry or immunoblotting.

Protein extracts or eluted proteins from the IP assay were separated on 10% SDS polyacrylamide gels and transferred to 0.45 μm polyvinylidene difluoride (PVDF) membranes (Beyotime, #FFP28). The membranes were blocked in TBS buffer with 0.1% Tween 20 and 5% skimmed milk at RT for 1 hr. The blots were probed with Rabbit anti-TBC1D23 (1:1000), Mouse anti-HDCA1 (1:1000) (CST, #5356), Rabbit anti-sumo1 (1:1000) (HUABIO, #ET1606-53), Rabbit anti-Sumo2/3 (1:1000) (HUABIO, #ET1701-17), Mouse anti-Flag (1:1000), Mouse anti-GFP (1:1000), Mouse anti-HA (1:1000) or Mouse anti-RanBP2 (1:1000) (Santa Cruz Biotechnology, #sc-74518) at 4 °C overnight. Then the membranes were incubated with the Goat anti-Rabit IgG Seconadry antibody HRP conjugated IgG (1:2000) (SAB, #L3012-2) or Goat anti-Mouse IgG Seconadry antibody HRP conjugated IgG(H + L) (1:2000) (SAB, #L3032-2) at RT for 1 h. The chemiluminescence signal was detected using a ChemiDOCTMMP Imaging System (Bio-Rad Laboratories). Image J software was used to analyze the gray value of the WB.

**IP-Mass**. Gel pieces (number of technical and/or biological replicates=1/group) were destained in 50 mM NH$_4$HCO$_3$/50% ACN, dehydrated with ACN, and subjected to reduction with 10 mM DTT (56 °C, 60 min) and alkylation with 55 mM iodoacetamide (room temperature, dark, 45 min). After washing, gel pieces were digested with 10 ng/μL trypsin at 37 °C overnight. Peptides were extracted with 50% ACN/5% FA followed by 100% ACN, dried, and reconstituted in 2% ACN/0.1% FA. The tryptic peptides were dissolved in 0.1% formic acid (solvent A), directly loaded onto a home-made reversed-phase analytical column (15-cm length, 75 μm i.d.). The gradient was comprised of an increase from 6% to 23% solvent B (0.1% formic acid in 98% acetonitrile) over 16 min, 23% to 35% in 8 min and climbing to 80% in 3 min then holding at 80% for the last 3 min, all at a constant flow rate of 400 nl/min on an EASY-nLC 1000 UPLC system. MS analysis was performed on a Q Exactive™ Plus mass spectrometer with NSI ionization (2.0 kV). Full MS scans (m/z 350–1800) were acquired at 70,000 resolutions, followed by data-dependent MS/MS scans (NCE 28, 17,500 resolution) with a dynamic exclusion of 15.0 s. Data were processed using Proteome Discoverer search engine (v.2.4). Tandem mass spectra were searched against Homo Sapiens (SwissProt, 20366 entries) database with trypsin/P digestion (max 2 missed cleavages), precursor mass tolerance of 10 ppm, fragment tolerance of 0.02 Da, carbamidomethyl (C, fixed), and oxidation (M, variable). Peptides were filtered at high confidence with ion score > 20.

**Immunofluorescence**. The cells were loaded on poly-lysine treated slides and fixed with 4% paraformaldehyde at RT for 15 min. After washing with PBS for three times, cells were permeabilized by 0.1% TritonX-100 at RT for 15 min followed by blocking with 10% FBS in PBS at RT for 1 h. Cell staining was performed using primary antibodies

including Rabbit anti-TGN46 (1:300, Abcam, ab50595), Mouse anti-GM130 (1:200, BD, #610822), Mouse anti-Cl-MPR (1:200, Bio-Rad, #MEM-2), Rabbit anti-ZFPL1 (1:200, Invitrogen, #PA5-53593), Rabbit anti-TBC1D23 (1:50, Sigma, #HPA038152) or Mouse anti-RanBP2 (1:1000) at 4 °C overnight. The cells were washed again with PBS for three times and incubated with secondary antibodies, including FITC affinipure goat anti-Mouse IgG antibody (1:2000, Jackson ImmunoResearch), FITC affinipure goat anti-Rabbit IgG antibody (1:2000, Jackson ImmunoResearch, #115-095-003) or Alexa Fluor 647 affinipure goat anti- Mouse IgG antibody (1:2000, Jackson ImmunoResearch, #115-605-003) at 37 °C for 1 h. The nuclei were stained with DAPI (Sangon Biotech, #E607303). After mounting with VectaShield medium (Vector Laboratories), images were acquired using confocal microscopy (Olympus V3000). The images were used to quantify the Fluorescence intensity using ImageJ.

**SUMO proteome analysis**. TBC1D23 KO K562 cells, or cells expressing only TBC1D23-L or -S (cell number=4 × 10$^7$, number of technical and/or biological replicates=1) were collected for quantification of SUMO modifications using a PTMScan HS Ubiquitin/SUMO remnant kit (Cell Signaling Technologies, #59322) according to the manufacturer's instructions. In brief, the cells were harvested in 9 M Urea Lysis buffer. The lysate was sonicated using a sonicator (BioSafer, #650-92) with a microtip at 15 W output with 3 bursts of 15 s each. Clear the lysate by centrifugation at 20,000 g for 15 min at room temperature and transfer the protein extract (supernatant) into a new tube. Protein concentrations were determined by bicinchoninic acid (BCA) assay (Thermo Fisher Scientific, #23227). DTT (1.25 M) was used for protein reduction, and then iodoacetamide solution (Thermo Fisher Scientific, #A39271) was added to the cleared cell supernatant, and diluted 4-fold with 20 mM HEPES pH 8.0 to a final concentration of ~2 M urea, 20 mM HEPES, pH 8.0. The samples were digested overnight at 37 °C with WALP enzyme (Cell Signaling Technology, #33036) 0.4 mg/mL stock at the ratio of 1 mg enzyme: 100 mg substrate. Complete digestion was confirmed by SDS-PAGE. The digested sample was acidified with 1/20 volume of 20% trifluoroacetic acid to a final concentration of 1%. The pH was checked by spotting a small amount of peptide sample on a pH strip (the pH should be under 3). After acidification, allow the precipitate to form by letting the sample stand for 15 min on ice. Centrifuge the acidified peptide solution for 15 min at 1,780 g at room temperature to remove any precipitate. The peptide-containing supernatant was transferred into a new 50 mL conical tube without dislodging the precipitated material. Acidified and cleared digest was loaded onto the C18 column, which was pre-wetted with 5 mL 100% ACN and washed sequentially with 1, 3, and 6 mL of Peptide Purification Equilibration Solution (0.1% TFA). The column was then washed sequentially with 1, 3, and 6 mL of Peptide Purification Equilibration Solution (0.1% TFA) and with 2 mL of Peptide Purification Wash buffer. Place columns above new 50 mL polypropylene tubes to collect eluate. Elute peptides with a sequential wash of 3 mL and then 7 mL of Peptide Purification Elution Solution (0.1% TFA, 50% acetonitrile). TFA was removed by freezing the eluate on dry ice (or -80 °C freezer) overnight and lyophilizing frozen peptide solution for a minimum of 2 days. The tube containing lyophilized peptides was centrifuged for 5 minutes at 2,000 g at room temperature to collect material for dilution in HS IAP Bind buffer #1. PTMScan HS antibody-bead slurry in ice-cold phosphate-buffered saline (PBS) was added to each clarified sample. Samples were incubated on an end-over-end rotator at 4 °C for 2 h. The beads were collected using a magnetic stand and then washed sequentially with HS IAP Wash buffer and LCMS water to wash the beads. The beads were resuspended in IAP Elution buffer (0.15% TFA) and the eluted sample was collected in a new microcentrifuge tube.

**Mass spectrometry**. Peptides were desalted and resuspended in buffer A (2% ACN, 0.1% FA). Then analyzed by EASY-nanoLC 1000 coupled

to a high-resolution mass spectrometer (Q Exactive Plus, Thermo Fisher Scientific). The samples were loaded onto a 75 µm (inner diameter) × 2 cm (length) trap column and a 75 µm (inner diameter) × 15 cm (length) analytical column, which were packed with C18 particles (DIKMA) in-house. The samples were loaded into a 75 µm (ID) × 2 cm (length) trap column and a 75 µm (ID) × 15 cm (length) analytical column, respectively. Data-dependent acquisition (DDA) was performed in positive ion mode, and samples were analyzed with a 105 min or 65 min gradient from 12 or 13 to 100% buffer B (80% acetonitrile, 0.1% formic acid) for proteomics at a flow rate of 330 nL/min. Full MS scans were obtained with a range from 350 to 1600 m/z with a resolution of 70,000 at m/z = 200, with an AGC target value of $3 \times 10^6$ and a maximum injection time of 20 ms. The AGC value for MS/MS was set to $1 \times 10^5$, the maximum injection time was 100 ms, and the resolution was 35,000. Precursor ions with charge state z = 1, 7–8 or unassigned were excluded. For matching and database searching methods for WaLP digested samples refer to previous studies[105]. GO enrichment analyses were performed by R package clusterProfiler[94] (v4.2.2) for the down-regulated expressed SUMO-modified proteins of TBC1D23-KO compared to cells expressing only TBC1D23-S or TBC1D23-L.

### Reporting summary

Further information on research design is available in the Nature Portfolio Reporting Summary linked to this article.

## Data availability

For lineage differentiation datasets, raw sequencing data of mouse are deposited in the Gene Expression Omnibus (GEO) database under the accession number GSE142216, and raw sequencing data of human are available from the Blueprint Consortium website (http://www.blueprint-epigenome.eu) and the European Genome-phenome Archive with accession number EGAD00001000745. For FHO development datasets, the raw sequencing data are available through the NCBI: human (https://www.ncbi.nlm.nih.gov/bioproject/PRJEB26969), rhesus (https://www.ncbi.nlm.nih.gov/bioproject/PRJEB26956), rabbit (https://www.ncbi.nlm.nih.gov/bioproject/PRJEB26840), mouse (https://www.ncbi.nlm.nih.gov/bioproject/PRJEB26869), rat (https://www.ncbi.nlm.nih.gov/bioproject/PRJEB26889), Opossum (https://www.ncbi.nlm.nih.gov/bioproject/PRJEB27035) and chicken (https://www.ncbi.nlm.nih.gov/bioproject/PRJEB26695). For zebrafish, the accession number for the RNA-seq data is the GEO database: GSE120581. The data of single-cell RNAseq, including sequencing reads and single-cell expression matrices, are available from the GEO: GSE276911. The mass spectrometry proteomics data have been deposited to the ProteomeXchange Consortium (https://proteomecentral.proteomexchange.org) via the iProX partner repository[106,107] with the dataset identifier PXD072614 and PXD072611. Source Data are provided with this paper. The intermediate data related to model training and prediction have been deposited in the Figshare database (https://doi.org/10.6084/m9.figshare.29069336.v4). All newly generated plasmids and other relevant materials are available upon request from the corresponding author. Source data are provided with this paper.

## Code availability

FAScore code, documentation and tutorials are available at GitHub: (https://github.com/LuChenLab/FAScore.git[108]).

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

## Acknowledgements

This work was supported by Noncommunicable Chronic Diseases-National Science and Technology Major Project (Grant No. 2023ZD0500500 to L. C.), the National Natural Science Foundation of China (Grant No. 82370233 to L. C., 92369116 to J.-w.L., 82300133 to C. T., 92254302 and 32430027 to D. J.), National Key Research and Development Program of China (Grant No. 2022YFA1105200 to D. J.), and National Science Fund for Distinguished Young Scholars (Grant No. 32125012 to D. J.). We also appreciate the help of Prof. Lunzhi Dai at Sichuan University for the analysis of Proteomics and mass spectrometry. We would like to thank Huifang Li at Core Facilities of West China Hospital for her assistance in cell sorting.

## Author contributions

L.C., D.J., J.-w.L, and N.S. conceived and supervised the project. X.H. performed all the mouse studies and analyzed the data. Y.Z., G.W., J.L., Z.Z. assisted in the mouse studies. J.W. performed all the cellular studies. X.H. and S.N. assisted in the plasmid construction. Q.Y. performed all zebrafish studies. Li.C. and M.T. performed the machine learning and bioinformatics analysis. Z.X., D.Z., Z.H., and D.L. assisted in bioinformatics analysis. X.H., Li.C., J.W., and Q.Y. wrote the manuscript and generated the figures. B.M., C.T., L.C., D.J., and J.-w.L. edited the manuscript. All authors read and approved the final version of the manuscript.

## Competing interests

The authors declare no competing interests.

## Additional information

[1]Department of Laboratory Medicine, West China Second University Hospital. Key Laboratory of Birth Defects and Related Diseases of Women and Children, Ministry of Education, State Key Laboratory of Biotherapy, Sichuan University, Chengdu, China. [2]Biosafety Laboratory of West China Hospital, Center for Biological and Translational Research, West China Hospital, Sichuan University, Chengdu, China. [3]Human Technopole, Fondazione Human Technopole, Milan, Italy. [4]Wellcome Sanger Institute, Wellcome Genome Campus, Hinxton, UK. [5]Department of Haematology, University of Cambridge, Cambridge Biomedical Campus, Puddicombe Way, Cambridge, UK. [6]British Heart Foundation Centre of Research Excellence, University of Cambridge, Cambridge, UK. [7]National Institute for Health and Care Research Blood and Transplant Research Unit in Donor Health and Behaviour, University of Cambridge, Cambridge, UK. [8]These authors contributed equally: Xiao Hu, Jinrui Wang, Li Chen, Qin Yang, Manuel Tardaguila. [9]These authors jointly supervised this work: Nicole Soranzo, Jing-wen Lin, Da Jia, Lu Chen. ✉e-mail: ns6@sanger.ac.uk; lin.jingwen@scu.edu.cn; jiada@scu.edu.cn; luchen@scu.edu.cn

