## [Transparent Peer Review file · Nature Communications]

The functional landscape of alternative splicing in hematopoietic lineage commitment

Corresponding Author: Professor Lu Chen

Version 0:

Reviewer comments:

Reviewer #1

(Remarks to the Author)

Hu et al report on the functional landscape of alternative splicing in hematopoietic lineages. Alternative splicing is an important post-transcriptional RNA-processing step and contributes to transcript and protein isoform diversity. Importantly, while cataloging AS has become common practice determining the function impact of AS remains difficult to experimentally define and prioritize. This manuscript intends to address this gap by developing a machine learning model, FAScore, that incorporates static information (such as genomic sequence features) and dynamic features (such as expression) to predict which splicing events are functionally relevant. The model employs a random forest framework employing a list of features that incorporates expression, structure and evolutionary conservation. Using the model, they were able to characterize events into “functional” and “non-functional” categories and validate some of the “functional” ones. While there is some excitement regarding the modeling, the curation and assessment of splicing was unclear. However, the experimental follow-up in its current form is challenging to follow as it lacks details. The paper follows primarily one splicing event in the Tbc1d23-E15 in mouse

In summary this is a paper with a significant amount of data however in its current form is challenging to evaluate.

Major

The paper is dense and written as a short form article, but this makes it very challenging to understand and would benefit from more main figures and a significant improvement in writing (primarily adding more information).

Experimental sections lack of sufficient information regarding species, model systems used, validation etc. Much of this information is crucial for an appropriate review.

For validation of mechanism, omic-scale experiments are offered such as scRNAseq, IP-mass spec, mass spec for sumo but these are provided with little context or interpretation.

While I appreciate the work for obtaining FAScores, I'm not sure that the experimental components and the selection of AS for validation necessarily requires this score. One might prioritize the same events based on conservation, protein domain impact, and lineage. Thus, more should be done to discuss why the model facilitates discovery beyond the standard practices of looking at dynamics and conservation.

Description of “Dynamically expressed genes” is difficult to follow. This section needs rewriting and perhaps schematics to help explain the method in more detail. Representative example data for certain genes or alternative splicing events would be helpful.

For expression breadth, it is mentioned that TPMs for each gene was used. TPMs are normalized within samples and are not the best measure for between-sample comparisons. Scaling it between 0 and 1 could lead to confounding effects. What does the overall distribution of dynamic scores look like for genes and AS events. How was 0.4 chosen?

For expression gradient, it's important to show that the relationship was linear between time points and gene expression and AS events which might not always be the case. How were these dealt with? Furthermore, no reasoning was provided for capping a continuous scale such as regression coefficient to -1 and 1. Examples of events that agree with expression

specificity score would be helpful.

Alternative splicing patterns are already shown to be lineage specific (Merkin 2013 Science), so dynamic AS being lineage specific could just be a sub-effect of that.

Sequence conservation is part of the formula used to calculate the score and is one of the top features based on gini index. Could the model be biased towards events with high conservation (i.e. is this circular)? Conservation and protein domain composition is a key feature of gene importance, thus is an ML approach required for identifying these events and prioritizing them?

Does any non-functional or uncertain event have literature on their functional relevance? What is the false negatives on this model?

The statement: "In short, alternative splicing of TBC1D23-E15 leads to differential binding affinity to the RanBP2/RanGAP1 complex and may affect cellular SUMOylation level." is not fully supported by experiment. The raw gels indicate minimal difference in interaction and exclusively in an over-expression setting, further the term affinity is used but not measured using biophysical assays. Similarly, the impact of sumoylation needs to be further validated beyond Co-IP experiments.

Methods are lacking as is model validation for zebrafish and mouse.

(Remarks on code availability)

Reviewer #2

(Remarks to the Author)

This manuscript describes a Random Forest based approach to score alternative splicing (AS) events based on their likelihood to have functional relevance for specific hematopoietic lineages. Based on this score (FAScore) as well as the differential regulation of sequence inclusion, they then selected four AS events, which they functionally validated. They follow up on one of them (exon 15 of TBC1D23), and elucidated the mechanistic details on how the skipping/inclusion of that exon leads to its physiological impact.

Overall, I found the experimental tests and validations to be very well executed, specially for TBC1D23. The development of the FAScore is not particularly original, perhaps with the exception of making it lineage-specific (instead of "global"). However, the authors seem to then average FAScores and used them in combination with the DyScore (i.e. classic delta PSI) to select the events of interest (Fig. 2d).

I have some issues/concerns:

- 1) The positive training set to develop the FAScore is of 63 events according to Sup Table 3 (even though the Methods section says 71). This number seems quite low for an RF approach. Also, looking at the references in Table S3, for some of these events I am not sure there is actual evidence of functionality, only of protein translation by Western blot. This should be clarified. Finally, most of the events used for training are of the exon skipping type (see point 2 below). These limitations should be explicitly acknowledged in the main text.
- 2) It is unclear to me how the benchmarking results in Figures 1 and 2 and in the associated Supplementary Figures look for each type of AS event separately. All events seem to be analyzed and treated together, which I believe is potentially problematic. As mentioned above, FAScore has been mainly trained for exon skipping events and it is known that the level of conservation, functionality, etc. is very different for different types of events.
- 3) Another aspect I found unclear is how the authors have dealt with constitutive, cryptic and "real" AS exons. Based on the number of AS events analyzed, it seems to me they have applied their score to all exons in the genome. What does it mean if a constitutive exon has a high FAScore? Does it mean that its mis-splicing will have a functional impact (as expected)? Again, it would be good to see how the FAScore ranges for the different types of events (constitutive, AS, etc.), and how the benchmarkings perform separately per inclusion type. For instance, if all highly regulated exons (high DyScore) have high FAScores, this means the FAScore is mainly a non-useful score on its own. This score seems to categorize thousands of exons/events as functional, which is likely questionable.
- 3) Perhaps more important: the standard in the field of AS to rank candidates for experimental validations is to select evolutionarily conserved events that change their inclusion levels across lineages and/or a time course. Is the FAScore approach really much better than this? The authors selected four events with high FAScores and high DyScores (i.e. change in their inclusion levels across lineages and/or the time course). If I understood correctly, the most important contributor to the FAScore is conservation and then lineage regulation (which makes it redundant with DyScores). In fact, based on Fig. 2d, the events selected for follow up have a fairly modest FAScore (0.55-0.70) and there are at least dozens of events with much higher FAScores. Why have those events not been prioritized? In other words, the FAScore is perhaps useful, but it seems potentially an oversell.
- 4) What gene sets were used as background for the GO enrichment analyses? In the Methods section it is unclear if they

have used any background at all. If not, it is possible that the enrichments they obtained were due to lineage-specific gene expression biases, which are known to be a major confounder for AS analyses and could explain why they fit the lineages functions well.

5) The zebrafish morpholino experiment is much weaker than all the other experiments. In my opinion, it is OK to keep these results, but I would move them after the first results in mouse (i.e. "white blood cell counts remained unchanged") and introduce them in a very specific manner, e.g.: "To assess the conservation of this phenotype, we"

In summary, I believe the experimental and mechanistic aspects of this manuscript are much stronger than the FAScore, which is perhaps not needed. My recommendation is to downplay this part, and simply sell it as a way to prioritize exons for experimental follow up. As it is now, it has potential flaws and its relevance is oversold, which may require substantial work to solve.

(Remarks on code availability)

Reviewer #3

(Remarks to the Author)

Hu et al. surveyed the transcriptome of the hematopoietic system in six vertebrate species and trained a machine learning model to identify functional lineage-specific alternative splicing events. Their approach relies on a Random Forest model trained on a large dataset of differentially spliced events between lineages and leveraging positive unlabeled learning to take advantage of a set of 71 validated functional events curated from the literature. The interpretation of the model yielded a set of 19 features which were predictive for alternative event functionality. The model was then applied to identify several candidate functional events in lymphoid, myeloid and erythroid commitment. The authors go on to show that one of their candidates, a vertebrate cassette inclusion event in TBC1D23, is important for erythropoiesis using in vivo and ex vivo experiments in the zebrafish and mouse hematopoietic cells. They finally provide mechanistic evidence to demonstrate that the inclusion of the alternative TBC1D23 exon 15 in the erythroid lineage decreases binding to its partner complex RANBP2/RANGAP1, likely leading the activation of transcription factors involved in erythroid differentiation through lower SUMOylation of HDAC1.

This study tackles the extent and the role of alternative splicing during hematopoiesis. This is an important and poorly studied problem with wide ranging applications in fundamental and translational immunology. Notably, the manuscript presents three valuable components to the field: 1) a detailed survey of alternative splicing events across hematopoietic lineages, between species and between developmental stages; 2) a predictive model for splicing event functionality; 3) the characterization of a novel alternative event in TBC1D23 that is key during erythropoiesis.

The methodology leading to the development of the FAScore and DyScore is reasonable and robust. The methods for the experimental validation of candidate functional events are generally sound, but key experiments and details lack in some sections, in particular in demonstrating the lineage specific of TBC1D23 exon 15 inclusion and differential SUMOylation of HDAC1, as described below.

Specific comments:

The astronomical number of differential/lineage-specific splicing events reported and differentially expressed genes in the TBC1D23-E15KO mouse is hard to believe and likely obscures some of the most biologically relevant events. This could be due to lenient filtering criteria and/or specific tools and methods used for transcriptomic analyses. Depending on the version and parameters used, SUPPA2 can have a very high false-positive rate. The choice of performing scRNA-Seq rather than FACS followed by bulk RNA-Seq to identify DEGs in the TBC1D23-E15KO is puzzling. The authors should at least comment on these points and possibly consider revisiting their filtering criteria.

A lot of the evidence cited to support assumptions is anecdotal. For example, the identification of FYN-E5 and PIK3R1-E1 among high FAScore and DyScore events is used as a demonstration that high FAScore and DyScore can be used to find lineage-specific functional events, but how many known lineage-specific functional events can be found among low FAScore/DyScore events?

The experiments in Figure 2E-G must be better described. Currently it is impossible to tell from the main text or figure legend that these experiments use mouse progenitor cells that were transduced with exon targeting gRNAs and then sorted by lineage to perform the colony forming assays. Even the methods do not specify that the experiment was performed in mouse cells and what mouse strain were used. There should also be a panel showing baseline exon inclusion levels by RT-PCR in the control GM, GEMM and BFU-E lineages.

The specificity of TBC1D23 exon 15 inclusion is not shown in any of the figures. This is a critical piece of missing data. Evidence should be provided in the form of RT-PCR on several sorted human and/or mouse hematopoietic cell populations.

Does the MO in Fig 3A-B also affect the levels of the TBC1D23 exon skipping transcript? Currently, it is impossible to confirm that the phenotype is due to loss of TBC1D23-E15 specifically or loss/decrease of all TBC1D23 expression altogether.

Supplementary Fig 5D should include a panel showing TBC1D23 isoform ratios by RT-PCR using a single primer pair in flanking exons.

Nowhere is the difference in HDAC1 SUMOylation between TBC1D23-E15KO and WT shown. This is another critical piece of missing data.

TBC1D23 E15 may alter interaction with partners, but the changes shown in co-IP experiments with RANBP2/RANGAP1 are rather small and downstream changes in HDAC1 SUMOylation are not quantified; how can the loss of a TBC1D23 isoform be expected to lead to the differential expression of close to 6,000 genes?

Line 209: "classed" should be "classified".

(Remarks on code availability)

Version 1:

Reviewer comments:

Reviewer #2

(Remarks to the Author)

Overall, the authors have addressed my initial concerns; however, while doing so (and in response to the other reviewers), I have to say that they have often created more confusion. Specific comments:

- Abstract: "We curated transcriptome data of fetal hematopoietic organ development in six vertebrates and hematopoietic cell differentiation in human and mice, and identified 84,933 cassette events, originating from 30,255 coding proteins." I found this sentence confusing. Checking the main text, it seems it refers to protein-coding genes (as cassette exons do not originate from proteins). Thus, I assume this includes all species together (as there are fewer than 30,255 protein-coding genes in human or mouse alone), which is misleading without further context. Also, the exons are likely non-independent (i.e., orthologous?). This is all too complex for the abstract, in my opinion.

- In the revised version they claim they focus on 58 experimentally validated cassette exons. However, looking at their updated supplementary table (again), there are only 37 unique exons, as many entries involve the same exon coordinate. And these unique exons are actually even fewer, since the human and mouse orthologs are often present in the table. This does not affect the training per se, as this set is only used to define the positive and training sets. However, this is a major reporting mistake, to say the least.

- The use of FAScore throughout the text is confusing and, if I understood it well, a bit misleading. The authors claim that: "To identify functional exon-skipping events, we developed a machine-learning model interrogating 19 features (...), and integrated them into a single prediction score, named FAScore" (Abstract), and "Using these features, we developed a machine-learning model named FAScore (Functional AS Score) to predict functional AS events". If I understood well, there is no single FAScore, but one FAScore for each lineage and species, i.e. eight FAScores if I got it correctly. In other words, the text should be, for instance " " To identify functional exon-skipping events *in a given differentiation lineage and species*, we developed a machine-learning model interrogating 19 features (...), and integrated them into a single prediction score, named FAScore" or "Using these features, we developed machine-learning *models* named FAScores (Functional AS Scores) to predict functional AS events in a given differentiation lineage and species".

The concern is that the term FAScore is used as universal, without this clarification in most instances in the manuscript, which creates confusion. In my opinion, this aspect should be explicitly mentioned and they should refer to a given FAScore as human FAScore_Erythroid, FAScore_Lymphoid, etc. And if they pool all FAScores together for all lineages (as they often do), this should be made explicit.

- A specific case of this confusing use can be seen in Fig. 3b: what are really the data points in each boxplot? The FAScores for each age class for each differentiation lineage? I.e. 4 data points? This would be fine, but it is not clear in the current text and it is not explained in the caption. (Another confusing aspect about this plot: why are there two dots in some cases? Also, I would recommend plotting the proportion of funct/non-funct/uncertain for each age category not the other way around.)

- Another source of confusion: "we measured the FAScores for 188,202 annotated and unannotated AS events". Since they replied to my initial comment that they discard constitutive exons, I assume these 188,202 events are for a mix of species or otherwise the numbers do not add up just for humans. Or are they double counting the metrics for each FAScore type?

- DyScore: this is now more intuitively explained in the first section of Results. However, as above, since the DyScore is specific to a given lineage differentiation (Erythroid, Lymphoid or Myeloid), it should be clear which lineage they are referring to. For example, I assume the cases in Fig. 1 relate only to Erythroid differentiation. If so, this should be more clearly stated, and call it e.g. DyScore_Erythroid to avoid confusion. If my understanding is not correct, then the authors need to clarify this well to avoid misunderstandings.

- Perhaps more relevant, since the DyScore is integrated into the FAScore and not used anymore beyond Fig. 1, why is it even needed?

Minor:

- Fig 2c: the FAScore_Lymphoid seems to do quite poorly. Is it the case?

- Extended data Fig. 6: as far as I understood, Mfuzz was designed to cluster continuous (temporal) data series, not unrelated classes as those here. If so, this implies that the order in which the independent classes are provided could affect the results, which would not make sense.

(Remarks on code availability)

Reviewer #3

(Remarks to the Author)

Hu et al. provided a substantially revised version of their manuscript. Although none of the initial findings or claims had to be revisited, there were important pieces of evidence and technical details missing from the original submission. The authors appropriately clarified or modified the methodology to address the reviewers' comments. They also provided key results to support their initial claims. Most notably, they explained better how they came to such a large amount of differentially expressed/spliced genes and re-adjusted their analysis parameters to highlight the most striking changes, they revisited their prediction model for exon skipping, they provided baseline quantifications of TBC1D23 isoforms across all relevant hematopoietic cell types, they added missing controls in many experiments, and they substantiated their claims linking the change in TBC1D23 isoform ratios to HDAC1 SUMOylation through the RanGAP1-RanBP2 complex with standard biochemical and biophysical assays.

There are the multiple typos and incorrect statements (e.g. a mention of "30,255 coding proteins" in the abstract - but what are "coding proteins"?) that populate the text and that can easily be fixed.

We believe that the revised manuscript delivers the main message of the authors - that they were able to draft a summary of functional alternative splicing events during hematopoiesis - much more successfully than the original manuscript.

(Remarks on code availability)

Response to Reviewers

MS Submission ID: NCOMMS-25-31345

MS title: The functional landscape of alternative splicing in hematopoietic lineage commitment

Dear editors and reviewers:

We thank the editor and the reviewers for their valuable comments and suggestions. In response to the comments, we performed additional analysis and experiments, revised the manuscript (the changes are shown in **red font**) and updated the accompanying figures and tables. We addressed the concerns point-to-point under each comment. And the changes that we introduced into the revised manuscript (MS) are shown in *italics* with page and line numbers in the revised MS indicated.

Yours sincerely,

Lu Chen, PhD

Reviewer #1:

Hu et al report on the functional landscape of alternative splicing in hematopoietic lineages. Alternative splicing is an important post-transcriptional RNA-processing step and contributes to transcript and protein isoform diversity. Importantly, while cataloging AS has become common practice determining the function impact of AS remains difficult to experimentally define and prioritize. This manuscript intends to address this gap by developing a machine-learning model, FAScore, that incorporates static information (such as genomic sequence features) and dynamic features (such as expression) to predict which splicing events are functionally relevant. The model employs a random forest framework employing a list of features that incorporates expression, structure

and evolutionary conservation. Using the model, they were able to characterize events into “functional” and “non-functional” categories and validate some of the “functional” ones. While there is some excitement regarding the modeling, the curation and assessment of splicing was unclear. However, the experimental follow-up in its current form is challenging to follow as it lacks details. The paper follows primarily one splicing event in the *Tbc1d23-E15* in mouse

1. The paper is dense and written as a short form article, but this makes it very challenging to understand and would benefit from more main figures and a significant improvement in writing (primarily adding more information).

Response: We thank the reviewer for the comments and suggestions. We revised our results section and updated the figures to improve the clarity (Revised **Fig. 1**, Revised **Extended Data Fig. 2, 3, 4, 11, 12**), per your suggestion.

2. Experimental sections lack of sufficient information regarding species, model systems used, validation etc. Much of this information is crucial for an appropriate review.

Response: We apologize for the lack of sufficient information and related sections were revised as follows:

(1) In the ‘Results’ section:

For mouse model (Pg. 11, ln. 314-318):

*“To investigate the role of TBC1D23-E15 in vertebrate hematopoiesis, we generated a transgenic mouse line in which *Tbc1d23-E15* was deleted (**Fig. 4a**). In contrast to wildtype (WT) mice expressing both long and short isoforms of *Tbc1d23*, the *Tbc1d23^{ΔE15/ΔE15}* (E15-KO) mice express only the*

short isoform (**Fig 4b, Extended Data Fig. 7d**) with the overall *Tbc1d23* protein level was comparable to WT mice (**Fig. 4c, d**)”

For zebrafish (Pg.12, ln. 321-326):

“In addition, to assess the conservation of this phenotype, we targeted the long isoform of *Tbc1d23* in zebrafish (*Tbc1d23-L*, including *E15*) by morpholino (MO) treatment. The *Tbc1d23-L-MO* (*E15-MO*) treatment specifically reduced the mRNA level of the long isoform while increasing the expression of the short isoform, with no significant effect on total *Tbc1d23* protein level comparable to the fish treated with the control MO (Ctrl-MO) (**Fig. 4g-i, Extended Data Fig. 7g**).”

For cell lines (Pg.13, ln.357-359):

“We generated three in both K562 and HEK293T, one lacking *TBC1D23* expression (KO) and the other two expressing only *TBC1D23-L* or *TBC1D23-S* (**Extended Data Fig. 9a, Fig. 5c, d**)”

(2) The ‘Methods’ section (Pg.31-32, ln.895-933):

“**Mice.** C57BL/6J mice were purchased from GemPharmatech Co. Ltd. The *Tbc1d23* ^{$\Delta E15/\Delta E15$} (*E15-KO*) mice were generated in the C57BL/6J background by GemPharmatech Co. Ltd, whose introns flanking exon 15 of *Tbc1d23* were targeted by 4 sgRNAs using CRISPR/Cas9 technology. *E15-KO* and wildtype littermate controls (WT) of the same sex aged between 8 and 12 weeks were used in all comparison experiments. Rosa-Cas9 knock-in mice were originally purchased from the Jackson Laboratory and backcrossed onto C57BL/6J background. Both heterozygous (*Cas9*^{+/-}) and homozygous (*Cas9*^{+/+}) Rosa-Cas9 knock-in mice were used in this study and collectively designated as *Cas9*⁺ (**Extended Data Fig. 6c**). The oligo sequences of sgRNA and genotyping primers were listed in **Supplementary Table 10**.

All mice were housed and bred under SPF (Specific Pathogen Free) conditions at the Laboratory Animal Center of West China Second University Hospital and were allowed access to diet and water ad libitum. All animal experiments were carried out following the protocols approved by the ethics committee of West China Second University Hospital [(2018) Animal Ethics Approval No.004].

Genotyping. The Mouse Tail Direct PCR kit (FORE GENE, #TP-01331) was used to extract DNA from mouse tails. Genomic DNA was extracted from sorted GFP⁺ cells using the TIANamp Genomic DNA Kit (TIANGEN, #DP304) and cells were lysed in elution buffer containing proteinase K and incubated at 65 °C for 1 hour, followed by heat inactivation at 95 °C for 5 minutes. PCR was performed for genotyping with primer sequences were listed in **Supplementary Table 10.**

Zebrafish. All zebrafish (*Danio rerio*) experiments were performed according to standard procedures as previously described^{56,99,100}. Both adult fish and embryos were maintained at 28.5°C in Aquatic Ecosystems. AB (wild-type) and Tg (*gata1a:DsRed*) strains were used in this study. Larvae were imaged at 48 hours postfertilization (hpf), a developmental stage at which sex is not yet distinguishable. All experimental protocols were approved by the Animal Ethical Committee, West China Hospital of Sichuan University.

Morpholino injection and validation. Antisense morpholino oligonucleotides (MO) were purchased from Gene Tools. *Tbc1d23*-E15-MO (E15-MO) targets the exon 15 of the zebrafish *Tbc1d23* gene. Control-MO (Ctrl-MO) is a standard, mismatched control. MO were injected (3 ng per embryo) into the embryo at the one-cell stage. To determine the efficiency of MO, total RNAs of approximately 40 zebrafish embryos at 48 hpf were extracted and were reverse transcribed to complementary DNA (cDNA) for RT-qPCR (qPCR) and RT-PCR (see below). Oligo sequences of MO and primers were listed in **Supplementary Table 10.**

3. For validation of mechanism, omic-scale experiments are offered such as scRNA-seq, IP-mass spec, mass spec for sumo but these are provided with little context or interpretation.

Response: We apologize for not providing sufficient details and updated the 'Methods' and 'Results' sections in the revised MS to include relevant information.

(1) We added the details of IP-mass spectrometry and SUMO proteome analysis in 'Methods' (Pg.40-42, Ln.1167-1217; Pg.42-43, Ln.1235-1270):

***“Immunoprecipitation and Immunoblotting.** Cells were harvested and lysed in RIPA (Thermo Fischer Scientific, #89900) containing Mini EDTA-free protease inhibitor cocktail (Thermo Fischer Scientific, #A32955) on ice for 30 min, and pelleted. The supernatant was solubilized in 0.1 M Tris-HCl containing 2% SDS and 0.05 M DTT at 95°C for 8 min.*

For immunoprecipitation (IP), cells were harvested and lysed in Western-IP lysis buffer (Beyotime, #P0013) containing 10mM protease inhibitors (Bimake, #B14002), 20mM N-Ethylmaleimide (Sigma, #128-53-0), 1mM DTT similar to previous studies(Wang, Cao et al. 2019). Lysates were centrifuged at 14,000g at 4 °C for 20 min. Rabbit anti-TBC1D23 (Proteintech, #17002-1-AP), mouse anti-FLAG (Proteintech, #20543-1-AP), Rabbit anti-HDCA1 (CST, #34589) antibodies were equilibrated with Tris-buffered saline (TBS) (Sangon Biotech, #C520002) before use. The supernatant from cell lysates was mixed with the protein A/G magnetic beads (Bimake, #B23202) and incubated with antibodies at 4 °C overnight. The protein-antibody-magnetic beads complexes were then washed using TBS three times and eluted with sodium dodecyl sulfate (SDS) loading buffer (YEASEN, #20315ES) at 95 °C for 10 min. The solutes were loaded and separated by 10% SDS-

polyacrylamide gel electrophoresis (SDS-PAGE) gel (Epizyme, #PG112) for mass spectrometry or immunoblotting.

Protein extracts or eluted proteins from the IP assay were separated on 10% SDS polyacrylamide gels and transferred to 0.45 μ m polyvinylidene difluoride (PVDF) membranes (Beyotime, #FFP28). The membranes were blocked in TBS buffer with 0.1% Tween 20 and 5% skimmed milk at RT for 1 hr. The blots were probed with Rabbit anti-TBC1D23 (1:1000), Mouse anti-HDCA1 (1:1000) (CST, #5356), Rabbit anti-sumo1 (1:1000) (HUABIO, #ET1606-53), Rabbit anti-Sumo2/3 (1:1000) (HUABIO, #ET1701-17), Mouse anti-Flag (1:1000), Mouse anti-GFP (1:1000), Mouse anti-HA (1:1000) or Mouse anti-RanBP2 (1:1000) (Santa Cruz Biotechnology, #sc-74518) at 4 °C overnight. Then the membranes were incubated with the Goat anti-Rabbit IgG Secondary antibody HRP conjugated IgG (1:2000) (SAB, #L3012-2) or Goat anti-Mouse IgG Secondary antibody HRP conjugated IgG(H+L) (1:2000) (SAB, #L3032-2) at RT for 1 h. The chemiluminescence signal was detected using a ChemiDOCTMMP Imaging System (Bio-Rad Laboratories). Image J software was used to analyze the gray value of the WB.

IP-Mass. *Gel pieces were destained in 50 mM NH_4HCO_3 /50% ACN, dehydrated with ACN, and subjected to reduction with 10 mM DTT (56°C, 60 min) and alkylation with 55 mM iodoacetamide (room temperature, dark, 45 min). After washing, gel pieces were digested with 10 ng/ μ L trypsin at 37°C overnight. Peptides were extracted with 50% ACN/5% FA followed by 100% ACN, dried, and reconstituted in 2% ACN/0.1% FA. The tryptic peptides were dissolved in 0.1% formic acid (solvent A), directly loaded onto a home-made reversed-phase analytical column (15-cm length, 75 μ m i.d.). The gradient was comprised of an increase from 6% to 23% solvent B (0.1% formic acid in 98% acetonitrile) over 16 min, 23% to 35% in 8 min and climbing to 80% in 3 min then holding at 80% for the last 3 min, all at a constant flow rate of 400 nl/min on an EASY-nLC 1000 UPLC system. MS analysis was performed*

on a Q Exactive™ Plus mass spectrometer with NSI ionization (2.0 kV). Full MS scans (m/z 350–1800) were acquired at 70,000 resolutions, followed by data-dependent MS/MS scans (NCE 28, 17,500 resolution) with a dynamic exclusion of 15.0 s. Data were processed using Proteome Discoverer search engine (v.2.4). Tandem mass spectra were searched against Homo Sapiens (SwissProt, 20366 entries) database with trypsin/P digestion (max 2 missed cleavages), precursor mass tolerance of 10 ppm, fragment tolerance of 0.02 Da, carbamidomethyl (C, fixed), and oxidation (M, variable). Peptides were filtered at high confidence with ion score > 20.

...

SUMO proteome analysis. TBC1D23 KO K562 cells (4×10^7), or cells expressing only TBC1D23-L or -S were collected for quantification of SUMO modifications using a PTMScan HS Ubiquitin/SUMO remnant kit (Cell Signaling Technologies, #59322) according to the manufacturer's instructions. In brief, the cells were harvested in 9M Urea Lysis buffer. The lysate was sonicated using a sonicator (BioSafer, #650-92) with a microtip at 15 W output with 3 bursts of 15 s each. Clear the lysate by centrifugation at 20,000 g for 15 min at room temperature and transfer the protein extract (supernatant) into a new tube. Protein concentrations were determined by bicinchoninic acid (BCA) assay (Thermo Fisher Scientific, #23227). DTT (1.25 M) was used for protein reduction, and then iodoacetamide solution (Thermo Fisher Scientific, #A39271) was added to the cleared cell supernatant, and diluted 4-fold with 20 mM HEPES pH 8.0 to a final concentration of approximately 2 M urea, 20 mM HEPES, pH 8.0. The samples were digested overnight at 37 °C with WALP enzyme (Cell Signaling Technology, #33036) 0.4 mg/mL stock at the ratio of 1 mg enzyme: 100 mg substrate. Complete digestion was confirmed by SDS-PAGE. The digested sample was acidified with 1/20 volume of 20% trifluoroacetic acid to a final concentration of 1%. The pH was checked by spotting a small amount of peptide sample on a pH strip (the pH should be under 3). After acidification,

allow the precipitate to form by letting the sample stand for 15 min on ice. Centrifuge the acidified peptide solution for 15 min at 1,780 g at room temperature to remove any precipitate. The peptide-containing supernatant was transferred into a new 50 mL conical tube without dislodging the precipitated material. Acidified and cleared digest was loaded onto the C18 column, which was pre-wetted with 5 mL 100% ACN and washed sequentially with 1, 3, and 6 mL of Peptide Purification Equilibration Solution (0.1% TFA). The column was then washed sequentially with 1, 3, and 6 mL of Peptide Purification Equilibration Solution (0.1% TFA) and with 2 mL of Peptide Purification Wash buffer. Place columns above new 50 mL polypropylene tubes to collect eluate. Elute peptides with a sequential wash of 3 mL and then 7 mL of Peptide Purification Elution Solution (0.1% TFA, 50% acetonitrile). TFA was removed by freezing the eluate on dry ice (or -80°C freezer) overnight and lyophilizing frozen peptide solution for a minimum of 2 days. The tube containing lyophilized peptides was centrifuged for 5 minutes at 2,000 g at room temperature to collect material for dilution in HS IAP Bind buffer #1. PTMScan HS antibody-bead slurry in ice-cold phosphate-buffered saline (PBS) was added to each clarified sample. Samples were incubated on an end-over-end rotator at 4 °C for 2 h. The beads were collected using a magnetic stand and then washed sequentially with HS IAP Wash buffer and LCMS water to wash the beads. The beads were resuspended in IAP Elution buffer (0.15% TFA) and the eluted sample was collected in a new microcentrifuge tube.”

Additionally, we added 2 schematic diagrams to illustrate the experimental designs and workflow for both IP-mass spectrometry and SUMO proteome analysis (Revised **Fig. 5a, g**).

Revised **Fig. 5a**, Scheme of IP-MS study design created with Biorender.com. Plasmids encoding empty mCherry vector (mCherry), mCherry-TBCD23-L, or mCherry-TBCD23-S were overexpressed in TBC1D23-KO K562 cells, followed by IP assays using an anti-TBC1D23 antibody.

Revised **Fig. 5g**, Diagram of the experimental design and workflow for MS analysis of SUMOylation peptides affected by TBC1D23-L or TBC1D23-S. The plasmids of mCherry, mCherry-TBCD23-L and mCherry-TBC1D23-S were overexpressed in TBC1D23-KO K562 cells. Cells were lysed in a urea-based buffer, followed by WALP digestion of the proteins. The peptides were purified by reversed-phase, solid-phase extraction. Using magnetic antibody beads targeting the SUMO remnant motif (K- ϵ -GG), SUMOylation peptides were then captured by immunoprecipitation and eluted to concentrate them for LC-MS/MS analysis. Created with BioRender.com.

4. While I appreciate the work for obtaining FAScores, I'm not sure that the experimental components and the selection of AS for validation necessarily requires this score. One might prioritize the same events based on conservation, protein domain impact, and lineage. Thus, more should be done to discuss why

the model facilitates discovery beyond the standard practices of looking at dynamics and conservation.

Response: Thanks for the reviewer's comments. We would like to clarify how FAScore aides in the identification of functional AS in this study.

Firstly, while sequence conservation and protein domain are among the most important features, they are not the only determinants for prediction. Our model integrates 19 features into a unified FAScore and classified the candidates into 'functional AS', 'non-functional AS', and 'uncertain AS'. We observed the correlation between conservation and FAScore was not as high as one may imagine (Revised **Extended Data Fig. 3a**). Notably, FAScore successfully identifies many functional AS events with low sequence conservation. For instance, known functional AS events such as *Kdm1a-E7* (Zibetti, Adamo et al. 2010) and *EHMT2-E10* (Fiszbein, Giono et al. 2016) exhibit low conservation but high FAScores (Revised **Extended Data Fig. 3b**). This demonstrates that relying solely on conservation for prioritization would miss these important candidates.

Secondly, the features used in the model were of fine granularity, for example, for the protein domain features, we interrogated features including structural homologs and integrity, domain integrity, transmembrane helices, signal peptides, and target peptides. Our model can quantify the contribution of different features (Revised **Extended Data Fig. 2d**), avoiding the subjective biases in rule-based filtering approaches.

Finally, unlike traditional filtering methods that yield a large set of candidate AS events without internal prioritization, FAScore provides a continuous score. This enables ranking of candidates by their relative score, thereby offering a quantitative framework for prioritizing targets and increasing the efficiency of downstream experimental validation.

We also added text in the discussion regarding why the model facilitates discovery beyond the standard practices of looking at dynamics and conservation. (Pg.16, ln. 448-454)

“Our model integrates multiple features into a unified predictive score, and can automatically quantify the contribution of different features to the prediction, avoiding the subjective biases in features-based filtering approaches. Finally, compared to traditional selecting methods that yield a large set of candidate AS events without prioritization, FAScore provides a continuous prediction score. This enables ranking candidates by their relative score, thereby offering a quantitative framework for prioritizing targets and increasing the efficiency of downstream functional validation.”

Revised **Extended Data Fig. 3a**, Density plot showing the correlation between FAScore and evolutionary conservation across species in ES events, with darker regions indicating higher density of events. The red circle highlights a region containing splicing events with high FAScore but low conservation. Spearman correlation coefficient and corresponding p-value are shown.

Revised **Extended Data Fig. 3b**, Scatter plot showing the correlation between FAScore and evolutionary conservation score across species. Each point represents a known functional splicing event.

Revised **Extended Data Fig. 2d**. Importance ranking of features used in the final model by average Gini index

5. Description of “Dynamically expressed genes” is difficulty to follow. This section needs rewriting and perhaps schematics to help explain the method in more detail. Representative example data for certain genes or alternative splicing events would be helpful.

Response: We thank the reviewer for this constructive suggestion. We included a schematic diagram to visually illustrate the workflow for identification of dynamically expressed genes and AS events across hematopoietic lineages or developmental stages (Revised **Fig. 1b**). We also incorporated examples of dynamically-expressed gene and splicing events, such as *MEIS1*, *MEIS1-E12*, a known functional AS event (Zeddies, Jansen et al. 2014), and *MEIS1-E1* (Revised **Fig. 1c-e**).

(Pg. 5-6, ln. 124-160)

“Genes or AS events that are dynamically regulated during developmental or cell differentiation trajectories are more likely to play critical roles in these biological processes, as their temporal regulation often reflects direct functional involvement in key cellular transitions or fate decisions³⁴. To quantify the dynamic changes in gene expression and alternative splicing, we computed a series of feature scores based on the expression levels or PSI values across developmental time points or differentiated cell types (**Fig. 1b**). Specifically, we calculated the expression breath (Gene.Range and AS.Range) to assess the magnitude of change, the correlation coefficient (Gene.cor and AS.cor) and its statistical significance (Gene.cor.p and AS.cor.p) to evaluate the association with temporal or differentiation trajectories, the linear regression slope (Gene.slope and AS.slope) and its statistical significance (Gene.slope.p and AS.slope.p) to determine the rate and statistical significance of the progressive changes, as well as tau (Gene.tau and AS.tau) to measure expression specificity (**Fig. 1b, see Methods**). As a representative example, the MEIS1, a transcription factor whose inactivated impaired hematopoietic stem cell niche development, megakaryocyte and platelet deficiencies in mice⁴²⁻⁴⁴. We quantified six features of the dynamic gene expression of MEIS1. It showed a gradually decreased expression in hematopoiesis with an upregulation in megakaryocyte (**Fig. 1c**), consistent with a previous report⁴⁵. Meanwhile, MEIS1 contains two ES events, the exon (E) 1 and 12. The MEIS1-E12 exhibits high level of changes (**Fig. 1d**), consistent with its function in inducing the fate decision of human hematopoietic progenitors toward megakaryocyte-erythroid progenitors (MEPs)⁴⁶. In contrast, the MEIS1-E1 showed a stable expression across cell types with low values of dynamic features (**Fig. 1e**).

To facilitate intuitive assessment of the dynamics of genes and AS events, we integrated the dynamic features described above into a unified dynamism score (DyScore). Based on its bimodal distribution, we applied Gaussian

Mixture Modeling (GMM) to classified the genes or AS events into "dynamic" and "non-dynamic" (**Extended Data Fig. 1c, e, see Methods**). Overall, 27% of AS events and 56% of genes were classified as dynamic (**Extended Data Fig. 1d, f**). Notably, in both human and mouse, dynamically regulated AS events were more likely to be cell lineage-specific, with 63% dynamic in only one hematopoietic lineage compared to 29% of dynamically expressed genes (**Extended Data Fig. 1g**). Moreover, dynamically regulated AS exhibited significantly higher evolutionary conservation scores in the regions flanking the splice sites across all lineages and fetal hematopoietic organ (paired t-test, $p < 1.5 \times 10^{-4}$, **Extended Data Fig. 1h**). These results suggest that dynamically regulated AS events may play specialized and conserved regulatory roles in hematopoietic lineage, highlighting the importance of splicing dynamics in the spatiotemporal control of transcriptome complexity during hematopoiesis."

Revised Fig.1. The examples of dynamically regulated gene and AS events.

b, Schematic representation of dynamic profiling for gene expression and alternative splicing (AS). A linear fit (blue dashed line) and a Loess curve (green solid line) are shown, with the shaded area representing the 95% confidence interval. Derived features include: range of genes (Gene.range) or AS (AS.range) to assess variation

breadth; correlation coefficient (*Gene.cor*, *AS.cor*) and its *p*-value (*Gene.cor.p*, *AS.cor.p*) to quantify association with temporal or differentiation trajectories; linear regression slope (*Gene.slope*, *AS.slope*) and its *p*-value (*Gene.slope.p*, *AS.slope.p*) to estimate rate and significance of change; and tau (*Gene.tau*, *AS.tau*) to evaluate specificity.

c-e, Expression and splicing patterns of *MEIS1* and its two AS events during erythroid differentiation. (c) Expression values (transformed TPM) of *MEIS1* (Left) and the values of dynamic features (Right). Percent spliced in (PSI) values (Left) and dynamic feature scores (Right) for the exon 12 skipping event in transcript *MEIS1-201* (d) and the exon 1 skipping event in transcript *MEIS1-202* (e). A linear fit (blue dashed line) and a Loess curve (green solid line) are shown, with the shaded area representing the 95% confidence interval.

6. For expression breadth, it is mentioned that TPMs for each gene was used. TPMs are normalized within samples and are not the best measure for between-sample comparisons. Scaling it between 0 and 1 could lead to confounding effects. What does the overall distribution of dynamic scores look like for genes and AS events. How was 0.4 chosen?

Response: We used TPMs because TPMs normalize both sequencing depth and gene length (Zhao, Li et al. 2021). This double normalization makes TPM values comparable both between samples (accounting for sequencing depth) and between genes (accounting for gene length) within a sample. This comparability is essential for robust calculation of dynamic scores and for meaningful cross-gene comparisons (such as expression range) of the scores. Methods such as DESeq2 (based on library size) (Love, Huber et al. 2014) are designed for comparisons between samples/conditions; however, it is not ideal for comparisons across genes needed for dynamic scores since DESeq2 does not normalize gene length.

In a comparison analysis, we found high correlations between TPM and DESeq2 normalized counts within the same samples (median Spearman ρ =

0.96), and we also found high consistency between two normalisation methods for the same gene (median Spearman $\rho = 0.80$), such as *Paqr8*, *Eya1* and *Stau2*. Therefore, we maintained the use of TPM as it inherently provides the cross-gene and cross-sample comparability which is fundamental to our analysis of the expression breadth and dynamics. The additional analysis was included in Revised **Extended Data Fig. 11a-c**.

Second, the expression breadth for each gene (Gene.range) was calculated by computing the difference (max - min) of the log₂-transformed values ($\log_2(\text{TPM} + 1)$) within each gene (Revised **Extended Data Fig. 11d**). Therefore, when the expression breadth is 0, it means no difference. Meanwhile, when the expression breadth is 1 or greater than 1, it means a minimum 2-fold expression difference ($\log_2(\text{FC}) = 1$) within a gene. We use this cutoff to category a gene as highly variable (Revised **Extended Data Fig. 11e**). The rationale is that the DyScore of a gene is calculated by adding these features together, so we need the expression breadth to have an upper limit. Otherwise, the DyScore may be inflated by a single feature. For AS events, we directly computed the difference of PSI values within each AS event (AS.range). Since PSI is intrinsically bounded between 0 and 1, the AS.range values also naturally fall within the 0 to 1 range, consistent with genes (Revised **Extended Data Fig. 11f**).

Third, we added the distribution of absolute dynamic score (Revised **Extended Data Fig. 1c, e**). Based on its bimodal distribution, we applied Gaussian Mixture Modeling (GMM) to classify the genes or AS events into "dynamic" and "non-dynamic". A GMM with $k=2$ components effectively captured the bimodal distribution (G1 and G2). The optimal threshold separating the two components was determined by the probability density intersection method, resulting in a cut-off of 0.467 for genes (Revised **Extended Data Fig. 1e**) and 0.324 for AS events (Revised **Extended Data Fig. 1c**). We then calculated the arithmetic mean of the gene-level and AS-level cutoffs, which was 0.40.

We revised this section in 'Methods' as follows:

(1) Expression breadth: range value (R) (Pg.19, ln.548-559)

*“The value range (R) represents the difference between the maximum and minimum expression values, reflecting the extent of variation. For genes, expression breadth (R_{gene}) was calculated as follows: a pseudo-count of 1 was added to the TPM values, which were then \log_2 -transformed; R_{gene} was then computed as the difference (max – min) among the transformed values for each gene (**Extended Data Fig. 11d**). when the expression breadth is 0, it means no difference. Meanwhile, when expression breadth is 1 or greater than 1, it means a minimum 2-fold expression difference ($\log_2(FC) = 1$) within a gene. Thus, we applied a transformation step wherein any R_{gene} value ≥ 1 was set to 1 (**Extended Data Fig. 11e**). For AS events quantified using Percent Spliced In (PSI), the range (R_{AS}) was directly computed as the difference in PSI values for each AS event. Since PSI is naturally bounded between 0 and 1, R_{AS} values also inherently fall within the 0–1 range (**Extended Data Fig. 11f**).”*

(2) Definition of dynamic genes and AS (Pg.22, ln.620-628)

“Based on bimodal distribution of absolute dynamic score, we applied Gaussian Mixture Modeling (GMM) to classify the genes or AS events into “dynamic” and “non-dynamic”. A GMM with $k = 2$ components effectively captured the bimodal distribution. The optimal threshold separating the two components was determined by probability density intersection method, resulting in a cut-off of 0.467 for genes and 0.324 for AS events. We then calculated the arithmetic mean of the gene-level and AS-level cutoffs, and it was 0.40. Finally, genes or AS events were divided into two classes, including dynamically expressed genes/AS events (≥ 0.4 or ≤ -0.4), and non-dynamically expressed genes/AS events ($-0.4 \sim 0.4$).”

Revised Extended Data Fig. 11. The distribution of expression breadth of genes and alternative splicing events.

a-c, Scatter plots showing the correlation between TPM and DESeq2-normalized expression values for three example genes: (a) *Paqr8* ($R = 0.94$, $p < 2.2 \times 10^{-16}$), (b) *Eya1* ($R = 0.96$, $p < 2.2 \times 10^{-16}$), and (c) *Stau2* ($R = 0.96$, $p < 2.2 \times 10^{-16}$). The blue line in each panel represents the linear regression fit.

d-f, Density distributions of expression breadth of genes and AS, including Gene.range (d), Capped Gene.range (e) and AS.range (f).

Revised **Extended Data Fig. 1. c, e**, Distribution of absolute dynamic score (DyScore) shown as overlaid histogram and KDE for AS events (c) and genes (e). G1 and G2 denote the two fitted Gaussian components used for classifying dynamic and non-dynamic elements.

7. For expression gradient, it's important to show that the relationship was linear between time points and gene expression and AS events which might not always be the case. How were these dealt with? Furthermore, no reasoning was provided for capping a continuous scale such as regression coefficient to -1 and 1. Examples of events that agree with expression specificity score would be helpful.

Response: We agree that temporal relationships between time points and gene expression and AS events are not necessarily linear. We used generalised additive modeling (GAM) and effective degrees of freedom (edf) to objectively

characterize relationship linearity. In GAMs, the edf is a measure of a smooth term's "wiggleness" or complexity, reflecting how much the model is allowed to deviate from a straight line. According to the common classification method of edf(Hunsicker, Kappel et al. 2016), AS or genes can be divided into three categories: linear (edf \approx 1), weakly non-linear ($1 < \text{edf} < 2$) and strongly non-linear (edf > 2) (Revised **Extended Data Fig. 12a**). The result showed predominant linear tendencies (51.8%), with edf values clustering near 1 (Revised **Extended Data Fig. 12a**).

Additionally, we found slope value from linear regression was representative in the three categories. For linear patterns (edf \approx 1), slopes provided optimal quantification of the change rates (e.g., *MED24*; Revised **Extended Data Fig. 12b**). For weakly nonlinear patterns ($1 < \text{edf} < 2$), slopes remained suitable change rate indicators in 51% of cases (validated by $>50\%$ variance explained) (Revised **Extended Data Fig. 12b**, e.g., *MAD1L1*). Within strongly nonlinear (edf > 2), slopes effectively captured the change rates for genes/AS exhibiting monotonic trends (Revised **Extended Data Fig. 12b**, e.g., *FGG*). However, for strongly nonlinear features with non-monotonic oscillations, the sign of slopes represents only the direction of change (Revised **Extended Data Fig. 12c**, e.g., *HCCS*). To address this, we introduced Range metrics as the complementary global descriptors. Range provides supplementation when slopes underestimate dynamics. Thus, linear fitting is an important feature we assess for dynamic gene or AS regulation. In sum, the majority of temporal relationships between time and expression and AS events are linear or close to linear. To simplify the process and comparability, we used linear regression in our study.

For the regression coefficient (slope), to address potential skewness in gene expression distributions while preserving biological fidelity, we performed the $\log_2(\text{TPM}+1)$ transformation for gene expression values followed by linear regression against normalized timepoints. While this transformation mitigated biases from highly expressed genes, initial slopes spanned from -2 to 2 (with

the majority of them distributed around 0). However, these slopes exhibited only low correlation with raw TPM slopes (Pearson $r = 0.17$, Revised **Extended Data Fig. 12d**). Subsequently, we applied the capping at $[-1,1]$ to extreme values (affecting $<1\%$ of genes), and this optimization substantially improved the correlation to r of 0.84 (Revised **Extended Data Fig. 12e**), suggesting an improved biological relevance. The optimized slope values also enabled direct additive integration of different features, providing a mathematically consistent foundation for downstream dynamic score calculation. For AS events, PSI values inherently generated slopes within $[-1,1]$ without processing (Revised **Extended Data Fig. 12f**), maintaining analytical consistency with the processed gene expression metrics.

We revised the 'Methods' section (Pg.20-21, ln.576-600) as follows:

*“To objectively characterize the linearity of relationships, we employed generalized additive modeling (GAM) and effective degrees of freedom (edf), which quantifies the complexity or “wiggleness” of a smooth term, reflecting the extent to which the model deviates from a straight line. The observation that edf values clustered near 1 (Hunsicker, Kappel et al. 2016) indicates that gene expression and splicing levels exhibit predominantly linear tendencies (51.8%) (**Extended Data Fig. 12a-c**). Accordingly, we primarily conducted linear regression analyses, where the regression coefficient (β) represents the magnitude of influence exerted by the independent variable on the dependent variable, and the p -value of the regression variable (P_β) reflects the statistical significance of this association.*

To mitigate potential skewness in gene expression distributions while preserving biological interpretability, we applied a $\log_2(\text{TPM} + 1)$ transformation to gene expression values, which were then subjected to linear regression against normalized differentiation cell types or developmental time points. Although this transformation reduced bias from highly expressed genes, the resulting slopes initially ranged from -2 to 2, with

most values concentrated near 0. These initial slopes showed only weak correlation with slopes derived from raw TPM values (Pearson $r = 0.17$, **Extended Data Fig. 12d**). We therefore applied capping to extreme values at the range $[-1, 1]$, which affected fewer than 1% of genes. This adjustment substantially improved the correlation to $r = 0.84$ (**Extended Data Fig. 12e**), suggesting enhanced biological relevance. The optimized slope values also support direct additive integration across different features, providing a mathematically consistent basis for downstream dynamic score calculation. For alternative splicing (AS) events, PSI values inherently yield slopes within $[-1, 1]$ without additional processing (**Extended Data Fig. 12f**), ensuring analytical consistency with processed gene expression metrics. Furthermore, p -values were binarized: values below 0.05 were assigned a significance label of 1, and all others were labeled 0.”

Revised Extended Data Fig. 12. Systematic validation of linear modeling approach as dynamics features.

a, Distribution of effective degrees of freedom (edf) from generalized additive models (GAM). Categories: linear ($edf \approx 1.0$), weakly nonlinear ($1 < edf < 2$), strongly nonlinear ($edf > 2$).

b, Dynamics of expression of representative genes (MED24, MAD1L1, FGG) across edf pattern categories. The yellow solid lines represent the GAM fits, while the blue dashed line indicates the linear regression fits. Gray shaded areas represent 95% confidence intervals.

c, Dynamics of expression of representative genes (HCCS) in high edf and near-zero slope. The yellow solid lines represent the GAM fits, while the blue dashed line

indicates the linear regression fits. Gray shaded areas represent 95% confidence intervals.

d, Correlation between expression slopes of raw and log-transformed TPM expression per gene. Pearson's correlation coefficients (R) and p -value are labeled.

e, Correlation of capped slope between raw and log-transformed TPM expression per gene. Pearson's correlation coefficients (R) and p -value are labeled.

f, Density distribution of slopes of alternative splicing based on PSI.

8. Alternative splicing patterns are already shown to be lineage specific (Merkin 2013 Science), so dynamic AS being lineage specific could just be a sub-effect of that.

Response: Merkin et al. (2013 Science) used "lineage" in the context of evolutionary divergence (e.g., rodent-specific, primate-specific, or mammalian-conserved AS events). This analysis compared AS patterns across species (mouse, rat, macaque, cow, chicken) to identify phylogenetically restricted splicing and established the concept of lineage-specific alternative splicing (AS) in evolutionary biology. The word "lineage" in our study refers to the hematopoietic cell lineage during including myeloid, erythroid and lymphoid lineages. To avoid confusion, we used the wording of "cell lineage-specific" in the revised MS.

9. Sequence conservation is part of the formula used to calculate the score and is one of the top features based on gini index. Could the model be biased towards events with high conservation (i.e. is this circular)? Conservation and protein domain composition is a key feature of gene importance, thus is an ML approach required for identifying these events and prioritizing them?

Response: Although sequence conservation and protein domain are the top contributing features, they are not always the decisive ones, given that our model integrates multiple features into a unified predictive score.

As shown in the Revised **Extended Data Fig. 3a**, the conservation and FAScores are weakly correlated, and FAScores predicts many AS events with low sequence conservation. Indeed, when we plotted the conservation and FAScores of the AS events with known function (published in literature), some are of low sequence conservation, such as *Kdm1a-E7* (Zibetti, Adamo et al. 2010), *EHMT2-E10* (Fiszbein, Giono et al. 2016) (Revised **Extended Data Fig. 3b**). Therefore, if we only use conservation to prioritize AS events, these events would be missed.

To address the concerns on the model biased towards events with high conservation, the two heat maps in revised **Extended Data Fig. 3c, d** demonstrated that the conservation and FAScore are weakly related as there are events with high conservation but having low FAScore and events with high FAScore with low conservation (boxed in revised **Extended Data Fig. 3c,d**).

As to protein domain composition, it ranked the fifth contributing 5.6% importance using Gini index. Moreover, the percentage of functional AS events with defined domain was only 28.5%. The selection based on domain composition may also miss interesting AS events. For example, in this study we showed that *TBC1D23-E15*, which only encodes 15 amino acids with no indicated domain or motif, plays important roles in erythropoiesis.

Revised Extended Data Fig. 3. Correlation between FAScore and evolutionary conservation across alternative splicing events.

a, Density plot showing the correlation between FAScore and evolutionary conservation across species in ES events, with darker regions indicating higher density of events. The red circle highlights a region containing splicing events with high FAScore but low conservation. Spearman correlation coefficient and corresponding p -value are shown.

b, Scatter plot showing the correlation between FAScore and evolutionary conservation score across species. Each point represents a known functional splicing event.

c, Heatmap displays normalized feature scores for high conservation with high FAScore (50 were randomly selected) or low FAScore (50 were randomly selected) AS

events identified via PU-learning. Each row represents an AS event, and each column corresponds to a predictive feature, with the final FAScore.

d, Heatmap displays normalized feature scores for high FAScore with high conservation (50 were randomly selected) or low conservation (50 were randomly selected) AS events identified via PU-learning. Each row represents an AS event, and each column corresponds to a predictive feature, with the final FAScore.

10. Does any non-functional or uncertain event have literature on their functional relevance? What is the false negatives on this model?

Response: Our final model was trained using reliable positive and negative sets inferred by the positive set from the literature through a PU-learning process, with 20% of the data held out as a validation set. The trained model achieved a low false negative rate of 0.050 on this validation set (10-fold cross validation, sd = 0.012).

As shown in **Figure R1.2**, the true positive set in the validation set mostly have high FAScore. As the publications on functional AS were still limited, we did not find 'false negative' examples, however, *Mbnl1-E5*, was classified as "uncertain" with a score of 0.470, while it was reported to play a functional role during terminal erythroid differentiation in mice (Cheng, Shi et al. 2014).

Figure R1.2 Rank-ordered distribution of FAScore for all AS events in the validation set from PU-learning strategy. AS events are ranked by descending FAScore (0.00 to 1.00). The functional AS events from literature are explicitly labeled.

11. The statement: "In short, alternative splicing of TBC1D23-E15 leads to differential binding affinity to the RanBP2/RanGAP1 complex and may affect cellular SUMOylation level." is not fully supported by experiment. The raw gels indicate minimal difference in interaction and exclusively in an over-expression setting, further the term affinity is used but not measured using biophysical assays. Similarly, the impact of sumoylation needs to be further validated beyond Co-IP experiments.

Response: We endeavored to obtain recombinant proteins for RanBP2 and RanGAP1 for assays such as SPR, but unfortunately not succeeded. We therefore revised the MS to avoid statement of 'binding affinity' and rephased the wording as "*binding capacity*".

To further elucidate the differential binding capacity of the two isoforms, we repeated the co-IP assay using cell lines expressing only TBC1D23-L or -S isoform which was generated by reconstitution of the isoform expression in the

TBC1D23 full gene KO cell line and the total protein level of TBC1D23 (L or S) was comparable to the control cells (Revised **Fig. 5c, d**, see below). Using these cell lines, we showed that the level of RanBP2 and RanGAP1 bound to TBC1D23-S was 4 and 5 times higher than TBC1D23-L, respectively (Revised **Fig. 5e, f**). The additional experiment and the accompanying figures were included in the revised MS (Pg. 13, In. 356-368)

*“To investigate how TBC1D23-E15 regulates erythropoiesis, we analyzed the interactomes of long and short isoforms of TBC1D23 by an integrated multi-omics approach. We generated three in both K562 and HEK293T, one lacking TBC1D23 expression (KO) and the other two expressing only TBC1D23-L or TBC1D23-S (**Extended Data Fig. 9a, Fig 5c, d**), and performed IP-MS in the K562 cell lines (**Fig. 5a, Supplementary Table 6**). RanBP2 was identified as one of the candidate proteins showing different binding toward TBC1D23-S or TBC1D23-L (**Extended Data Fig.9b, c**). RanBP2 is a known Sumo E3 ligase that forms a complex with SUMO1-modified RanGAP1⁶¹. The immunofluorescence analysis showed that TBC1D23 co-localized with RanBP2 (**Fig 5b**). Using the TBC1D23-KO, TBC1D23-L and TBC1D23-S KEK293T cell lines overexpressing RanBP2 or RanGAP1 E3 SUMO ligase domain, we confirmed TBC1D23-S consistently precipitated more RanBP2 and RanGAP1 than TBC1D23-L by 5 and 4 times, respectively (**Fig 5e, f**) suggesting that E15 exclusion enhances engagement with the complex (**Fig 5e, f**).”*

Additionally, the SUMO proteomics analysis, presented in revised **Fig. 5i**, identified 11 hematopoietic regulators with significantly differential SUMOylation patterns in TBC1D23-L and -S expressing cells, including HDAC1-K476.

Finally, we performed additional Co-IP analysis of SUMOylated HDAC1 using bone marrow cells isolated from both WT and E15-KO mice and the results were included in the revised **Fig. 5m, n** and MS (Pg. 14, In. 401-402):

“Furthermore, HDAC1 SUMOylation was elevated by 1.5-fold in BMCs from E15-KO mice compared to those of WT controls”

Revised **Fig 5c**, Western blot analysis of TBC1D23 protein expression levels in HEK293T cells under four conditions: sgRNA-control cells, TBC1D23-KO cells reconstituted with either empty Flag vector (Flag), Flag-tagged short isoform of TBC1D23 (Flag-TBC1D23-S), or Flag-tagged long isoform of TBC1D23 (Flag-TBC1D23-L). Representative results from three independent experiments are shown. Revised **Fig 5d**, Statistical analysis of Tbc1d23 expression level. The level was determined by normalizing the image gray values in (c) (n=3).

Revised **Fig 5e**, Co-immunoprecipitation (co-IP) analysis of the binding capacity of TBC1D23-S and TBC1D23-L to the RanBP2 complex. HA-RanGAP1 and EGFP-RanBP2-E3 were co-transfected into TBC1D23-KO HEK293T cells reconstituted with Flag, Flag-TBC1D23-S, or Flag-TBC1D23-L. Cell lysates were subjected to

immunoprecipitation (IP) with anti-Flag antibody and immunoblot with anti-Flag, anti-HA, and anti-EGFP antibodies.

Revised **Fig 5f**, Images of membranes in (e) were analyzed using ImageJ software to calculate the relative intensity of bands for HA-RanGAP1 and EGFP-RanBP2-E3 in IP sample (n=3).

Revised **Fig 5i**, Heatmap displaying the abundance of proteins involved in the regulation of the hematopoiesis pathway. Abundance values are scaled, with row labels indicating protein names and SUMO-modified sites.

Revised **Fig 5m**, Co-IP analysis of the HDAC1 SUMOylation level in WT and E15-KO mice. Bone marrow cells (1×10^8) were lysed and subjected to IP with anti-HDAC1 antibody and immunoblot with anti-SUMO1 and anti-Hdac1 antibodies.

Revised **Fig 5n**, Membrane images in (m) were analyzed using ImageJ software to quantify the relative HDAC1 SUMOylation level in E15-KO mice compared to WT mice (n=3).

12. Methods are lacking as is model validation for zebrafish and mouse.

Response: We apologize for the lack of information. As shown above in point 2, we updated the 'Methods' section in the revised MS to include the validation methods.

Reviewer #2:

This manuscript describes a Random Forest based approach to score alternative splicing (AS) events based on their likelihood to have functional relevance for specific hematopoietic lineages. Based on this score (FAScore) as well as the differential regulation of sequence inclusion, they then selected four AS events, which they functionally validated. They follow up on one of them (exon 15 of TBC1D23), and elucidated the mechanistic details on how the skipping/inclusion of that exon leads to its physiological impact. Overall, I found the experimental tests and validations to be very well executed, specially for TBC1D23. The development of the FAScore is not particularly original, perhaps with the exception of making it lineage-specific (instead of "global"). However, the authors seem to then average FAScores and used them in combination with the DyScore (i.e. classic delta PSI) to select the events of interest (Fig. 2d).

1. The positive training set to develop the FAScore is of 63 events according to Sup Table 3 (even though the Methods section says 71). This number seems quite low for an RF approach. Also, looking at the references in Table S3, for

some of these events I am not sure there is actual evidence of functionality, only of protein translation by Western blot. This should be clarified. Finally, most of the events used for training are of the exon skipping type (see point 2 below). These limitations should be explicitly acknowledged in the main text.

Response: Thanks for the reviewer's constructive suggestions. We fully agree with comment regarding the training of exon-skipping (ES) type and the events with 'actual evidence'. We therefore retained only the exon skipping events in the analysis and the positive events with experimental evidence of functionality, while others with only Western blot evidence were removed (Kalsotra and Cooper 2011, Barbosa-Morais, Irimia et al. 2012, Merkin, Russell et al. 2012, Mazin, Khaitovich et al. 2021) (Revised **Supplementary Table 3**). Although the exon skipping event is the most experimentally studied in the field of alternative splicing, we have tried our best to include ES events with published functions and the resulting positive training set consists of 58 ES events with function.

We acknowledge that the size of our positive set remains limited for conventional machine-learning methods. To address this, we have employed a PU-learning (Positive-Unlabeled Learning) framework (Hao, Colak et al. 2015, Bekker and Davis 2020), which is particularly suited for scenarios where confirmed positive samples are scarce and unlabeled samples are abundant (lacking definitive negative labels). This approach is well-aligned with the constraints of our dataset. By the method, we defined a reliable positive set (4,803) and a negative set (5,245) to form the final training set. The clear separation between positive and negative sets across most features showed the robustness of the PU-learning selection and the discriminative power of the feature set (**Figure R2.1**). Moreover, the size of the training set was acceptable for the Random Forest model.

Figure R2.1. The heat map displays normalized feature scores for 4,803 reliable positive and 5,245 reliable negative AS events identified via PU-learning. Each row represents an AS event, and each column corresponds to a predictive feature, with the final FAScore. Values are z-score normalized.

2. It is unclear to me how the benchmarking results in Figures 1 and 2 and in the associated Supplementary Figures look for each type of AS event separately. All events seem to be analyzed and treated together, which I believe is potentially problematic. As mentioned above, FAScore has been mainly trained for exon skipping events and it is known that the level of conservation, functionality, etc. is very different for different types of events.

Response: We have updated the positive set of exon-skipping events by incorporating functional annotations from the literature, applied PU-learning to expand the positive set and infer the negative set, and reconstructed a prediction model specifically for exon skipping events. This updated model

demonstrates an improved performance (ROC-AUC 0.993) compared to the previous one (0.931) (Revised **Fig. 2a**).

*Revised **Fig. 2a**. Evaluation of FAScore prediction performance using two complementary metrics, including the Receiver Operating Characteristic (ROC) curve. The grey diagonal line represents the performance of a random classifier. AUCmax: the max area under the curve (AUC) value in 10-fold-validation.*

3. Another aspect I found unclear is how the authors have dealt with constitutive, cryptic and "real" AS exons. Based on the number of AS events analyzed, it seems to me they have applied their score to all exons in the genome. What does it mean if a constitutive exon has a high FAScore? Does it mean that its mis-splicing will have a functional impact (as expected)? Again, it would be good to see how the FAScore ranges for the different types of events (constitutive, AS, etc.), and how the benchmarkings perform separately per inclusion type. For instance, if all highly regulated exons (high DyScore) have high FAScores, this means the FAScore is mainly a non-useful score on its own. This score seems to categorize thousands of exons/events as functional, which is likely questionable.

Response: In our study, we focused only on alternative splicing exons without constitutive exons. Our updated methods were trained based on exon-skipping events. We do not think that it can infer the FAScores for constitutive exons,

given the level of conservation and genomic features between constitutive and AS exons are different.

Our study includes annotated ("real") and unannotated ("cryptic/novel") exons, to minimize the potential of noise, we included splice junctions that meet the following criteria: (1) independent detection by ≥ 10 uniquely mapped reads in at least two individual samples, (2) ≥ 100 uniquely mapped reads supporting the junction when summed across all samples, and (3) unannotated splice junctions required to possess one of the canonical intron motifs (GT/AG, GC/AG, or AT/AC). Next, we compared the FAScores of annotated ("real") and unannotated ("cryptic/novel") exons, which showed comparable distribution (**Figure R2.2a**).

Regarding the relationship between DyScore and FAScore, the scatter plot (**Figure R2.2b**) showed a weak correlation coefficient (0.31). Upon close inspection, we observed that the high density of events exhibiting a high FAScore and low DyScore pattern (as indicated in the red circled area), clearly demonstrating that FAScore can capture events with low DyScore that may be missed by filtering based only on DyScore. Furthermore, we plotted the AS events in the positive set, revealing that some known functional AS events had low DyScore values (**Figure R2.2c**). To further dissect whether the model is biased towards events with high DyScore, we plotted a heat map and found that FAScore and DyScore were not related (**Figure R2.2d**). These observations further underscore the necessity of using an integrated model for candidate selection, such as FAScore, rather than focus on one or two key feature(s).

Figure R2.2. Correlation between FAScore and absolute dynamic score across alternative splicing events.

a, Density curves show the distribution of FAScore values (ranging from 0.00 to 1.00) for annotated events (yellow) and unannotated events (blue). Both distributions are bimodal, with a major peak at low FAScore values (non-functional AS) and a minor peak at high FAScore values (functional AS).

b, Density plot showing the correlation between FAScore and absolute DyScore in exon skipping events, with darker regions indicating higher density of events. The red circle highlights a region containing splicing events with high FAScore but low dynamic score. Spearman's correlation coefficient and corresponding p-value are shown.

c, Scatter plot showing the correlation between FAScore and absolute DyScore. Each point represents a known functional splicing event. The top five and last five AS events are labeled.

d, Heat map displays normalized feature scores for high DyScore with high FAScore (50 were randomly selected) or low FAScore (50 were randomly selected) AS events identified via PU-learning. Each row represents an AS event, and each column corresponds to a predictive feature, with the final FAScore. Values are z-score normalized with blue indicating low values and red indicating high values.

4. Perhaps more important: the standard in the field of AS to rank candidates for experimental validations is to select evolutionarily conserved events that change their inclusion levels across lineages and/or a time course. Is the FAScore approach really much better than this? The authors selected four events with high FAScores and high DyScores (i.e. change in their inclusion levels across lineages and/or the time course). If I understood correctly, the most important contributor to the FAScore is conservation and then lineage regulation (which makes it redundant with DyScores). In fact, based on Fig. 2d, the events selected for follow up have a fairly modest FAScore (0.55-0.70) and there are at least dozens of events with much higher FAScores. Why have those events not been prioritized? In other words, the FAScore is perhaps useful, but it seems potentially an oversell.

Response: As shown above in point 9 in Reviewer 1, we found that the conservation and FAScores are weakly correlated (Revised **Extended Data Fig. 3a**), FAScores have added value in predicting functional AS events with low conservation. Indeed, the functional events with published experimental evidence such as *Kdm1a-E7*, *EHMT2-E10* have low sequence conservation however had high FAScores of 1.00 and 0.98, respectively (Revised **Extended Data Fig. 3b**), showcasing the value of FAScore in the selection of AS with low sequence conservation, which integrates multiple features, such as conservation, dynamic changes and transcript structural composition.

Regarding the selection of candidate events using FAScore and DyScore, we agree with the raised concern and we now updated the selection using species-

and cell lineage-specific FAScores. Briefly, we selected AS candidates based on FAScore on a cell lineage in human or mouse (Revised **Fig. 3d**). For each lineage, we showcased one or two events with a high FAScore: erythroid lineage, *TBC1D23-E15* (0.998) and *EPB41L1-E14* (0.995); myeloid lineage, *KLF6-E3* (0.997) and *SSBP3-E6* (0.915); lymphoid lineage, *FYN-E7* (0.987), which was reported to play an important role in T cell signal transduction in mice (Kumar, Li et al. 2021). Given their high FAScores in their specific lineage, they were then validated in the experiments.

Revised Extended Data Fig. 3. Correlation between FAScore and evolutionary conservation across alternative splicing events.

a, Density plot showing the correlation between FAScore and evolutionary conservation across species in ES events, with darker regions indicating higher density of events. The red circle highlights a region containing splicing events with high FAScore but low conservation. Spearman correlation coefficient and corresponding p-value are shown.

b, Scatter plot showing the correlation between FAScore and evolutionary conservation score across species. Each point represents a known functional splicing event.

Revised Fig. 3d. Rank-ordered distribution of functional AS scores for exon skipping events in five clusters (11, 8, 22, 5, 12). AS events are ranked by descending FAScore (0.00 to 1.00). Each point represents an AS event. The number of SE events in each cluster is indicated in parentheses. The candidate AS events are highlighted in red. The marginal miniatures in the lower-left corners depict the average FAScore trend of events within the cluster across four hematopoietic contexts: fetal hematopoietic organ (FHO), erythroid (Ery), lymphoid (Lym), and myeloid (Mye) lineages. Event sizes (N) for each cluster are annotated below the miniatures.

5. What gene sets were used as background for the GO enrichment analyses? In the Methods section it is unclear if they have used any background at all. If not, it is possible that the enrichments they obtained were due to lineage-specific gene expression biases, which are known to be a major confounder for AS analyses and could explain why they fit the lineage functions well.

Response: We appreciate the reviewer's valuable suggestion. The background set for all GO enrichments comprised all genes with GO annotations in the organism-specific database (org.*.eg.db). However, we fully acknowledge the reviewer's concern that this default background may introduce lineage-specific expression biases, potentially confounding AS-based functional enrichment.

To address this, we performed GO enrichment analyses using a new background gene set consisting of genes that were expressed (TPM \geq 0.5) in at least one sample in the lineage studied. Lineage-specific functional signatures were still exhibited in the lineage-restricted background (Revised **Fig. 2e**). For example, genes harbouring predicted functional AS in erythroid lineage consistently enriched “erythrocyte differentiation”. Genes with predicted functional AS in enriched in myeloid cell differentiation and neutrophil activation. We revised the ‘Methods’ section (Pg. 29, In 823-827) to explicitly state:

“Gene Ontology (GO) enrichment analysis was performed separately for each lineage using the R package clusterProfiler⁹⁵ (v4.2.2). For each lineage, the set of genes harboring functional and uncertain SE events was tested for enrichment against a background set comprising all genes that were expressed (TPM \geq 0.5) in at least one sample within one lineage to control for cell lineage-specific expression biases”

Revised **Fig. 2e**. Bubble plot shows the GO pathways analysis of functional AS-harboring genes using cell lineage-specific background sets across FHO development and lineage differentiation (including erythroid, lymphoid, and myeloid) in human (Left)

and mice (Right). The size of each bubble corresponds to the number of genes in the input list that are mapped to a particular GO term. The x-axis represents the significance of enrichment (p -value < 0.05).

6. The zebrafish morpholino experiment is much weaker than all the other experiments. In my opinion, it is OK to keep these results, but I would move them after the first results in mouse (i.e. “white blood cell counts remained unchanged”) and introduce them in a very specific manner, e.g.: “To assess the conservation of this phenotype, we”

Response: Thanks for the constructive suggestion. We have moved the zebrafish results after the results in mouse. And revised the MS as follows (Pg. 11-12, ln. 318-326).

“...E15-KO mice exhibited significant reductions in red blood cell (RBC) numbers, hematocrit (HCT) and hemoglobin (HGB) levels compared to the WT littermate controls while white blood cell counts remained unchanged (Fig. 4e, f). In addition, to assess the conservation of this phenotype, we targeted the long isoform of Tbc1d23 in zebrafish (Tbc1d23-L, including E15) by morpholino (MO) treatment. The Tbc1d23-L-MO (E15-MO) treatment specifically reduced the mRNA level of the long isoform while increasing the expression of the short isoform, with no significant effect on total Tbc1d23 protein level comparable to the fish treated with the control MO (Ctrl-MO) (Fig 4g-i, Extended Data Fig. 7g)...”

7. In summary, I believe the experimental and mechanistic aspects of this manuscript are much stronger than the FAScore, which is perhaps not needed. My recommendation is to downplay this part, and simply sell it as a way to prioritize exons for experimental follow up. As it is now, it has potential flaws and its relevance is oversold, which may require substantial work to solve.

Response: As mentioned, we updated our model using 'true' positive set and focusing on only exon-splicing events and the updated FAScore demonstrated improved precision and accuracy. We believe FAScore is valuable in identification of interesting AS such as TBC1D23-E15 which has no defined domains or motifs and events like *Kdm1a-E7*, *EHMT2-E10* which have low sequence conservation.

Reviewer #3:

Hu et al. surveyed the transcriptome of the hematopoietic system in six vertebrate species and trained a machine learning model to identify functional lineage-specific alternative splicing events. Their approach relies on a Random Forest model trained on a large dataset of differentially spliced events between lineages and leveraging positive unlabeled learning to take advantage of a set of 71 validated functional events curated from the literature. The interpretation of the model yielded a set of 19 features which were predictive for alternative event functionality. The model was then applied to identify several candidate functional events in lymphoid, myeloid and erythroid commitment. The authors go on to show that one of their candidates, a vertebrate cassette inclusion event in TBC1D23, is important for erythropoiesis using *in vivo* and *ex vivo* experiments in the zebrafish and mouse hematopoietic cells. They finally provide mechanistic evidence to demonstrate that the inclusion of the alternative TBC1D23 exon 15 in the erythroid lineage decreases binding to its partner complex RANBP2/RANGAP1, likely leading to the activation of transcription factors involved in erythroid differentiation through lower SUMOylation of HDAC1.

This study tackles the extent and the role of alternative splicing during hematopoiesis. This is an important and poorly studied problem with wide ranging applications in fundamental and translational immunology. Notably, the manuscript presents three valuable components to the field: 1) a detailed

survey of alternative splicing events across hematopoietic lineages, between species and between developmental stages; 2) a predictive model for splicing event functionality; 3) the characterization of a novel alternative event in TBC1D23 that is key during erythropoiesis.

The methodology leading to the development of the FAScore and DyScore is reasonable and robust. The methods for the experimental validation of candidate functional events are generally sound, but key experiments and details lack in some sections, in particular in demonstrating the lineage specific of TBC1D23 exon 15 inclusion and differential SUMOylation of HDAC1, as described below.

1. The astronomical number of differential/lineage-specific splicing events reported and differentially expressed genes in the TBC1D23-E15KO mouse is hard to believe and likely obscures some of the most biologically relevant events. This could be due to lenient filtering criteria and/or specific tools and methods used for transcriptomic analyses. Depending on the version and parameters used, SUPPA2 can have a very high false-positive rate. The choice of performing scRNA-Seq rather than FACS followed by bulk RNA-Seq to identify DEGs in the TBC1D23-E15KO is puzzling. The authors should at least comment on these points and possibly consider revisiting their filtering criteria.

Response: We thank the reviewer for the comments and suggestions. As we used 10x Genomics 3' single-cell RNA-seq followed by NGS, which provides only short reads at the 3' end of the transcripts for differential gene expression analysis; we therefore did not perform differential splicing analysis. We agree that the version and parameters of SUPPA2 can affect the identification of the number of splicing events, we therefore used SUPPA2 only to identify the type of splicing events according to the GTF annotation files (e.g., exon skipping, alternative 5'/3' splicing, etc.). The large numbers of AS events are the overall counts of all exon-skipping exons from multiple lineages and species.

Regarding the differential gene expression analysis between the WT and TBC1D23-E15 KO mouse in the scRNA-seq, the 5,915 unique differentially expressed genes (DEGs) across all 20 cell types were identified using commonly-used single-cell thresholds in Seurat (adj. $p < 0.05$ & $|\text{avg_log2FC}| \geq 0.25$), with the Pro-E cells showing the highest number of DEGs (3,346, **Figure R1a**), suggesting the KO mouse may have the largest difference in Pro-E cells when compared to WT mouse.

To assess whether the filtering criteria affect our results, we implemented two stricter thresholds: $|\text{avg_log2FC}| \geq 0.58$ (2,492 DEGs, **Figure R3.1b**) and $|\text{avg_log2FC}| \geq 1$ (802 DEGs, **Figure R3.1c**). GO enrichment analysis confirmed relatively consistent pathway signatures (**Figure R3.1d**) across all three thresholds (23 pathways, including critical processes like erythrocyte differentiation, **Figure R3.1e**). We agree that the $|\text{avg_log2FC}| \geq 0.25$ cutoff is lenient, we therefore used $|\text{avg_log2FC}| \geq 1$ as the threshold.

Regarding our choice of scRNA-Seq rather than FACS followed by bulk RNA-Seq, the reasons are as follows: 1) we wanted to analyze the cell-types influenced most by TBC1D23-E15 KO and to compare with FAScore prediction, and such information is better provided by scRNA-seq than bulk RNA-seq dependent on the sorted cell-types; 2) scRNA-seq enables us to perform transcript factor regulatory networks analysis but not bulk RNA-seq data. The analysis provides potential networks of TBC1D23-E15 KO, this may provide insights into the underlying mechanisms TBC1D23-E15 regulation.

Figure R3.1 Comparison of different threshold values of differentially expressed genes (DEGs) analysis across cell types.

a-c, Bar plots quantifying cell-type-specific DEGs counts under increasing fold-change thresholds (adjusted p.value < 0.05): $|\log_2FC| \geq 0.25$ (a), $|\log_2FC| \geq 0.58$ (b), $|\log_2FC| \geq 1.0$ (c). Cell types are color-coded, and the number of DEG is labeled on the bar.

d, Venn diagram illustrating the overlapping of significantly enriched GO pathways of DEGs in three thresholds (Biological Processes, adjusted p-value < 0.05).

e, Dot plot displaying the overlapped pathways in GO-BP enrichment for DEGs identified in (d). The erythroid-related pathway is colored in red.

2. A lot of the evidence cited to support assumptions is anecdotal. For example, the identification of FYN-E5 and PIK3R1-E1 among high FAScore and DyScore events is used as a demonstration that high FAScore and DyScore can be used to find lineage-specific functional events, but how many known lineage-specific functional events can be found among low FAScore/DyScore events?

Response: As mentioned, Reviewer 2 point 1, we agreed that the positive set should only include 'actual' functional events with experimental evidence and therefore updated the positive set (Revised **Supplementary Table 3**). Based on the updated positive set, we revised a prediction model specifically for exon skipping. Moreover, by previously Reviewer 2's comment (point 4), we revised the selection criteria of candidate AS to rank AS events only based on the FAScore in different species without DyScore.

Regarding the question of how many known functional events have low FAScores (i.e., potential false negatives), we included all known functional AS events with published experimental evidence in the training and validation set. The updated model was trained using reliable positive and negative sets derived via a PU-learning framework, with 20% of the data reserved for validation. The trained model achieved a false negative rate of 0.05. As shown in **Figure R3.2**, the true positive set in the validation set have high FAScores. Among these, *Mbnl1-E5* predicted to have a score with 0.470 was classified as uncertain, although it has been previously reported in the literature to play a functional role during terminal erythroid differentiation in mice (Cheng, Shi et al. 2014)

Figure R3.2 Rank-ordered distribution of FAScore for all AS events in validation set from PU-learning strategy. AS events are ranked by descending FAScore (0.00 to 1.00). The functional AS events from literature are explicitly labeled.

3. The experiments in Figure 2E-G must be better described. Currently it is impossible to tell from the main text or figure legend that these experiments use mouse progenitor cells that were transduced with exon targeting gRNAs and then sorted by lineage to perform the colony forming assays. Even the methods do not specify that the experiment was performed in mouse cells and what mouse strain were used. There should also be a panel showing baseline exon inclusion levels by RT-PCR in the control GM, GEMM and BFU-E lineages.

Response: Following the reviewer’s suggestion, we have updated the corresponding Results, Figure legends, and ‘Methods’ section.

For Results (Pg. 10-11, ln. 270-292):

“SSBP3-E6 and KLF6-E3 are more frequently included in the myeloid lineage (average PSI: 0.27 and 0.92, respectively) compared to the erythroid (0.08 and 0.77) and lymphoid (0.10 and 0.75) lineages (Fig. 3e). In contrast, TBC1D23-E15 is more likely to be included in the erythroid lineage (average

PSI: 0.78), while tends to be skipped in the lymphoid (average PSI: 0.32) and myeloid (average PSI: 0.39) lineages (**Fig. 3e**). We first validated that all of the candidate exons were all transcribed in the colony-forming unit-granulocyte-macrophage (GM), the colony-forming unit-granulocyte-erythrocyte-monocyte-megakaryocyte (GEMM) and the burst-forming unit-erythrocyte (BFU-E) colonies derived from colony-forming unit (CFU) assays (**Extended Data Fig. 6c**). To validated that they are functional in lineages which they have high FAScores, we performed CFU assays on the mouse HSPCs in which the candidate exons were deleted using GFP-expressing retroviral vector having two sgRNAs targeting the flanking intronic regions of the exons (**Fig. 3f**). The transduced cells with GFP expression were sorted and plated for CFU assays. The CFU assays revealed that the deletion of SSBP3-E6 or KLF6-E3 in mouse *c-kit*⁺ cells resulted in a reduced formation of myeloid colonies by 58% and 38% respectively, while the erythroid colonies were of wildtype level (**Fig. 3g**). Similarly, the CFU assays showed that the deletion of the *Tbc1d23*-E15 inhibited erythroid colony formation by 35% without a significant impact on myeloid populations (**Fig. 3g**). *EPB41L1*-E14 which exhibits a higher inclusion rate in mature myeloid cells (**Fig. 3e**) and the deletion of *Epb41l1*-E14 markedly enhanced erythropoiesis with an increase of 38% of BFU-E colonies while no significant difference on the myeloid colonies was observed (**Fig. 3g**). In summary, FAScore accurately identified AS events play important roles in lineage fate commitment.”

For Figure legends (Pg. 61-62):

“Fig.3 | FAScore predicts key AS events that are dynamically regulated during lineage commitment.

e, Schematic tree view of mean Percent Spliced In (PSI) values for *EPB41L1* exon 14 (*EPB41L1*-E14), *TBC1D23* exon 15 (*TBC1D23*-E15), *KLF6* exon 3 (*KLF6*-E3), *SSBP3* exon 6 (*SSBP3*-E6) skipping events derived from RNA-seq data of hematopoietic differentiation in human. HSC: Hematopoietic stem cell, MPP: Multipotent progenitor, CLP: Common lymphoid progenitor,

CMP: Common myeloid progenitor, GMP: Granulocyte monocyte progenitor, MEP: Megakaryocyte erythrocyte progenitor, EB: Erythroblasts, MK: Megakaryocytes, Neu: Neutrophil, Mo: Monocyte, Mp: Macrophage, Mp_act: Activated macrophage, Mp_inf: Inflammatory macrophage, B: B cell, CD4: CD4⁺ T cell, CD8: CD8⁺ T cell, NK: Natural killer cell.

f. Schematic of CRISPR/Cas9-mediated exon deletion in mouse c-kit⁺ cells. c-kit⁺ cells isolated from 8-week female Rosa-cas9 knock-in mice were transduced with retroviruses expression two sgRNAs targeting the upstream intron (5' sgRNA) and downstream intron (3' sgRNA) intronic regions flanking the target exon (indicated by scissors) (Top). PCR was performed using exon-KO-F and exon-KO-R primers to confirm exon deletion. Genome PCR analysis of targeted deletion: Epb4111 exon 4 deletion (Epb4111ΔE14), Klf6 exon 3 deletion (Klf6ΔE3), and Ssbp3 exon 6 deletion (Ssbp3ΔE6), and Tbc1d23 exon 15 deletion (Tbc1d23ΔE15). The red arrows indicate the correct bands, and all bands were confirmed by Sanger sequencing (Bottom).

g. Erythroid-myeloid differentiation potential of 5000 control or mutant c-kit⁺ cells (Epb4111ΔE14, Tbc1d23ΔE15, Klf6ΔE3, and Ssbp3ΔE6) from f). Colony types quantified: BFU-E, colony-forming unit-granulocyte-macrophage (GM), and colony-forming unit-granulocyte-erythrocyte-monocyte-megakaryocyte (GEMM) colonies (n = 3 per group)."

For Methods (Pg.33, ln. 964; Pg. 40, ln. 1156-1165; Pg. 34, ln. 976-985):

"Blood routine, cell isolation and flow cytometry analysis. ...HSPCs were further enriched using c-kit/CD117 MicroBeads (Miltenyi Biotec, #130-09224)..."

"Generation of exon-deleted and overexpression mouse HSPCs. The target exon of Tbc1d23, Ebp4111, Ssbp3, and Klf6 were deleted by two sgRNAs (listed in **Supplementary Table 8**) were cloned into pMSCV-U6-

sgRNA-IRES-GFP, which was modified from pMSCV-IRES-GFP (Addgene, plasmid #20672). The coding sequence of HDAC1 and HDAC1^{mut} were cloned into pMSCV-IRES-GFP for overexpression.

The resulting constructs were cotransfected with retroviral packaging plasmids (pCL-Eco, pVSV-G) to generate retroviral particles as described above. The viral supernatants were harvested and added into c-kit⁺ cells of Cas9⁺ or WT mice with 8 µg/mL polybrene. At 24 h after transfection, the target cells were enriched by GFP.”

“Colony-forming unit assay. *Freshly isolated whole BM cells, enriched c-kit⁺ cells or sorted GFP⁺ cells were plated in methylcellulose culture medium (MethoCult™, STEMCELL Technologies) and incubated at 37°C with 5% CO₂. For cells plated on MethoCult™ GF M3334, colonies were counted and scored as colony-forming unit erythroid progenitor cells (CFU-E) at 48 h after plating. Cells plated on MethoCult™ GF M3436 were evaluated 10–14 days after plating and scored as burst-forming unit erythroid progenitor cells (BFU-E). For cells plated on MethoCult™ GF M3434, colonies derived from BFU-E and granulocyte-macrophage progenitor cells (CFU-GM, CFU-G and CFU-M), multipotent granulocyte, erythroid, macrophage, megakaryocyte progenitor cells (CFU-GEMM) were counted 9–12 days after plating.”*

Additionally, regarding using RT-PCR to show the baseline exon inclusion levels of Tbc1d23, Epb4111, Klf6, Ssbp3 in GM, GEMM, and BFU-E colonies, we washed down the colonies from plates of *MethoCult™ GF M3434* using PBS and sorted the cells from these clones by FACS and performed RT-PCR to determine the inclusion levels of these exons in these cells. In revised **Extended Data Fig. 6c**, we confirmed that the transcription of these AS events (with exon inclusion) in the corresponding lineages.

Revised Extended Data Fig. 6c. Quantification of Percent Spliced In (PSI%) values for *Klf6* exon 3, *Ssbp3* exon 6, *Tbc1d23* exon 15, and *Epb4111* exon 14 was performed by RT-PCR analysis in the following cell populations isolated from colony-forming unit (CFU) colonies: *c-kit*⁺ cells, megakaryocytes (MK), erythroid burst-forming units (BFU-E), macrophages, and granulocytes.

4. The specificity of TBC1D23 exon 15 inclusion is not shown in any of the figures. This is a critical piece of missing data. Evidence should be provided in the form of RT-PCR on several sorted human and/or mouse hematopoietic cell populations.

Response: As suggested, we performed RT-PCR using sorted mouse hematopoietic cell populations, and the PSIs collated well with the PSIs derived from bulk RNA-seq analysis of mouse HSPCs (Revised **Extended Data Fig. 7b, c**).

Revised **Extended Data Fig. 7b**, Box plot of PSI values for *Tbc1d23-E15* skipping events derived from RNA-seq data of hematopoietic differentiation in mouse. LT-HSC: Long-time hematopoietic stem cell, ST-HSC: Short-time hematopoietic stem cell, MPP: Multipotent progenitor, CLP: Common lymphoid progenitor, CMP: Common myeloid progenitor, GMP: Granulocyte monocyte progenitor, MEP: Megakaryocyte erythrocyte progenitor, Ery: Erythroblasts, MK: Megakaryocytes, Mo: Monocyte, Mp: Macrophage, B: B cell, CD4: CD4+ T cell, CD8: CD8+ T cell, NK: Natural killer cell.

Revised **Extended Data Fig. 7c**, Differential expression of spliced transcripts was validated in the mouse hematopoietic cells through RT-PCR and corresponding agarose gel electrophoresis. Calculate the Percent Splice In (PSI) value of each lane was calculated by dividing the gray value of the longer transcript by the sum of the gray value of the longer and shorter transcripts.

5. Does the MO in Fig 3A-B also affect the levels of the TBC1D23 exon skipping transcript? Currently, it is impossible to confirm that the phenotype is due to loss of TBC1D23-E15 specifically or loss/decrease of all TCB1D23 expression altogether.

Response: To address the concern, we measured total TBC1D23 protein levels in MO-treated embryos and found no significant change in total TBC1D23 protein expression level (Revised **Fig. 4h, i**), suggesting E15-MO primarily affects the expression of the long isoform but not the exon-skipping short isoform.

Revised **Fig. 4h**, Immunoblot analysis of total *Tbc1d23* protein levels in zebrafish embryos following Ctrl-MO and E15-MO treatment.

Revised **Fig. 4i**, Images of membranes in (h) were analyzed using ImageJ software to calculate the relative protein level of TBC1D23 ($n=3$).

6. Supplementary Fig 5D should include a panel showing TBC1D23 isoform ratios by RT-PCR using a single primer pair in flanking exons.

Response: We appreciate the reviewer's suggestion. To clarify, the RT-PCR analysis of TBC1D23 isoform ratios (using primers in flanking exons) was presented in the original Fig. 3D. However, we acknowledge that this may have caused confusion with the genotyping data. As suggested, we have now restructured the figures as follows: (1) The genotyping PCR results were included in revised **Fig. 4a**, (2) the original RT-PCR isoform-ratio data were moved to revised **Fig. 4b**, and (3) the qPCR results quantifying total *Tbc1d23* mRNA levels were moved to revised **Extended Data Fig. 7d**.

Revised **Fig. 4a**, Schematic representations of the WT and E15-KO alleles, with arrows indicating primer binding sites and the expected PCR product sizes labeled (Top) and genotyping of E15-KO mice (Bottom).

Revised **Fig. 4b**, RT-PCR validated the specific deletion of the exon 15-containing isoform (*Tbc1d23-L*) in E15- KO mice.

Revised **Fig. 7f**. RT-qPCR analysis of *Tbc1d23-L* (left), and *Tbc1d23-S* (right) mRNA levels in WT and E15-KO mice.

7. Nowhere is the difference in HDAC1 SUMOylation between TBC1D23-E15KO and WT shown. This is another critical piece of missing data.

Response: As suggested, we performed an additional Co-IP analysis of SUMOylated HDAC1 using bone marrow cells isolated from both WT and E15-KO mice and the results were included in the Revised **Fig. 5m-n** and MS (Pg. 14, ln. 401-402):

“Furthermore, HDAC1 SUMOylation was elevated by 1.5-fold in BMCs from E15-KO mice compared to those of WT controls (Fig 5m, n).”

Revised Fig. 5m, Co-IP analysis of the HDAC1 SUMOylation level in WT and E15-KO mice. Bone marrow cells (1×10^8) were lysed and subjected to IP with anti-HDAC1 antibody and immunoblot with anti-SUMO1 antibody.

Revised Fig. 5n, Membrane images in (m) were analyzed using ImageJ software to quantify the relative HDAC1 SUMOylation level in E15-KO mice compared to WT mice ($n=3$).

8. TBC1D23 E15 may alter interaction with partners, but the changes shown in co-IP experiments with RANBP2/RANGAP1 are rather small and downstream changes in HDAC1 SUMOylation are not quantified; how can the loss of a TBC1D23 isoform be expected to lead to the differential expression of close to 6,000 genes?

Response: As shown above in point 7, we now confirmed the HDAC1 SUMOylation level was increased by 50% in the bone marrow cells of E15-KO mice. To address the concern about interaction the changes of interaction between TBC1D23 and RANBP2/RANGAP1, we performed an additional Co-IP assay using cells expressing only long or short isoform, which were generated by expressing long or short isoform in the full gene KO cells and the expression level was analyzed by Western blot showing the overall protein

expression level was comparable as control cells (Revised **Fig. 5c, d**). Quantitative analysis revealed a 4-fold higher RANBP2 binding by the short versus long isoform under these physiological conditions (Revised **Fig. 5e, f**). This difference was substantially attenuated under overexpression conditions (only 1.5-fold difference, $p < 0.01$), confirming that physiological expression levels are crucial for revealing authentic isoform-specific binding properties.

Regarding to the transcriptional changes, we repeated the analysis using more stringent cut-off (FDR < 0.05 , fold-change ≥ 2) and identified 802 differentially expressed genes between E15 KO and WT cells. As HDAC1 is a master epigenetic regulator, and it has been reported in the literature that SUMOylation of HDAC1 alters its biochemical activity (David, Neptune et al. 2002) and enhances its transcriptional repression mechanisms (Yang and Sharrocks 2004), it is reasonable that the 50% upregulated level of HDAC1 SUMOylation leads to scalable changes downstream.

Revised **Fig. 5c**, Western blot analysis of TBC1D23 protein expression levels in 293T cells under four conditions: sgRNA-control cells, TBC1D23-KO cells reconstituted with either empty Flag vector (Flag), Flag-tagged short isoform of TBC1D23 (Flag-TBC1D23-S), or Flag-tagged long isoform of TBC1D23 (Flag-TBC1D23-L). Representative results from three independent experiments are shown.

Revised **Fig. 5d**, Statistical analysis of Tbc1d23 expression level. The level was determined by normalizing the image gray values in (c) ($n=3$).

Revised **Fig. 5e**, Co-immunoprecipitation (co-IP) analysis of the binding capacity of TBC1D23-S and TBC1D23-L to the RanBP2 complex. HA-RanGAP1 and EGFP-RanBP2-E3 were co-transfected to TBC1D23-KO 293T cells reconstituted with Flag, Flag-TBC1D23-S, or Flag-TBC1D23-L. Cell lysates were subjected to immunoprecipitation (IP) with anti-Flag antibody and immunoblot with anti-Flag, anti-HA, and anti-EGFP antibodies.

Revised **Fig. 5f**, Images of membranes in (e) were analyzed using ImageJ software to calculate the relative intensity of bands for HA-RanGAP1 and EGFP-RanBP2-E3 in IP sample ($n=3$).

9. Line 209: “classed” should be “classified”.

Response: We apologize for the mistake. It was corrected in the revised MS

“These AS events were then classified as primate-specific (age class 1), mammal-specific (class 4) or ancient (present in all vertebrates, class 5, dating back 429 million years ago (MYA) (**Fig. 3a**)

References

- Barbosa-Morais, N. L., M. Irimia, Q. Pan, H. Y. Xiong, S. Gueroussov, L. J. Lee, V. Slobodeniuc, C. Kutter, S. Watt, R. Colak, T. Kim, C. M. Misquitta-Ali, M. D. Wilson, P. M. Kim, D. T. Odom, B. J. Frey and B. J. Blencowe (2012). "The evolutionary landscape of alternative splicing in vertebrate species." *Science* **338**(6114): 1587-1593.
- Bekker, J. and J. Davis (2020). "Learning from positive and unlabeled data: a survey." *Machine Learning* **109**(4): 719-760.
- Cheng, A. W., J. Shi, P. Wong, K. L. Luo, P. Trepman, E. T. Wang, H. Choi, C. B. Burge and H. F. Lodish (2014). "Muscleblind-like 1 (Mbnl1) regulates pre-mRNA alternative splicing during terminal erythropoiesis." *Blood* **124**(4): 598-610.
- David, G., M. A. Neptune and R. A. DePinho (2002). "SUMO-1 modification of histone deacetylase 1 (HDAC1) modulates its biological activities." *J Biol Chem* **277**(26): 23658-23663.
- Fiszbein, A., L. E. Giono, A. Quaglino, B. G. Bernardino, L. Sigaut, C. von Bilderling, I. E. Schor, J. H. Enrique Steinberg, M. Rossi, L. I. Pietrasanta, J. J. Caramelo, A. Srebrow and A. R. Kornblihtt (2016). "Alternative Splicing of G9a Regulates Neuronal Differentiation." *Cell Rep* **14**(12): 2797-2808.
- Hao, Y., R. Colak, J. Teyra, C. Corbi-Verge, A. Ignatchenko, H. Hahne, M. Wilhelm, B. Kuster, P. Braun, D. Kaida, T. Kislinger and P. M. Kim (2015). "Semi-supervised Learning Predicts Approximately One Third of the Alternative Splicing Isoforms as Functional Proteins." *Cell Rep* **12**(2): 183-189.
- Hunsicker, M. E., C. V. Kappel, K. A. Selkoe, B. S. Halpern, C. Scarborough, L. Mease and A. Amrhein (2016). "Characterizing driver-response relationships in marine pelagic ecosystems for improved ocean management." *Ecol Appl* **26**(3): 651-663.
- Kalsotra, A. and T. A. Cooper (2011). "Functional consequences of developmentally regulated alternative splicing." *Nat Rev Genet* **12**(10): 715-729.
- Kumar, P., L. Li and J. H. Schatz (2021). PI3K/AKT Activation Mediates B-Cell Transformation By the " T" Splice-Variant of Fyn Kinase, American Society of Hematology Washington, DC.
- Love, M. I., W. Huber and S. Anders (2014). "Moderated estimation of fold change and dispersion for RNA-seq data with DESeq2." *Genome Biol* **15**(12): 550.
- Mazin, P. V., P. Khaitovich, M. Cardoso-Moreira and H. Kaessmann (2021). "Alternative splicing during mammalian organ development." *Nat Genet* **53**(6): 925-934.

Merkin, J., C. Russell, P. Chen and C. B. Burge (2012). "Evolutionary dynamics of gene and isoform regulation in Mammalian tissues." Science **338**(6114): 1593-1599.

Wang, T., Y. Cao, Q. Zheng, J. Tu, W. Zhou, J. He, J. Zhong, Y. Chen, J. Wang, R. Cai, Y. Zuo, B. Wei, Q. Fan, J. Yang, Y. Wu, J. Yi, D. Li, M. Liu, C. Wang, A. Zhou, Y. Li, X. Wu, W. Yang, Y. E. Chin, G. Chen and J. Cheng (2019). "SEN1-3 Signaling Controls Mitochondrial Protein Acetylation and Metabolism." Mol Cell **75**(4): 823-834.e825.

Yang, S. H. and A. D. Sharrocks (2004). "SUMO promotes HDAC-mediated transcriptional repression." Mol Cell **13**(4): 611-617.

Zeddies, S., S. B. Jansen, F. di Summa, D. Geerts, J. J. Zwaginga, C. E. van der Schoot, M. von Lindern and D. C. Thijssen-Timmer (2014). "MEIS1 regulates early erythroid and megakaryocytic cell fate." Haematologica **99**(10): 1555-1564.

Zhao, Y., M. C. Li, M. M. Konate, L. Chen, B. Das, C. Karlovich, P. M. Williams, Y. A. Evrard, J. H. Doroshov and L. M. McShane (2021). "TPM, FPKM, or Normalized Counts? A Comparative Study of Quantification Measures for the Analysis of RNA-seq Data from the NCI Patient-Derived Models Repository." J Transl Med **19**(1): 269.

Zibetti, C., A. Adamo, C. Binda, F. Forneris, E. Toffolo, C. Verpelli, E. Ginelli, A. Mattevi, C. Sala and E. Battaglioli (2010). "Alternative splicing of the histone demethylase LSD1/KDM1 contributes to the modulation of neurite morphogenesis in the mammalian nervous system." J Neurosci **30**(7): 2521-2532.

Response to Reviewers

MS Submission ID: NCOMMS-25-31345A

MS title: The functional landscape of alternative splicing in hematopoietic lineage commitment

Dear reviewers:

We thank the reviewers for their valuable comments and suggestions. In response to the comments, we performed additional analysis and revised the manuscript (the changes are shown in **red font**) and updated the accompanying figures and tables. We addressed the concerns point-to-point under each comment. And the changes that we introduced into the revised manuscript (MS) are shown in *italics* with page (Pg.) and line (Ln.) numbers in the revised MS indicated.

Yours sincerely,

Lu Chen, PhD

Reviewer #2

Overall, the authors have addressed my initial concerns; however, while doing so (and in response to the other reviewers), I have to say that they have often created more confusion. Specific comments:

1. Abstract: "We curated transcriptome data of fetal hematopoietic organ development in six vertebrates and hematopoietic cell differentiation in human and mice, and identified 84,933 cassette events, originating from 30,255 coding proteins." I found this sentence confusing. Checking the main text, it seems it refers to protein-coding genes (as cassette exons do not originate from proteins). Thus, I assume this includes all species together (as there are fewer than 30,255 protein-coding genes in human or mouse alone), which is misleading without further context. Also, the exons are likely non-independent (i.e., orthologous?). This is all too complex for the abstract, in my opinion.

Response: We thank the reviewer for pointing this out and great suggestion. We have added an explicit supplementary table (*Revised **Supplementary Table 2***) listing the number of genes and exons for a specific differentiation lineage and species. The figures of 84,933 cassette exons and 30,255 protein-coding genes were obtained by first performing lineage-level deduplication within each species and then summing the counts across different species. We agree that this is too complex, we have revised the sentence

in the Abstract (Pg. 2, Ln. 35-39) as follows:

"We curated transcriptomic data on fetal hematopoietic organ development in six vertebrates and hematopoietic cell differentiation in humans and mice. To identify functional exon-skipping events among thousands of cassette exons in protein-coding genes for a specific differentiation lineage and species, we developed a machine-learning model..."

Table S2 No. of genes or AS events for a specific differentiation lineage and species

Species	Lineage	No. of Gene	No. of AS	No. of FuncAS	No. of NonFuncAS	No. of Uncertain	No. of Gene (Func)	No. of Gene (NonFunc)	No. of Gene (Uncertain)
HomSap	Erythroid	7594	23737	2938	5902	14897	1818	3207	6110
HomSap	FHO	8935	25152	1635	11986	11531	1113	5728	5656
HomSap	Lymphoid	8595	28824	4764	7393	16667	2790	3706	6786
HomSap	Myeloid	9015	32400	5481	7529	19390	3075	3783	7287
MacMul	FHO	5702	12280	820	3208	8252	666	2053	4386
MusMus	Erythroid	6292	13954	2744	4537	6673	1894	2782	3892
MusMus	FHO	6244	13623	3325	3941	6357	2098	2607	3690
MusMus	Lymphoid	6405	14357	4992	2417	6948	3110	1636	3952
MusMus	Myeloid	6101	13448	1893	4173	7382	1336	2652	4123
RatNor	FHO	3578	7489	1893	1697	3899	1101	1088	2171
DanRer	FHO	1824	2953	140	671	2142	113	513	1372

Revised Supplementary Table 2. Number of genes or AS events for a specific differentiation lineage and species.

2. In the revised version they claim they focus on 58 experimentally validated cassette exons. However, looking at their updated supplementary table (again), there are only 37 unique exons, as many entries involve the same exon coordinate. And these unique exons are actually even fewer, since the human and mouse orthologs are often present in the table. This does not affect the training per se, as this set is only used to define the positive and training sets. However, this is a major reporting mistake, to say the least.

Response: We have now corrected **Supplementary Table 3** by consolidating orthologous exons across species and lineages as well as exons with identical genomic coordinates, clearly identifying 37 unique cassette exons. Corresponding clarifications have also been added to the main text and Methods section to prevent any potential misunderstanding.

In the 'Results' section (Pg. 7, Ln. 170-173):

*"Due to the lack of a verified negative set and the limited number of known functional AS events with experimental evidence (37 unique exons, with redundancies due to cross-species conservation and multi-lineage functionality, **Supplementary Table 3**, See **Methods**)..."*

In the 'Methods' section (Pg. 23, Ln. 662-666):

*"A total of 37 uniquely exon-skipping events were experimentally verified as functional in organ development or hematopoietic differentiation, among 2 AS events observed in two species and 15 observed in multiple lineages or organs (**Supplementary Table 3**), adding up to 58 positive instances. These were used to train model totally."*

developed a machine-learning model interrogating 19 features including dynamic expression, protein structure, and evolutionary conservation, and integrated them into a single prediction score, named Functional AS Score (FAScore). ”

(Pg. 6, Ln. 165-170)

“Using these features, we developed a machine-learning model named FAScore (Functional AS Score) to predict functional AS events in a given differentiation lineage and species (**Fig. 1a**). For each AS event, the model assigns a species- and lineage-specific FAScore, designated as species_lineage_FAScore (e.g., Human_erythroid_FAScore, Mouse_erythroid_FAScore).”

(Pg. 10, Ln. 269-274)

“Next, we selected candidates with top FAScores from different cell lineages of mouse or human for experimental validation (**Fig. 3d**). For the erythroid lineage, TBC1D23-E15 (Human_erythroid_FAScore: 0.998) and EPB41L1-E14 (Mouse_erythroid_FAScore: 0.995) were chosen. For the myeloid lineage, KLF6-E3 (Human_myeloid_FAScore: 0.997) and SSBP3-E6 (Mouse_myeloid_FAScore: 0.915) were chosen. And for the lymphoid lineage, FYN-E7 (Mouse_lymphoid_FAScore: 0.987) was selected (**Extended Data Fig. 6b**).”

Revised **Fig. 3d**, Rank-ordered distribution of functional AS scores for ES events in five clusters (11, 8, 22, 5). AS events are ranked by descending FAScore (0.00 to 1.00). Each point represents an AS event. The number of ES events in each cluster is indicated in parentheses. The candidates AS events are highlighted in red.

Revised **Extended Data Fig. 6b**, Rank-ordered distribution of functional AS scores of humans and mice for ES events in cluster 12. AS events are ranked by descending FAScore (0.00 to 1.00). Each point represents an AS event. The number of ES events in each cluster is indicated in parentheses. The candidate AS event is highlighted in red.

4. A specific case of this confusing use can be seen in Fig. 3b: what are really the data points in each boxplot? The FAScores for each age class for each differentiation lineage? I.e. 4 data points? This would be fine, but it is not clear in the current text and it is not explained in the caption. (Another confusing aspect about this plot: why are there two dots in some cases? Also, I would recommend plotting the proportion of funct/non-funct/uncertain for each age category not the other way around.)

Response: The data points in each boxplot represent the proportions of AS exons for different age classes in each category (FuncAS, NonfuncAS, or Uncertain) in four lineages. Each boxplot summarizes four data points, and we have explicitly showed each data point using shape (**Figure R1**). Furthermore, following the reviewer's suggestion, we have revised this figure that illustrates the proportions of funct/non-funct/uncertain AS exons within each age category (**Fig. 3b**).

Figure R1. The percentage of AS exons for different age classes in each category (functional, non-functional, or uncertain) in four lineages in human. Each data point

represents the percentage of AS events for an age class within a specific category in one lineage.

Revised **Fig. 3b**. Proportion of AS exons categorized as functional, non-functional, or uncertain across distinct evolutionary age classes of exons in different lineages of human.

5. Another source of confusion: "we measured the FAScores for 188,202 annotated and unannotated AS events". Since they replied to my initial comment that they discard constitutive exons, I assume these 188,202 events are for a mix of species or otherwise the numbers do not add up just for humans. Or are they double counting the metrics for each FAScore type?

Response: As the reviewer correctly understood, 188,202 events indeed represents the total number of AS events across different species and differentiation lineages used in this study, We have added a revised **Extended Data Fig. 4b** and revised **supplementary table 2**. (Pg. 7-8, Ln. 199-207):

*"Next, we measured the FAScores for annotated and unannotated AS events from fetal hematopoietic organ development and hematopoietic cell differentiation per species (**Supplementary Table 2**), that were not used in the model training deriving a continuous score for AS, ranging from 0 to 1, with 1 as most likely to be functional. By fitting a three-component GMM, we defined the following classes by probability density intersection method: 'functional AS' (FuncAS), 'non-functional AS' (NonfuncAS), and 'uncertain AS' (Uncertain) (**Extended Data Fig. 4a**). In total, we predicted 30,615 functional AS (16.3%), 53,454 non-functional (55.3%), and 104,133 uncertain (28.4%) (**Extended Data Fig. 4b, Supplementary Table 2**)."*

Revised **Extended Data Fig. 4b**. Stacked bar plot showing the functional classification of AS events across all datasets. The y-axis indicates the proportion of events. Colors correspond to categories as functional, non-functional, or uncertain. The number in plot indicates the number of AS events per category.

Species	Lineage	No. of Gene	No. of AS	No. of FuncAS	No. of NonFuncAS	No. of Uncertain	No. of Gene (Func)	No. of Gene (NonFunc)	No. of Gene (UnCertain)
HomSap	Erythroid	7594	23737	2938	5902	14897	1818	3207	6110
HomSap	FHO	8935	25152	1635	11986	11531	1113	5728	5656
HomSap	Lymphoid	8595	28824	4764	7393	16667	2790	3706	6786
HomSap	Myeloid	9015	32400	5481	7529	19390	3075	3783	7287
MacMul	FHO	5702	12280	820	3208	8252	666	2053	4386
MusMus	Erythroid	6292	13954	2744	4537	6673	1894	2782	3892
MusMus	FHO	6244	13623	3325	3941	6357	2098	2607	3690
MusMus	Lymphoid	6405	14357	4992	2417	6948	3110	1636	3952
MusMus	Myeloid	6101	13448	1893	4173	7382	1336	2652	4123
RatNor	FHO	3578	7489	1893	1697	3899	1101	1088	2171
DanRer	FHO	1824	2953	140	671	2142	113	513	1372

Revised **Supplementary Table 2**. Number of genes or AS events for a specific differentiation lineage and species.

6. DyScore: this is now more intuitively explained in the first section of Results. However, as above, since the DyScore is specific to a given lineage differentiation (Erythroid, Lymphoid or Myeloid), it should be clear which lineage they are referring to. For example, I assume the cases in Fig. 1 relate only to Erythroid differentiation. If so, this should be more clearly state, and call it e.g. DyScore_Erythroid to avoid confusion. If my understanding it is not correct, then the authors need to clarify this well to avoid misunderstandings.

Response: We have added relevant lineage information to the description of the dynamic features, particularly for the cases presented in Fig. 1. The specific modifications are as follows:

(Pg. 5, Ln.138-139)

“We quantified six features of the dynamic gene expression of MEIS1 in human erythroid lineage.”

Revised Fig1b-e.

b, Schematic representation of dynamic profiling for gene expression and alternative splicing (AS). A linear fit (blue dashed line) and a Loess curve (green solid line) are shown, with the shaded area representing the 95% confidence interval. Derived features include: range of genes (Gene.range) or AS (AS.range) to assess variation breadth; correlation coefficient (Gene.cor, AS.cor) and its p-value (Gene.cor.p, AS.cor.p) to quantify association with temporal or differentiation trajectories; linear regression slope (Gene.slope, AS.slope) and its p-value (Gene.slope.p, AS.slope.p) to estimate rate and significance of change; and tau (Gene.tau, AS.tau) to evaluate specificity.

c-e, Expression and splicing patterns of MEIS1 and its two AS events during human erythroid differentiation. (c) Expression values (transformed TPM) of MEIS1 (Left) and the values of dynamic features (Right). Percent spliced in (PSI) values (Left) and dynamic feature scores (Right) for the exon 12 skipping event in transcript MEIS1-201 (d) and the exon 1 skipping event in transcript MEIS1-202 (e). A linear fit (blue dashed line) and a Loess curve (green solid line) are shown, with the shaded area representing the 95% confidence interval.

7. Perhaps more relevant, since the DyScore is integrated into the FAScore and not used anymore beyond Fig. 1, why is it even needed?

Response: We reckon DyScore is a easier for readers to understand, providing a single value to measure the level of dynamic change. These features of DyScore is indeed integrated into the overall FAScore, we calculated the DyScore primarily to quantify the dynamicity of genes/AS events. This quantification facilitates the dynamic categorization and comparison of genes/AS events, thereby helping to elucidate the importance of dynamic changes for functional prediction. Besides, DyScore can characterize the dynamic degree at both the gene and AS levels, providing a useful reference even for those focusing solely on gene-level functionality.

Minor:

8. Fig 2c: the FAScore_Lymphoid seems to do quite poorly. Is it the case?

Response: We thank the reviewer for this thoughtful observation. Our previous figure showed the difference in splicing site conservation between functional and non-functional AS events across various lineages. We update the figure by showing the original conservation score from functional and non-functional AS events. As shown in the figure below, consistent with all other lineages studied, functional AS events in the lymphoid lineage also exhibit significantly higher conservation around their splicing sites compared to non-functional AS events (Revised **Fig. 2c**). The lesser difference in the lymphoid lineage is due to the conservation level for non-functional AS events in this lineage is notably higher. We have revised the Fig. 2c to improve the clarity.

Revised Fig. 2c. Comparison of conservation scores (PhastCons) surrounding splice sites between functional and non-functional AS of annotated type in different lineages. The 3'SS

and 5'SS refer to the 3' and 5' splice sites, respectively. The x-axis indicates the position (bp) relative to the splice site, with "0" representing the splice site, while the regions to the left and right are intronic (100 bp) and exonic (50 bp). The y-axis shows the average PhastCons scores for each site of functional and non-functional AS.

9. Extended data Fig. 6: as far as I understood, Mfuzz was designed to cluster continuous (temporal) data series, not unrelated classes as those here. If so, this implies that the order in which the independent classes are provided could affect the results, which would not make sense.

Response: The Mfuzz package, which based on fuzzy c-means clustering, was designed for time-series or continuous gradient data, and its algorithm considers the order of data points. In our study, we used the core fuzzy c-means clustering algorithm (Hathaway and Bezdek 1986). Notably, fuzzy c-means algorithm is fundamentally a general-purpose machine learning method that does not mandatory require time- or order-based continuity between data points, and can be used for category data. The essence of this algorithm is to cluster data points based on their "distance" in a multidimensional space, which is independent of whether the dimensions represent time points. In our analysis, we treated different independent classes (FHO, erythroid, lymphoid and myeloid) as distinct dimensions in the multidimensional space, and the clustering aimed to identify groups of AS events with similar functional patterns across these dimensions.

To ensure the robustness and objectivity of our results, we altered the input order of the categories and re-ran the clustering. We found that the AS composition of each cluster and their function trend patterns remained highly consistent (**Figure R2**). This indicates that our clustering results are robust and their biological interpretation is not dependent on the input order of the categories.

To better match the intent of our analysis, we have replaced the original continuous line chart with a categorical box plot adding a red line which connects the median values of each category to clearly outline the overall trend across the different groups (**Revised Extended Data Fig. 6a**). This change was made to more accurately represent the categorical nature of our data and to avoid confusion of continuity of different lineage.

Figure R2. Comparison of fuzzy c-means clustering profiles under different category input orders. Shown are the patterns of four representative clusters (5, 8, 11, 22). Data point labels correspond to the number of AS events per cluster.

Revised Extended Data Fig. 6a. The distribution of the normalized average FAScore of ancient AS events (age class 5) for four hematopoietic lineages in 24 clusters from fuzzy *c*-means clustering. Plot numbers indicate the number of AS events assigned to each of these clusters. A red line connects the median values of the lineages, delineating the overall function trend across the four lineages for that cluster.

Reviewer #3

Hu et al. provided a substantially revised version of their manuscript. Although none of the initial findings or claims had to be revisited, there were important pieces of evidence and technical details missing from the original submission. The authors appropriately clarified or modified the methodology to address the reviewers' comments. They also provided key results to support their initial claims. Most notably, they explained better how they came to such a large amount of differentially expressed/spliced genes and re-adjusted their analysis parameters to highlight the most striking changes, they revisited their prediction model for exon skipping, they provided baseline quantifications of TBC1D23 isoforms across all relevant hematopoietic cell types, they added missing controls in many experiments, and they substantiated their claims linking the change in TBC1D23 isoform ratios to HDAC1 SUMOylation through the RanGAP1-RanBP2 complex with standard biochemical and biophysical assays.

1. There are the multiple typos and incorrect statements (e.g. a mention of "30,255 coding

proteins" in the abstract - but what are "coding proteins"?) that populate the text and that can easily be fixed.

We believe that the revised manuscript delivers the main message of the authors - that they were able to draft a summary of functional alternative splicing events during hematopoiesis - much more successfully than the original manuscript.(Remarks on code availability)

Response: We appreciate the reviewer's suggestion and apologize for the mistake. We have now thoroughly proofread the entire manuscript to correct all typos and inaccurate statements. Specifically, we have revised the sentence in the Abstract (Pg. 2, Ln. 35-39) as follows:

" We curated transcriptomic data on fetal hematopoietic organ development in six vertebrates and hematopoietic cell differentiation in humans and mice. To identify functional exon-skipping events among thousands of cassette exons in protein-coding genes for a specific differentiation lineage and species, we developed a machine-learning model..."

References

Hathaway, R. J. and J. C. Bezdek (1986). "Local convergence of the fuzzy c-Means algorithms." Pattern Recognition **19**(6): 477-480.